# Basal conditions of Denman Glacier from glacier hydrology and ice dynamics modeling

Koi McArthur[1], Felicity S. McCormack[2], and Christine F. Dow[1,3]

[1]Department of Applied Mathematics, University of Waterloo, Waterloo, Canada
[2]Securing Antarctica's Environmental Future, School of Earth, Atmosphere & Environment, Monash University, Australia
[3]Department of Geography and Environmental Management, University of Waterloo, Waterloo, Canada

**Correspondence:** Koi McArthur (kr2mcarthur@uwaterloo.ca)

**Abstract.** Basal sliding in Antarctic glaciers is often modeled using a friction law that relates basal shear stresses to the effective pressure. As few ice sheet models are dynamically coupled to subglacial hydrology models, variability in subglacial hydrology associated with the effective pressure is often implicitly captured in the basal friction coefficient – an unknown parameter in the basal friction law. We investigate the impact of using effective pressures calculated from the Glacier Drainage System (GlaDS) model on basal friction coefficients calculated using inverse methods in the Ice-sheet and Sea-level System Model (ISSM) at Denman Glacier, East Antarctica, for the Schoof and Budd friction laws. For the Schoof friction law, a positive correlation emerges between the GlaDS effective pressure and basal friction coefficient in regions of fast ice flow. Using GlaDS effective pressures generally leads to smoother basal friction coefficients and basal shear stresses, and larger differences between the simulated and observed ice surface velocities, compared with using an effective pressure equal to the ice overburden pressure plus the gravitational potential energy of the water. Compared with the Budd friction law, the Schoof friction law offers improved capabilities in capturing the spatial variations associated with known physics of the subglacial hydrology. Our results indicate that ice sheet model representation of basal sliding is more realistic when using direct outputs from a subglacial hydrology model, demonstrating the importance of coupling between ice sheet and subglacial hydrological systems. However, using our outputs we have also developed an empirical parameterization of effective pressure that improves application of the Schoof friction law without requiring explicit hydrological modeling.

## 1 Introduction

The health of Antarctic glaciers and their future susceptibility to climate-driven change is often assessed by the retreat rates of their grounding lines and melt rates of adjacent ice shelves. In the East Antarctic, Denman Glacier of the Denman-Scott catchment (Fig. 1) has seen some of the fastest grounding line retreat of the last 20 years, losing 5.4 km since 1996 (Brancato et al., 2020), and the highest ice shelf melt rates of $116\,\mathrm{m\,a^{-1}}$ from 2010-2018 (Adusumilli et al., 2020). Perched on a grounding line above a deep trough that descends 3.66 km below sea level (Morlighem, 2020), Denman Glacier is potentially at risk of marine ice sheet instability and rapid retreat, which is of significant concern given that the glacier drains a region containing 1.5 m of sea level equivalent (Brancato et al., 2020).

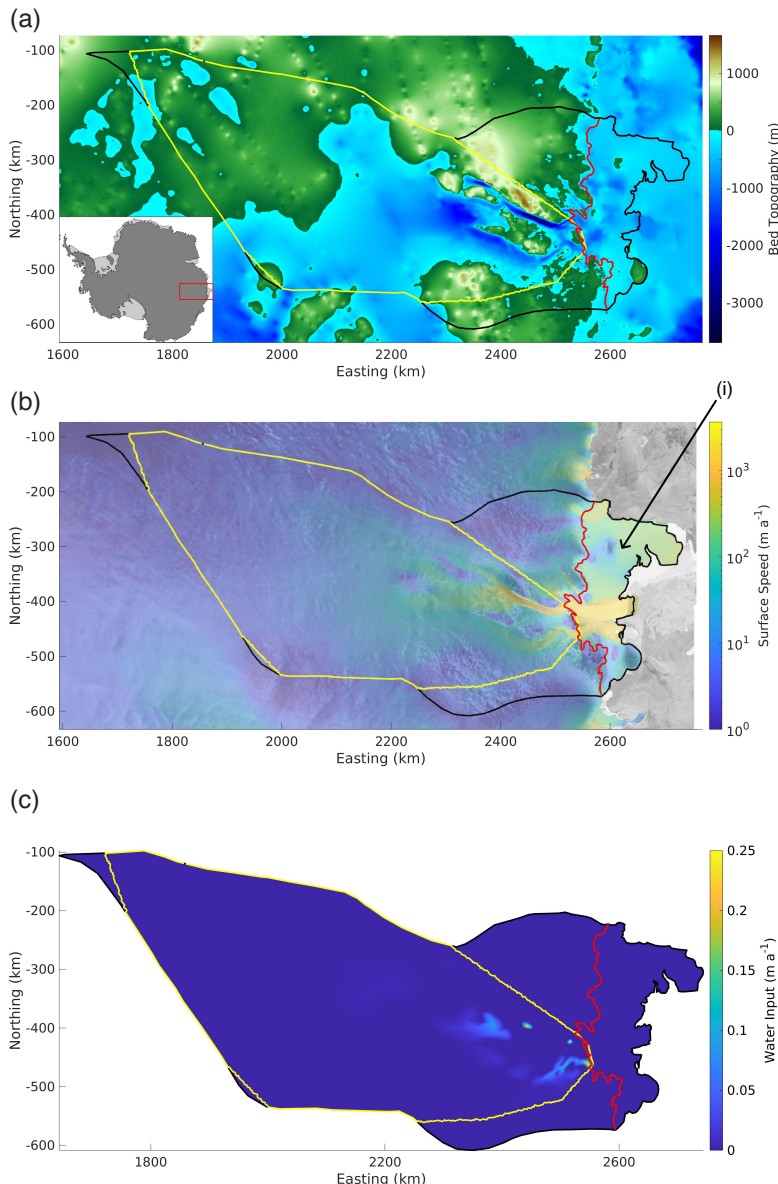

**Figure 1.** Denman-Scott catchment. **(a)** Bed topography (m) from Bechmachine v2 (Morlighem, 2020); **(b)** Ice surface speed (m a$^{-1}$) from MEaSUREs v2 (Rignot et al., 2011, 2017; Mouginot et al., 2012, 2017), using a logarithmic color scale; **(c)** Basal meltwater production $\eta$ (m a$^{-1}$) from (Seroussi et al., 2019); **(i)** The Shackleton Ice Shelf. The black lines in both panels show the ice catchment outline, defined by the drainage divide and calving front, the grounding line is shown in red, and the GlaDS domain in yellow. The $x-$ and $y-$axes are eastings and northings, defined in polar stereographic coordinates referenced to WGS84. These maps were made using the Antarctic Mapping Tools (Greene et al., 2017) and RAMP Radarsat Antarctic Mapping Project (Greene, 2022) toolboxes for MATLAB.

Determining the future of Denman Glacier, and others in the Antarctic, largely relies on ice dynamics models to capture the evolution of ice flow under changing climates. These models frequently use inversion techniques to constrain important parameters that control ice velocity, including those related to the basal environment beneath the ice sheet (Morlighem et al., 2013). One such parameter is the basal friction coefficient, which is a key component of friction laws including the Weertman (1957), Budd et al. (1979) and regularized Coulomb friction laws (Schoof, 2005; Gagliardini et al., 2007). Friction laws propose a functional dependency of the basal shear stress on the basal sliding velocity and in the case of the Budd and regularized Coulomb friction laws, the effective pressure $N = p_i - p_w$, which is defined as the ice overburden pressure ($p_i$) minus the subglacial water pressure ($p_w$). However, if the basal shear stress does not actually have the functional dependence on the basal sliding velocity and effective pressure proposed in the basal friction law, or if it has a functional dependence on other properties of the bed such as roughness, substrate, or temperature, then this will be implicitly captured in the basal friction coefficient. Therefore, a spatially variable basal friction coefficient suggests a friction law which either fails to capture the proper functional dependency on basal sliding velocity and effective pressure, or omits the dependency of the basal shear stress on other quantities. By consequence, a basal friction coefficient that is both smooth – has little local variability – and has limited domain-wide trends is desirable. In the search for ways of describing basal friction Beaud et al. (2022) proposed a generalized friction law to be used on both rigid and deformable beds. They suggest that the unknown parameters in their friction law can be found using ice surface velocity time series and inversion techniques on a variety of glaciers implemented at the global scale.

Antarctic subglacial hydrology is increasingly being shown to be varied and dynamic, with large channels that discharge into ice shelf cavities (Dow et al., 2020; Indrigo et al., 2021), and effective pressures that vary spatially both above and below zero (i.e. when subglacial water pressure exceeds the ice overburden pressure). Modeling of subglacial hydrology has shown close links between regions of low effective pressure and high ice surface velocity (Dow et al., 2018, 2020, 2022); this control of basal water pressure on sliding rates is also well known for Greenland and Alpine glacier systems (Nienow et al., 2017; Iken and Bindschadler, 1986).

Despite the key role of basal boundary conditions, and in particular subglacial water pressure, on ice dynamics, there has not yet been a systematic investigation of the impact of the effective pressure on basal sliding in different friction laws applied in the Denman-Scott catchment. Given the critical role of the friction law for accurate ice dynamics and sea level predictions in the future (e.g. Åkesson et al., 2021; Yu et al., 2018), this is an important missing link. Here, we address this by using the Glacier Drainage System (GlaDS; Werder et al., 2013) model to characterize the effective pressure and subglacial hydrology of the Denman Glacier. We use these effective pressures as inputs in the Budd and Schoof friction laws implemented in the Ice-sheet and Sea-level System Model (ISSM; Larour et al., 2012), and calculate the resulting basal friction coefficients using inversion.

## 2 Methods

### 2.1 GlaDS Setup

GlaDS is a 2D finite element subglacial hydrology model that calculates the development of a distributed water system on the elements, and channel growth, fed by the distributed system on the element edges (Werder et al., 2013). The model runs are initiated with no channels and therefore channel formation is dictated entirely by the evolving pressure and flow conditions across the model elements. GlaDS model equations are described in full in Werder et al. (2013) and its application to other Antarctic systems is covered in detail in Dow et al. (2018, 2020, 2022). The model parameters used in our simulations are summarized in Table 1. The bedrock bump height, cavity spacing, and sheet conductivity act together in the distributed system equations to either constrict or open the system to water flow as show in :

$$w - \tilde{A}h|N|^{n-1}N + \frac{\partial h_e}{\partial t} + \overrightarrow{\nabla} \cdot \overrightarrow{q} = \eta \tag{1}$$

$$\overrightarrow{q} = -kh^{\alpha}|\overrightarrow{\nabla}\phi|^{\beta-2}\overrightarrow{\nabla}\phi \tag{2}$$

Where $\tilde{A}$ is the rheological constant of the ice multiplied by an order one factor depending on the cavity geometry, $h$ is the hydrology sheet thickness, $N$ is the effective pressure, $n$ is the exponent in Glen's flow law, $h_e$ is the englacial storage, $t$ is time, $\eta$ is a prescribed source term shown in Fig. (1c), $\overrightarrow{q}$ is the hydrology sheet discharge, $k$ is the sheet conductivity, $\phi$ is the hydraulic potential, and $\alpha = 5/4$ and $\beta = 3/2$ are exponents in the Darcy-Weisbach law describing fully turbulent flow. $w$ is the hydrology sheet opening rate equal to $u_b(h_r - h)/l_r$ when $h < h_r$ and zero otherwise, here $u_b$ is the basal ice speed, $h_r$ is the typical bedrock bump height and $l_r$ is the typical cavity spacing.

As discussed in Dow (2023), when the system is overconstricted (i.e. it is difficult for water to flow through the hydrologic system) the pressures are unrealistically high – much of the domain is above ice overburden pressure – and the model fails to converge. When the system is underconstricted (i.e. water flows through the hydrologic system with ease) the pressures are well below ice overburden pressure for much of the domain (much of the domain is below 50 % of ice overburden pressure), which is unrealistically low for steady-state Antarctic systems that are not driven by surface water input. The variables controlling the constriction of the hydrologic system are $k$, $h_r$, and $\eta$, with a more constricted system arising from larger $\eta$ and smaller $k$ and $h_r$. We test order of magnitude changes in $k$ to determine a suitable level of constriction of the system. While there is some variation within the range of acceptable pressures, the output we present is the median and therefore is the most appropriate for representing the hydrologic pressure in ice sheet dynamics equations without further information from in situ measurements for example. Future work with full coupling of hydrology and ice dynamics can explore sensitivity to different distributed system inputs. Channel conductivity is, similarly, a median value applied in GlaDS in Antarctica (Dow et al., 2020). The ice flow constant is set for an average ice column temperature of -10 °C.

The domain for the GlaDS model is a subset of the grounded portion of the Denman-Scott catchment based on the hydraulic potential (at overburden) catchment for this region (Fig. 1). The mesh includes 17,700 nodes and the domain-wide mean edge length is 3.1 km. However, we refine the mesh to a minimum edge length of 200 m in regions where basal sliding velocities

are greater than $40\,\mathrm{m\,a}^{-1}$ and along the Denman Glacier grounding line. The basal and surface topography used in the GlaDS run is from BedMachine version 2 (Morlighem, 2020). Basal water and sliding velocity inputs are taken from the JPL_ISSM ISMIP6 Antarctic control run final time step (Seroussi et al., 2019) which was a thermal steady state simulation using the enthalpy formulation implemented in ISSM (Seroussi et al., 2013) and the Blatter-Pattyn (BP) approximation to the full Stokes equations (Blatter, 1995; Pattyn, 2003). This model solved for ice viscosity of the floating ice shelf and the basal friction coefficient using data assimilation techniques (MacAyeal, 1993; Morlighem et al., 2010), which differs from our application of ISSM to assess basal boundary conditions. However, lacking an alternative starting point for basal sliding and velocity, this is the best available option prior to full coupling between hydrology and ice dynamics in ISSM. The sliding velocity acts to open up distributed system cavities and, at velocities greater than $800\,\mathrm{m\,a}^{-1}$, can cause model instabilities and so is capped at this value. Tests of similar caps for model runs of winter conditions at Helheim Glacier (Poinar et al., 2019) demonstrate it has little impact on the effective pressure. We assume a temperate bed throughout, although regions with zero water input (in the interior or southernmost-regions of the domain) are not an active part of the hydrological system. The model is run for 10,000 days to near steady state (there are changes in the water sheet thickness in deep pockets on the order of $\mathrm{mm\,day}^{-1}$), providing outputs including channel size and discharge, distributed system discharge, water depth, and effective pressure.

| Parameter | Value | Units |
|---|---|---|
| Bedrock bump height ($h_r$) | 0.08 | m |
| Cavity spacing ($l_r$) | 2 | m |
| Channel conductivity ($k_c$) | $5 \cdot 10^{-2}$ | $\mathrm{m^{3/2} \cdot kg^{-1/2}}$ |
| Englacial void ratio ($e_\nu$) | $10^{-5}$ | |
| Glen's flow constant ($n$) | 3 | |
| Ice density ($\rho_i$) | 910 | $\mathrm{kg \cdot m^{-3}}$ |
| Ice flow constant ($\tilde{A}$) | $2.5 \cdot 10^{-25}$ | $\mathrm{Pa^3 \cdot s^{-1}}$ |
| Sheet conductivity ($k$) | $1 \cdot 10^{-4}$ | $\mathrm{m^{7/4} \cdot kg^{-1/2}}$ |
| Sheet width below channel ($l_c$) | 2 | m |

**Table 1.** GlaDS model parameters described in detail in Werder et al. (2013).

## 2.2 ISSM Setup

ISSM is a finite-element model that uses a non-uniform mesh to simulate ice dynamics. We employ the inverse capabilities within ISSM to estimate basal friction coefficients in the Denman-Scott catchment using various basal friction laws, as described below. The inverse model uses the shallow-shelf approximation (SSA; MacAyeal, 1989; Morland, 1987) to the full Stokes equations, described in Eq. (3) with $\bar{\mu}$ the depth-averaged effective viscosity $\mu$ which is given in Eq. (4). The SSA is

described in full in Larour et al. (2012).

$$\frac{\partial}{\partial x}\left(4H\bar{\mu}\frac{\partial u}{\partial x}+2H\bar{\mu}\frac{\partial v}{\partial y}\right)+\frac{\partial}{\partial y}\left(H\bar{\mu}\frac{\partial u}{\partial y}+H\bar{\mu}\frac{\partial v}{\partial x}\right)=\rho_i gH\frac{\partial S}{\partial x}$$
$$\frac{\partial}{\partial y}\left(4H\bar{\mu}\frac{\partial v}{\partial y}+2H\bar{\mu}\frac{\partial u}{\partial x}\right)+\frac{\partial}{\partial y}\left(H\bar{\mu}\frac{\partial u}{\partial y}+H\bar{\mu}\frac{\partial v}{\partial x}\right)=\rho_i gH\frac{\partial S}{\partial y}$$
(3)

$$\mu=\frac{\tilde{B}}{2\dot{\varepsilon}_e^{\frac{1-n}{n}}}$$
(4)

Where $u$ and $v$ are the $x$ and $y$ components of ice velocity respectively, $H$ is the ice thickness, $\rho_i$ is the density of ice, $g$ is the absolute value of the gravitational acceleration, $S$ is the surface elevation, $\dot{\varepsilon}_e$ is the effective strain rate tensor, $n$ is the exponent in Glen's flow law, and $\tilde{B}$ is the ice rigidity.

The domain for the ISSM model is the full Denman-Scott catchment, extending the boundary used for the GlaDS simulations to include the floating ice shelves (Fig. 1). The ISSM mesh is comprised of 66,518 nodes, with non-uniform mesh refinement for faster flowing ice using the MEaSUREs v2 ice surface speed from (Rignot et al., 2011, 2017; Mouginot et al., 2012, 2017), described in Appendix A2. The minimum mesh edge length is 500 m. We note that the GlaDS mesh used for the subglacial hydrology simulations differs from that used in the ISSM simulations for the ice dynamics as a result of differences in ice and hydrology drainage catchments; the impact of this is discussed in Appendix A1. We test the sensitivity of the basal friction inversion simulations to various different meshes, finding that the results are less sensitive to the effect of the mesh configuration than to the specification of effective pressure and basal friction law used (Appendix A2). The bed topography and surface elevation are from BedMachine v2 (Morlighem, 2020). The ice rigidity is calculated using inverse methods (Appendix B).

### 2.2.1 Solving for basal friction coefficients

We investigate the impact of the prescription of the effective pressure in the Budd and Schoof friction laws by calculating basal friction coefficients using inversion. The scalar form of the Budd friction law is given by the following expression:

$$\tau_b = \boldsymbol{\alpha}^2 N u_b^m,$$
(5)

where $\tau_b$ (Pa) is the basal shear stress, $\alpha$ ($\mathrm{s^{1/2}\,m^{-1/2}}$) is the basal friction coefficient, $N$ (Pa) is the effective pressure, $u_b$ ($\mathrm{m\,a^{-1}}$) is the basal sliding speed, and $m = 1$ is a power law exponent taken to be linear as in Åkesson et al. (2021); Åkesson et al. (2022); Yu et al. (2018); Choi et al. (2021); Baldacchino et al. (2022). The scalar form of the Schoof friction law can be written as:

$$\tau_b = \frac{\boldsymbol{C}^2|u_b|^{m-1}}{\left(1+(\boldsymbol{C}^2/(C_{\max}N))^{1/m}|u_b|\right)^m}u_b,$$
(6)

where $C$ ($\mathrm{kg^{1/2}\,m^{-2/3}\,s^{-5/6}}$) is the friction coefficient, $m$ is the power law exponent, here given by $m = 1/3$, and $C_{\max}$ ($\mathrm{m^{-1/3}\,s^{1/3}}$) is Iken's bound, which is the effective cap on the basal shear stress (Appendix A5; Iken, 1981).

For each friction law, we calculate inversion for the basal friction coefficients by minimizing a cost function that includes the contributions of absolute and logarithmic misfits between the observed and simulated ice surface speeds (Appendix B; Morlighem et al., 2013). Weighting parameters for the absolute and logarithmic misfit terms for the various friction laws are

shown in Appendix B, Table B1. We also add a Tikhonov regularization term to the cost function, which regulates the local variability in the basal friction coefficient by penalizing large gradients in the basal friction coefficient, making the basal friction coefficient smoother. The weighting parameter of the Tikhonov regularization term is determined using an L-curve analysis (Appendix B; Hansen, 2000) which allows us to find a basal friction coefficient with small local variability while maintaining a good absolute and logarithmic velocity misfit. The final weighting parameters for the Tikhonov regularization term for each

friction law are reported in Table B1.

Though it is not the primary focus of this work, we invert for ice rigidity as well, while initializing our model for the inversion for the basal friction coefficients. By inverting for the ice rigidity we capture ice rheological processes which are not explicitly accounted for in the model such as damage, anisotropy, chemical impurities, and liquid water.

We perform the following inversion procedure. First, we invert for the ice rigidity over the floating portion of the domain.

Next, we invert for the Budd basal friction coefficient over the grounded portion of the domain, using an ice rigidity on grounded ice specified by the Paterson function from Cuffey and Paterson (2010) and surface temperatures from RACMOv2.3 (van Wessem et al., 2018). After the Budd inversion we invert for the ice rigidity over the entire domain. We next use the basal friction coefficient estimated using the Budd friction law to compute an initial estimate of the basal friction coefficient for the Schoof friction law. We perform inversions for the basal friction coefficients of the Budd and Schoof friction laws with the ice

rigidity from the inversion prior, these are the main simulations discussed in the text that follows (it is worth noting that the Budd friction coefficient converges to the result of the initital Budd friction coefficient inversion). We perform a final rigidity inversion over the entire domain. The cost functions to be minimized for each inversion are described in detail in Appendix B.

We compare the difference in the basal friction coefficients when we use two different prescriptions for the effective pressure. (1) An effective pressure given by assuming water pressure equals the ice overburden pressure plus the gravitational potential

energy of the water $N_O = \rho_i g H + \rho_w g B$. Here $\rho_i$ is the density of ice, $\rho_w$ is the density of sea water, $g$ is the absolute value of the gravitational acceleration, $H$ is the ice thickness, and $B$ is the bed elevation which takes negative values below sea level. (2) The effective pressure taken directly from the GlaDS simulations, which we refer to as $N_G$. We cap the effective pressure at 1 % of ice overburden pressure for Budd runs and 0.4 % of ice overburden pressure for Schoof runs, due to numerical artefacting that arises for values smaller than this. The impact of these choice of caps is discussed in Appendix A3. Inverting for the

basal friction coefficient using a hydrology model output effective pressure and a friction law which explicitly depends on the effective pressure is similar to the work of Koziol and Arnold (2017).

## 3 Results

We present the results of the GlaDS modeling followed by the inversion results using ISSM, and compare the impact of using $N_G$ and $N_O$ for the Schoof and Budd friction laws.

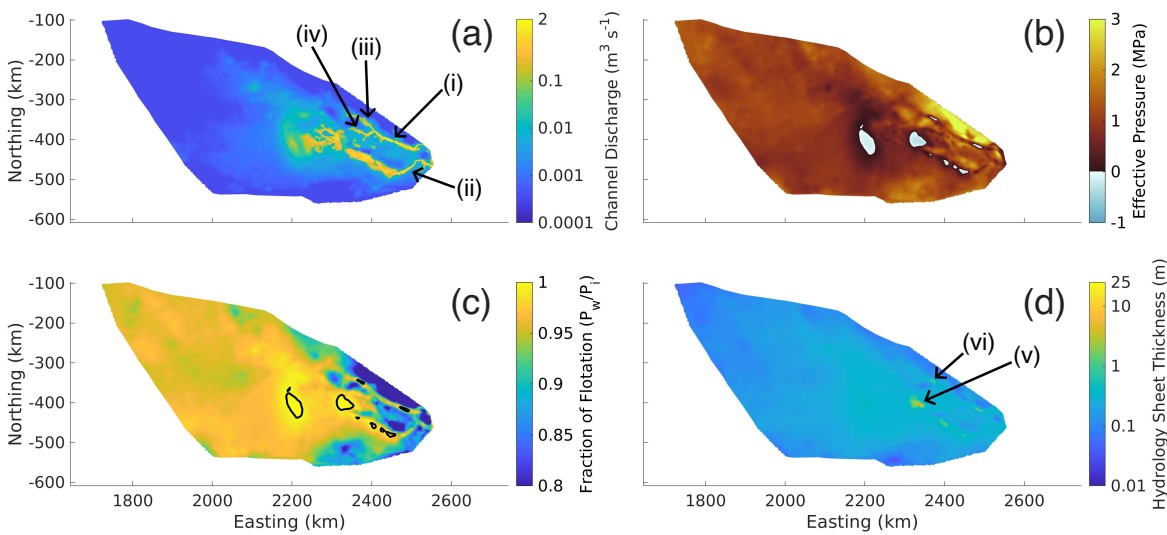

**Figure 2.** GlaDS simulation results. **(a)** Channel discharge ($m^3 s^{-1}$); **(b)** effective pressure (MPa); **(c)** fraction of flotation (i.e. the fraction of ice overburden pressure that each grid cell is pressurized to, $p_w/p_i$) with a black contour when the fraction reaches 1 (flotation); and **(d)** water sheet thickness (m). The locations marked in panel **(a)** are the: **(i)** Denman channel; **(ii)** Scott channel; **(iii)** western branch of the Denman channel; **(iv)** eastern branch of the Denman channel. The locations marked in panel **(d)** are the: **(v)** basin like feature; and **(vi)** lake like feature.

## 3.1 Subglacial hydrology

The GlaDS modeling indicates that major subglacial hydrology channels form in the Denman-Scott catchment as seen in Fig. 2a, with significant discharge through both the Denman (Fig. 2a (i)) and Scott Glaciers (Fig. 2a (ii)). For the former, the discharge is $15.8 \, m^3 s^{-1}$ and for the latter, $6.0 \, m^3 s^{-1}$. The Denman channel initiates as two branches that flow for 80 and 52 km through subglacial valleys before merging at the beginning of the significantly overdeepened trough. From here, the channel flows through the base of the trough, emerging 129 km downstream at the grounding line. The Scott channel, on the other hand, converges as a single channel 95 km from the grounding line. There is substantial water convergence with a maximum depth of 25 m in a basal depression (Fig. 2d (v)) that feeds the channels of both the Denman and Scott Glaciers, although with the strongest flux towards the eastern branch of the Denman channel (Fig. 2a (iii)). The bed topography of this basin feature lies at 1900 m below sea level (Fig. 1a), and subglacial water flows upslope by approximately 1200 m to drain downstream. A second 'lake-like' feature feeds the western branch of the Denman channel (Fig. 2a (iv)) and reaches a water depth of $\sim$8 m (Fig. 2d (vi)).

As effective pressure is the ice overburden pressure minus the subglacial water pressure, low effective pressure implies high subglacial water pressure, with negative effective pressure implying that the subglacial water pressure is greater than the ice

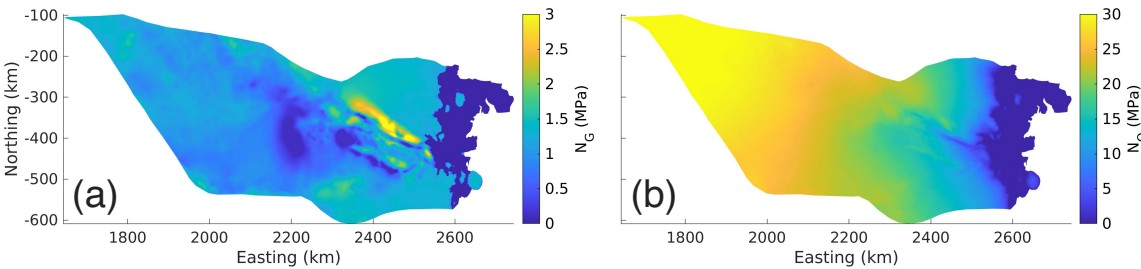

**Figure 3.** Effective pressure inputs (note the colormap scales are not all the same). **(a)** effective pressure (MPa) from GlaDS, $N_G$, capped at 0.4% of ice overburden pressure for the Schoof friction law shown here; **(b)** prescribed effective pressure (MPa), $N_O$.

overburden pressure. A low effective pressure corresponds to a high fraction of flotation which is defined to be the subglacial water pressure divided by the ice overburden pressure. When the subglacial water pressure is equal to the ice overburden pressure flotation is reached and the fraction of flotation is 100 %. Effective pressure in the GlaDS outputs is lowest in the basin feature (Fig. 2d (v)) and the lake-like feature (Fig. 2d (vi)), reaching -0.4 MPa in the former and -0.25 MPa in the latter (Fig. 2b). This translates to a maximum of 101 % fraction of flotation relative to the ice overburden pressure for both sites (Fig. 2c). Other regions of negative effective pressure occur in the troughs of Denman and Scott Glaciers. Low effective pressures, close to zero (>92 % fraction of flotation), persist through the upper regions of the domain and through the subglacial valleys and troughs of both glaciers. Close to the grounding line, effective pressures rise to 90 % fraction of flotation in the regions where the channels of both glaciers exit into the ice shelf cavity. These low and negative effective pressures persist at near steady state and don't cause instability in the GlaDS model.

### 3.2 Ice dynamics and inversion

We first consider the results for the Schoof friction law (Eq. 6). When using $N_G$ (Fig. 3a), the Schoof friction coefficient estimated using inversion is relatively uniform across the basin, with over 50 % of the basin recording basal friction coefficients that range between 2910 and 3580 $\text{kg}^{1/2}\,\text{m}^{-2/3}\,\text{s}^{-5/6}$ (Fig. 4a). Departures from this occur in isolated patches of very low and very high friction – corresponding to the basin-like feature (Fig. 2d (v)) feeding the eastern branch of the Denman channel and the Scott channel – and a region directly upstream of this feature where the ice surface speeds abruptly decrease. The basal friction coefficients are relatively lower in the Denman and Scott troughs, and alternating high and low "stripes" are evident in the region west of the Denman Glacier and south of the Shackleton Ice Shelf (Fig 1b (i)) where ice surface speeds are $<50\,\text{m}\,\text{a}^{-1}$ (Fig. 1a). We note that this region was excluded from the GlaDS model (see discussion in Appendix A1).

The Schoof basal friction coefficient estimated using $N_G$ is smoother compared with that using $N_O$ (Fig. 4a and Fig. 4d; Appendix B). Here, a smoother basal friction coefficient resulted in lower median differences and root mean square error (RMSE) and higher mean differences between the simulated and observed ice surface speeds for the $N_G$ simulation over the $N_O$ simulation (Table 2; Fig. 4c,f). We find a positive correlation ($r^2 = 0.291$) between $N_G$ and the basal friction coefficient (

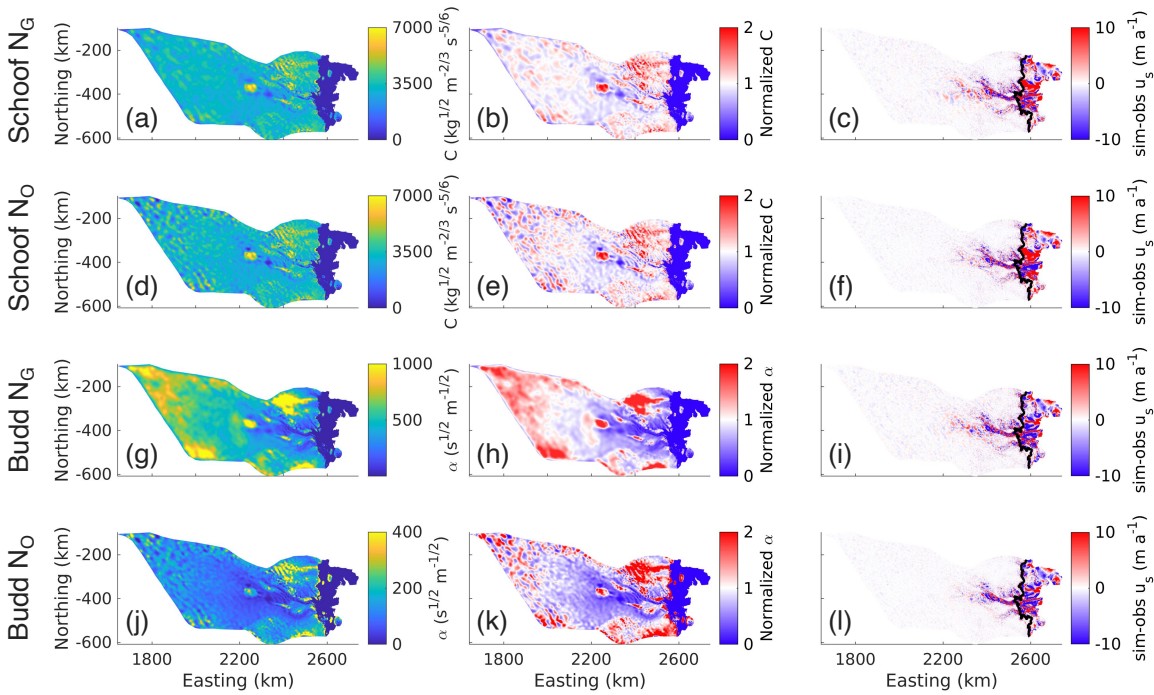

**Figure 4.** Ice dynamics outputs (note the colormap limits are not all the same). **(a)**, **(d)**, **(g)**, **(j)** are basal friction coefficients ($C$ for (a) and (d), $\alpha$ for (g) and (j)), (b), **(e)**, **(h)**, **(k)** are basal friction coefficients normalized to their respective means (normalized $C$ for (b) and (e), normalized $\alpha$ for (h) and (k)), **(c)**, **(f)**, **(i)**, **(l)** are the differences between the simulated and observed ice surface velocities (m a$^{-1}$), sim-obs $u_s$. **(a)**, **(b)**, and **(c)** show outputs from the Schoof friction law with $N_G$; **(d)**, **(e)**, and **(f)** are from the Schoof friction law with $N_O$; **(g)**, **(h)**, and **(i)** are from the Budd friction law with $N_G$; and **(j)**, **(k)**, and **(l)** are from the Budd friction law with $N_O$. In each panel, the black lines are the grounding lines.

| Friction Law | Effective Pressure | difference between simulated and observed surface speeds | | |
| --- | --- | --- | --- | --- |
| | | **median** (m a$^{-1}$) | **mean** (m a$^{-1}$) | **root mean square error** (m a$^{-1}$) |
| Schoof | $N_G$ | 0.163 | 0.391 | 7.31 |
| | $N_O$ | 0.249 | 0.315 | 7.34 |
| Budd | $N_G$ | 0.0855 | -0.0950 | 6.52 |
| | $N_O$ | 0.268 | -0.179 | 6.04 |

**Table 2.** Median, mean, and root mean square error (RMSE) of the differences between the simulated and observed ice surface speeds for the Schoof and the Budd friction laws using $N_G$ and $N_O$.

Fig. 5a) when considering areas where ice surface speeds are $\geq 10$ m a$^{-1}$, in the region where ice is more dynamic closer to the grounding line. Similarly, there is a positive correlation between $N_O$ and the basal friction coefficient ($r^2 = 0.266$, Fig. 5b).

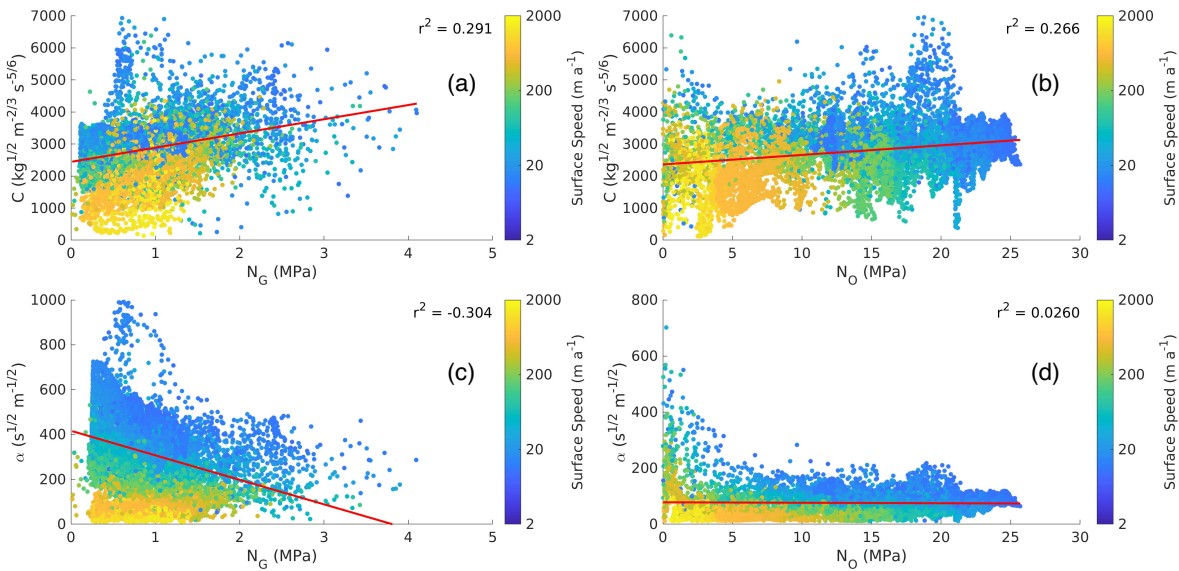

**Figure 5.** Relationship between the effective pressure and basal friction coefficient for: **(a)** Schoof basal friction coefficient $C$ with $N_G$; **(b)** Schoof basal friction coefficient $C$ with $N_O$; **(c)** Budd basal friction coefficient $\alpha$ with $N_G$; and **(d)** Budd basal friction coefficient $\alpha$ with $N_O$. In each panel, points are colored by the natural logarithm of the ice surface speed. The red line is the linear line of best fit, with the correlation coefficient reported ($r^2$).

We next compare the difference between basal friction coefficients with the Schoof and Budd friction laws and using $N_G$.
All basal friction coefficients are relatively lower in the Denman and Scott Glacier troughs and the lake-like and basin features that feed into these troughs, and relatively higher in the region between the lake-like and basin features (Fig. 4b,h). However, the range of the basal friction coefficient across the catchment estimated with the Budd friction law is substantially greater than that with the Schoof friction law, with relatively high values in the interior of the catchment compared with towards the coast. This leads to a greater standard deviation in the Budd friction coefficient compared with that of Schoof when both 210 are normalized to their respective means (0.228 for Schoof and 0.385 for Budd). Unlike the Schoof friction law, the choice of $N_O$ or $N_G$ has a significant impact on the distribution of the basal friction coefficient (Fig. B4) in the Budd friction law. This is because the upglacier portions of the catchment fall into a Weertman sliding regime in the Schoof friction law where $\tau_b << C_{\max} N$ (Fig. A6) and $C^2 \approx \tau_b/u_b^m$. Here, the choice of effective pressure will have minimal effect on the Schoof friction coefficient. In the Budd friction law $\alpha^2 = \tau_b/(N u_b)$ meaning that the effective pressure continues to play an important role 215 in the basal friction coefficient throughout the entire catchment, which propagates the large discrepancy between $N_O$ and $N_G$ to the basal friction coefficients obtained from using these various effective pressures. Despite the large range of the Budd friction coefficient across the catchment, the Budd friction coefficient is generally smoother than the Schoof friction coefficient, which may be a consequence of the choice of the Tikhonov regularisation coefficient used in the inversion procedure. The Budd friction law also predicts lower mean and median differences between the simulated and observed ice surface speeds

than the Schoof friction law (Table 2), for both $N_O$ and $N_G$. Finally, in contrast with the Schoof result, there is a negative correlation between $N_G$ and the basal friction coefficient using the Budd friction law, with basal friction coefficients increasing for decreasing effective pressures ($r^2 = -0.304$; Fig. 5c).

## 4 Discussion

### 4.1 The GlaDS effective pressure

The GlaDS-calculated effective pressure in this region of the Antarctic reflects variability in ice thickness and basal topography, with regions of negative effective pressure concentrated in deep basins, subglacial valleys and troughs. The effective pressure in the remainder of the domain, including the interior of the catchment, is close to zero with minimal spatial variation. A region of high effective pressure is found to the north of the Denman trough. In this area, the bed topography is steep and the ice thin; this causes water to drain rapidly and lower the local water pressure. It is unclear whether this is a realistic representation
of the basal hydrology in this region or whether the low pressures indicate a limitation of the hydrology model – in situ data collection is necessary to investigate this further.

### 4.2 Effective pressure and basal sliding laws

Using the Schoof friction law in the ice dynamics inversions, the resulting basal friction coefficient is smoother when $N_G$ is applied, compared to when $N_O$ is applied, irrespective of the Tikhonov regularization coefficient (Appendix B). When $N_O$ is
used, large gradients in the basal friction coefficient appear that are not aligned with or perpendicular to flow, and naturally increase as the Tikhonov parameter decreases. Previous studies have linked flow-perpendicular gradients in the basal friction coefficient to topography or hydro-mechanical feedbacks (Sergienko et al., 2014); however, that is unlikely to be the case here given that our gradients are not aligned with or perpendicular to flow. Rather, it is likely that these large gradients in the basal friction coefficient are artefacts of the inversion procedure that arise only when $N_O$ is used.
Our parameterization for $N_O$ is a function of the ice overburden pressure and bed elevation, $N_O = \rho_i g H + \rho_w g B$, which has been commonly used to define the effective pressure in ice sheet modeling (Åkesson et al., 2022; Yu et al., 2018; Åkesson et al., 2021). This parameterization is not the actual subglacial hydrology effective pressure, defined as $N = p_i - p_w$, where $p_i$ and $p_w$ are the pressures of ice and water respectively. Indeed, if at the overburden pressure, the effective pressure $N$ should be zero everywhere. However, the definition for $N_O$ here is in fact the overburden hydraulic potential, not an effective pressure,
and will produce high effective pressures (i.e. low basal water pressures) in regions of thick ice grounded above sea level, and low effective pressure in regions of thick ice grounded below sea level (Fig. 3b), yielding high friction in interior regions of the Denman catchment. The water pressure in this parameterization has a dependence on bed elevation, resulting in areas of negative water pressure anywhere the bed is above sea level, which is not physical. Previous studies have investigated alternative parameterizations for the effective pressure due to the computational cost or numerical instabilities associated with coupling of
an ice-sheet model to a complex subglacial hydrology model. Full coupling between these models is a recent development in

the field (Cook et al., 2022). For example, Brondex et al. (2017) use $N_B = \rho_i gH + \rho_w gB$ for areas grounded below sea level and $N_B = \rho_i gH$ for areas grounded above sea level, which leads to zero water pressure above sea level. Bueler and van Pelt (2015) used a parameterization that incorporates the effects of till at the base. Our results suggest that ice sheet models would be more accurate with improved representations of the effective pressure in friction laws (e.g. Cook et al., 2022), highlighting the need for further investigation into appropriate parameterizations for the effective pressure when a subglacial hydrology model output is not directly available. To this end, we suggest an alternative parameterization for the effective pressure in section 4.3 below.

Our finding of a positive correlation between $N_G$ and the basal friction coefficient for the Schoof friction law is unsurprising: effective pressures are lower where there is substantial basal lubrication, which generally leads to a lower basal friction coefficient and faster surface flow. This is evidence for subglacial hydrology being a key control on the surface speeds in regions of lower effective pressures for the Denman Glacier. However, in the regions of fastest flow at Denman Glacier, particularly towards the grounding line, the model also suggests the presence of substantial channels, which have previously been associated with increasing effective pressures and slowing ice flow (Nienow et al., 2017). The simultaneous presence of channels, fast ice flow, and low effective pressures has also been observed at Aurora Subglacial Basin (Dow et al., 2020) and in the Weddell Sea region (Dow et al., 2022). This is due to the channels being near steady-state which means they operate at effective pressures closer to that of the surrounding distributed system, compared to alpine-like hydrologic systems (Iken and Bindschadler, 1986). In the inland regions of the Denman catchment, distributed system effective pressure is also near zero, yet ice flow speed in this region is low. Here, longitudinal friction plays a key role in restraining inland ice flow, with fast flow near the grounding line requiring a combination of high driving stress and low basal effective pressures.

The relationship between low effective pressures and low basal friction did not hold for the Budd friction law, (Fig. 3a, Fig. 4e) despite a stronger negative correlation between the basal friction coefficient and ice surface speed for the Budd friction law compared to the Schoof friction law (Fig. 5a,c). Our study uses inverse methods to calculate the basal friction coefficient for a given $\tau_b$ and $N_G$. For the Budd friction law, this yields a basal friction coefficient that adjusts to compensate for large variations in basal ice velocities – the basal friction coefficient decreases as basal ice velocities increase and vice versa, and there is a strong inverse relationship between these two fields. However, the requirement for such a strong inverse relationship between basal friction coefficient and basal ice velocity leads to a generally weaker relationship between the basal friction coefficient and the effective pressure, since the effective pressure depends on factors other than the basal ice velocity: the topography, the ice thickness, and water accumulation along with efficiency of basal channels to some extent. For the Schoof friction law there is no requirement for a strong correlation between the basal friction coefficient and basal ice velocities, and we find that spatial variations in the basal friction coefficient more closely mirror those of the effective pressure, resulting in the stronger correlation between these two fields.

The choice of friction law plays an important role in grounding line migration and mass loss, with studies showing that friction laws that incorporate effective pressure yield more realistic representation of ice sheet dynamics, grounding line retreat, and mass loss than those that do not (Åkesson et al., 2021; Brondex et al., 2019). In an idealized flow line model, Brondex et al. (2017) showed that when using a Weertman friction law, which does not consider the strong dependence of $\tau_b$ on the

effective pressure, the grounding line can be stable on a retrograde slope, contrary to results from theoretical analysis (Schoof, 2007), and recommended the use of the Schoof friction law due to its strong physical basis. Regularized Coulomb friction laws have also demonstrated superior performance over the Budd friction law in capturing observed grounding line migration (Joughin et al., 2019) and more realistic estimates of future ice mass loss (Brondex et al., 2019). These results are largely a consequence of the cavitation effects captured in regularized Coulomb friction laws, which prevent unphysical, unbounded basal shear stresses for increasing ice velocities (Schoof, 2005). By contrast, the basal shear stress is unconstrained for finite values of $m$ in other friction laws, including the Weertman or Budd friction laws. Consistent with Brondex et al. (2017, 2019), our results favor the use of a regularized Coulomb friction law combined with effective pressure output from the GlaDS model for more realistic representation of the basal friction coefficient calculated using inversion.

In this work we have used an SSA ice flow model, which fails to capture bed-parallel vertical shear deformations. This may affect the results of our inversions for the basal friction coefficient in areas of non-negligible vertical shear, such as at the onset of fast-flowing ice streams of the Denman and Scott troughs. However, the use of the Glen flow relation may also impact the capacity of even higher-order models to accurately capture bed-parallel vertical shear deformations. For example, McCormack et al. (2022) showed that even when the BP approximation to the full Stokes equations is used, the unenhanced Glen flow relation fails to capture the vertical shear profile expected in regions where ice is frozen to the bed of the glacier. In our simulations that use the SSA approximation and the Glen flow relation, it is possible that sliding is overestimated, and the basal friction coefficient underestimated, where vertical shear is an important deformation process. However, this is a common issue to all of our model runs and is therefore unlikely to alter the main conclusions of this work that compare how the form of the effective pressure impacts the basal friction coefficient.

Initializing our temperature field with surface temperatures from RACMOv2.3 (van Wessem et al., 2018) could have an impact on the rigidity and basal friction coefficient inversions we performed in this work. Zhao et al. (2018) showed that initializing a model with a colder temperature field resulted in a decrease of the basal friction coefficient computed from inversion, due to stiffer ice. It is possible that if we used a thermal model and computed depth averaged temperatures to use in our model initialization – effectively increasing the initialization temperatures – that we could similarly see an increase in the basal friction coefficients. Like with the SSA approximation, these are issues that would be common in all of our model runs and would be unlikely to alter the main conclusions we came to regarding the effect of the form of the effective pressure on the computation of the basal friction coefficient.

The GlaDS model used here has been demonstrated to accurately represent observed properties of the Antarctic subglacial hydrologic system (Dow et al., 2018, 2020, 2022). Nevertheless, it has not yet been coupled with an ice sheet model for prognostic simulations of Antarctic ice dynamics. Given the key role that effective pressure plays in basal friction laws, it is clear that future implementations of ice dynamics models should explicitly take relevant subglacial hydrology processes into account, such as spatially and temporally-variable basal water pressure, if they are to accurately predict climate-driven changes in ice dynamics. A recent study by Kazmierczak et al. (2022) coupled simplified representations of subglacial hydrology to ice dynamics using the Budd friction law. This study by Kazmierczak et al. (2022) examined the impact of different representations of the effective pressure – approximated in turn by height above buoyancy (van der Veen, 1987; Huybrechts, 1990; Winkelmann

et al., 2011; Martin et al., 2011), reduced by the subglacial water pressure (modified from Bueler and Brown (2009)) or sliding related to water flux (Goeller et al., 2013) from a simple subglacial water routing model (Le Brocq et al., 2009), and an effective pressure in a till (Bueler and van Pelt, 2015) – on ice mass lost over Antarctica over the 21st Century. A key finding is that when the effective pressure is low in the grounding zone, the ice sheet is more sensitive to increases in climate forcing, ultimately increasing mass loss and resulting in sea level rise. The accuracy of these simplified subglacial hydrology models in representing the full physics of the ice-bed interface still needs to be tested. However, we found that the GlaDS model predicted low effective pressures in the grounding zone of the Denman-Scott catchment, which, following Kazmierczak et al. (2022), suggests this region might be sensitive to enhanced mass loss in the future. The next step is to couple the GlaDS model for a more complex representation of the basal hydrology with an ice dynamics model for application to Antarctic glaciers – an approach that has been applied to Greenland glaciers (e.g. Cook et al., 2022).

### 4.3 Empirical Parameterization

We suggest here an alternative parameterization of effective pressure, $N_E$, which is empirical in nature and based on spatial patterns in the GlaDS effective pressure output of Denman Glacier. Given the high correlation of water pressure to ice over-burden pressure ($r^2 = 0.998$ over the Denman domain) and the relatively constant effective pressure in the interior of Denman, we suggest a form of the water pressure proportional to the ice overburden pressure multiplied by a term $(r_l + (1 - r_l)g_x(H))$ where $g_x(H) = H^x/(\tilde{H}^x + H^x)$ is a saturation term such that the water pressure reaches a maximum fraction of flotation in areas of high ice thickness. Here, $\tilde{H}$ is defined in Eq. (8) and $x$ is defined in Eq. (9). The proposed parameterization is as follows:

$$N_E = \rho_i g H (1 - r_l) \frac{\tilde{H}^x}{\tilde{H}^x + H^x}, \tag{7}$$

$$\tilde{H} = \left( \frac{1 - \gamma}{\gamma - r_l} \right)^{1/x} H_t, \tag{8}$$

$$x = \frac{\ln\left(\frac{1-r_l}{\epsilon} - 1\right) + \ln(\gamma - r_l) - \ln(1 - \gamma)}{\ln(H_t) - \ln(H_s)}. \tag{9}$$

Here, $\rho_i$ is the density of ice, $g$ is the gravitational acceleration, $H$ is ice thickness, and the effective pressure is given by Eq. (7). $\gamma$ is a constant representing typical effective pressure as a fraction of ice overburden in areas of high ice thickness; $r_l$ is the water pressure as a fraction of ice overburden pressure as the ice thickness goes to zero; $H_t$ is a typical large ice thickness; $H_s$ is a typical small ice thickness; and $\epsilon$ is a small constant taken so that in areas of low ice thickness ($H_s$) the water pressure is at a typically low value $(r_l + \epsilon)\rho_i g H$. GlaDS output effective pressure and ice thickness data for the Denman Glacier was used to find values for $r_l$, $\gamma$, $H_s$, and $H_t$ used in $N_E$ ($\epsilon$ was taken to equal 0.05). This resulted in $r_l = 0.7$, $\gamma = 0.96$, $H_s = 500 \, \text{m}$, and $H_t = 2800 \, \text{m}$. The empirical nature of this parameterization means that these constants are not "optimized" and use of $N_E$ on future glaciers may require an alteration of these constants, specifically $H_s$ and $H_t$ which can be determined using ice thickness data. More details on the parameterization and its derivation are provided in Appendix C.

The proposed empirical effective pressure ($N_E$), is compared with the GlaDS output effective pressure ($N_G$), the typically prescribed effective pressure $N_O$, and the Brondex et al. (2017) prescribed effective pressure ($N_B$, Section 4.2) – all as a

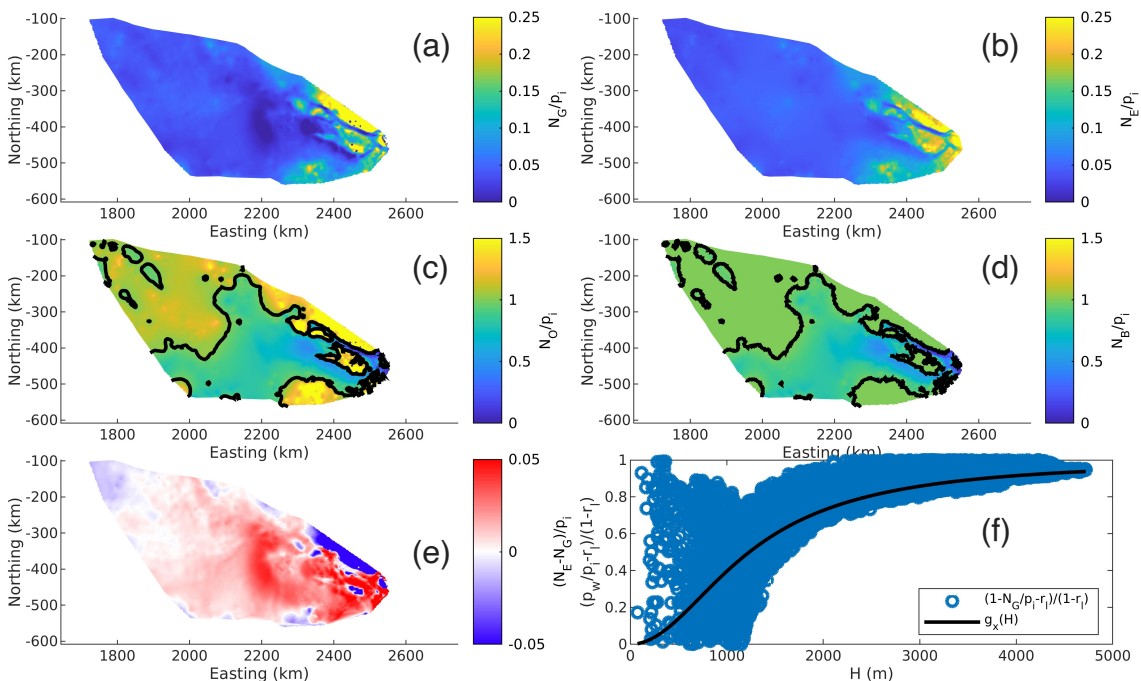

**Figure 6.** Effective pressures as a fraction of ice overburden pressure for: **(a)** GlaDS output effective pressure $N_G/p_i$, **(b)** empirical parameterization of effective pressure $N_E/p_i$, **(c)** typically prescribed effective pressure $N_O/p_i$, and **(d)** Brondex et al. (2017) prescribed effective pressure $N_B/p_i$. **(e)** The difference between the empirical parameterization of effective pressure and the GlaDS output effective pressure as a fraction of ice overburden pressure $((N_E - N_G)/p_i)$. **(f)** The saturation curve $(g_x(H))$ and the physically equivalent scatter for $N_G$ $((1 - N_G/p_i - r_l)/(1 - r_l))$. Black lines in **(c)** and **(d)** represent the $N/p_i = 1$ contour.

fraction of ice overburden pressure (Fig. 6). $N_E$ matches $N_G$ more closely than both $N_O$ and $N_B$ (note the different color bar scalings in Fig. 6a,b compared with Fig. 6c,d). This is further illustrated in Fig. 6e, which shows the difference between $N_E$
and $N_G$ as a fraction of ice overburden pressure: 72.9 % of the domain has a difference of <0.02, and 95.6 % of the domain has a difference of <0.05. Fig. 6f shows the saturation term $(g_x(H))$ as well as the physically equivalent scatter from the GlaDS output data, $(1 - N_G/p_i - r_l)/(1 - r_l)$. The form of the scatter is qualitatively similar to the saturation term in areas of moderate to high ice thickness. The mean percent difference excluding the area of the domain where $N_G \leq 0$, given by $|N_E/N_G - 1|$, is 118 %, though 45.4 % of this domain falls below 20 % and 77.0 % of the domain falls below 50 % so the data is skewed from
the areas of low effective pressure. Standing alone, this does not sound impressive and it is a result of the variability of the scatter seen in Fig. 6f. This is still a substantial improvement over $N_O$ which has a mean percent difference of 3688 % where 57.7 % of the domain falls between 2000 % and 3000 % difference.

Though this parameterization is not derived from physical principles and it lacks complete hydrological connectivity to the ocean, it produces physically realizable effective pressures $0 < N < \rho_i g H(1 - r_l)$, which match closely to GlaDS output

effective pressures over Denman Glacier. The success of $N_E$ over $N_O$ on Denman Glacier combined with its improved physical implications over $N_O$ warrants further investigation into its validity and potential use in future modeling studies, particularly on Antarctic glaciers.

## 5   Conclusions

The Denman-Scott catchment in East Antarctica is a region that is undergoing active glacial retreat and is vulnerable to ongoing mass loss as the climate warms. We have investigated the coupled interactions between the subglacial hydrological system and the ice sheet through the basal friction coefficient – an important tuning parameter used in basal friction laws – and its dependence on the form of the effective pressure. We find that when the Schoof friction law is used, there is a smoother basal friction coefficient and slightly improved match between the simulated and observed ice surface velocities is found when $N_G$ is used, compared to when $N_O$ is used. Of the two friction laws tested here – the Schoof friction law and the Budd friction law – only the Schoof friction law captures the predicted relationship between the subglacial hydrological and the ice sheet dynamics. That is, using the Schoof friction law, regions of lower effective pressures (both $N_G$ and $N_O$) tend to also have lower basal friction coefficients – evidence for the controlling role of the hydrological system.

In their application of the GlaDS model to the Aurora Subglacial Basin, Dow et al. (2020) showed the utility of geophysical observations in validating and constraining subglacial hydrology model parameterizations. By contrast with the Aurora Subglacial Basin, the Denman-Scott catchment is a relatively sparsely observed system, with few geophysical observations available to constrain our model simulations. Similarly to previous studies (Dow et al., 2020; Schroeder et al., 2013; Hager et al., 2022), we recommend the collection of: high-resolution radar surveys, with a particular focus on the regions at the boundary of the distributed and channelized subglacial hydrologic systems and over the lake-like features upstream of the Denman and Scott channels; and direct measurement of subglacial hydrological flux across the grounding line to constrain simulated subglacial meltwater volumes. Such observations are vital in understanding the role of the subglacial hydrological system and its coupled feedbacks with the ice sheet.

We proposed a new parameterization for the effective pressure based on empirical data of the Denman-Scott catchment and the GlaDS results, that more closely matches simulated effective pressures than suggestions from previous literature, and which shows promise in capturing key features of the subglacial system in the absence of a complete subglacial hydrology model output. However, given the empirical nature of this parameterization, our results highlight the importance of simulating coupled interactions between the subglacial hydrology and ice sheet systems to accurately represent ice sheet flow, and future work should focus on coupling of these two systems in transient simulations of the Antarctic Ice Sheet.

## Appendix A: Sensitivity tests

### A1    Grid domains and effective pressure extrapolation

The domains of the ISSM and GlaDS models differ, with the GlaDS domain being a subset of the ISSM domain due to the differing subglacial hydrological and ice catchments and limits to the GlaDS domain size requiring restriction to the primary hydrological outlets of Denman and Scott Glaciers. This means that there are regions within the ISSM domain for which the GlaDS effective pressure does not exist. We test the impact of two different methods to extrapolate the effective pressure in regions outside the GlaDS domain.

The first method relates the GlaDS effective pressure to the bed topography and ice surface speed. We partition the domain into three regions based on ice surface speed, with the first encompassing areas where $v<500\,\mathrm{m\,a^{-1}}$, the second where $500{\leq}v<1000\,\mathrm{m\,a^{-1}}$, and the third where $v{\geq}1000\,\mathrm{m\,a^{-1}}$. We then calculate a linear regression between the bed topography and the GlaDS effective pressures in each region, and linearly extrapolate the effective pressure onto the ISSM domain based on this relationship. The regression coefficients are shown in Table A1.

The second method uses $N_O$ in the areas of the ice sheet model domain where GlaDS was not run. The resulting effective pressure is shown in Fig. A1. The basal friction coefficient found using method two is generally more smooth than the basal friction coefficient found using method one, with the exception of transition areas between the GlaDS and extrapolated effective pressures to the far west and south of the GlaDS domain. The match between the simulated and observed ice surface speeds are similar for the two extrapolations, though method two produces more variation in the region upstream of the grounding line

than method one.

| Regression coefficient | $v<500\,\mathrm{m\,a^{-1}}$ | $500{\leq}v<1000\,\mathrm{m\,a^{-1}}$ | $v{\geq}1000\,\mathrm{m\,a^{-1}}$ |
|---|---|---|---|
| $\alpha$ | 286 | -248 | 136 |
| $\beta$ | $1.28\times10^{6}$ | $8.03\times10^{5}$ | $1.11\times10^{6}$ |

**Table A1.** Linear regression coefficients $\alpha$ and $\beta$ for the relationship $N=\alpha B+\beta$, where $N$ is the GlaDS effective pressure and $B$ is the bed elevation.

In general, there is no reason that the hydrology and ice dynamics domains should match exactly, and where this is the case, we would always expect regions where we will need to extrapolate boundary conditions between the models. The fact that we have had to extrapolate effective pressure is not in itself a problem, and our results show the importance of taking into consideration the method of extrapolation. There may be other methods for extrapolating the boundary conditions that we

did not consider, including parameterizing the effective pressure using different physical assumptions (e.g. Bueler and Brown, 2009; Bueler and van Pelt, 2015; Van der Wal et al., 2013; Huybrechts, 1990; Kazmierczak et al., 2022; van der Veen, 1987; Winkelmann et al., 2011), and future studies should establish best practice in this regard.

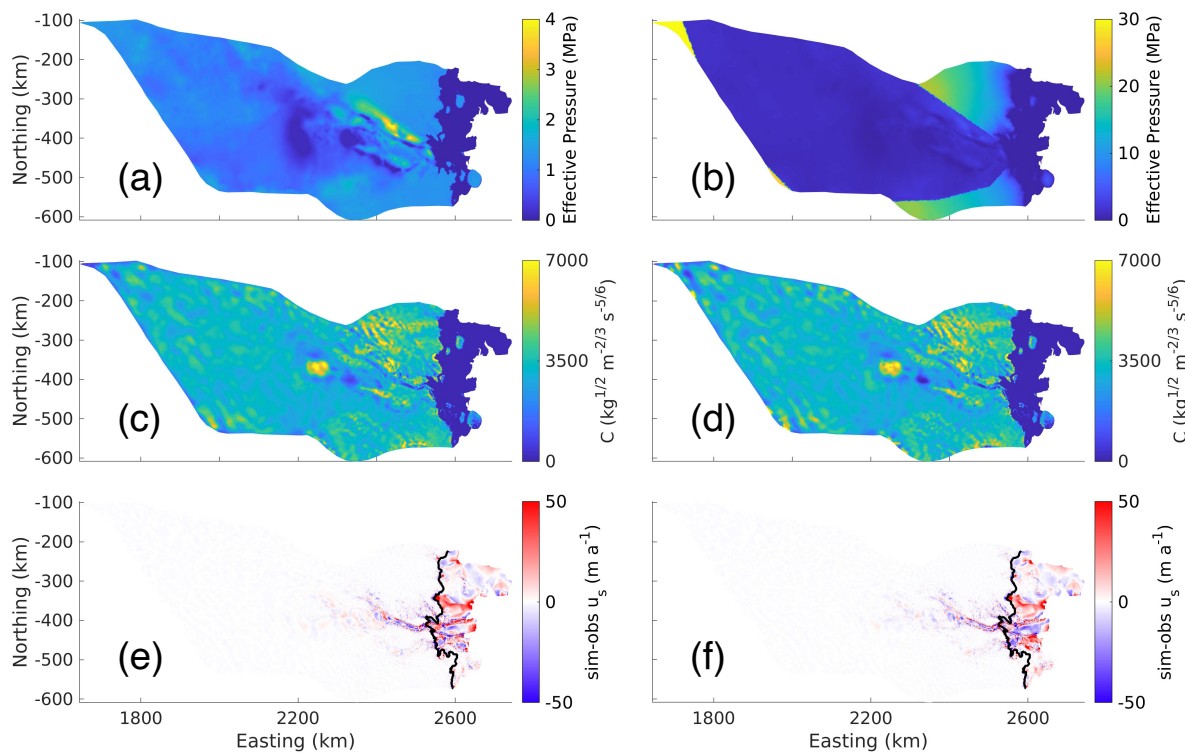

**Figure A1. (a)**-**(b)** Effective pressure (MPa); **(c)**-**(d)** basal friction coefficients estimated using inversion of the Schoof friction law ($C$, $\text{kg}^{1/2}\,\text{m}^{-2/3}\,\text{s}^{-5/6}$); and **(e)**-**(f)** difference between the simulated and observed ice surface speeds (sim-obs $u_s$, $\text{m}\,\text{a}^{-1}$). The left column **(a)**, **(c)**, **(e)** shows results for extrapolation method 1 and the right column **(b)**, **(d)**, **(f)** shows results for method 2. In each panel, the black lines are the grounding lines.

## A2 Mesh sensitivity

Depending on the degree of regularisation in the inverse method, small-scale features may be present in the basal friction
coefficient. We perform a sensitivity analysis to examine the extent to which such features may arise due to the location of
the nodes and vertices in the mesh. We generate five different meshes by adding a random perturbation in the surface velocity
field of up to 20 % of the MEaSUREs velocities, and then perform non-uniform mesh refinement on the velocity magnitude, as
follows: the typical edge length of the elements is 3000 m; for regions where the ice surface speed is $\geq$500 $\text{m}\,\text{a}^{-1}$, maximum
edge lengths are 500 m; and for regions where the ice surface speed is between 100 and 500 $\text{m}\,\text{a}^{-1}$, the maximum edge length
is 2000 m. Elements close to the grounding line have a maximum edge length of 1000 m.

Using each of these five meshes, we calculate the Budd and Schoof basal friction coefficients. The probability distributions
of the basal friction coefficients normalized to 3500 $\text{kg}^{1/2}\,\text{m}^{-2/3}\,\text{s}^{-5/6}$ (half of the maximum allowed value of $C$) are shown in
Fig. A2. The mean and standard deviation of the differences between both the Schoof and the Budd basal friction coefficients

for the perturbed mesh simulations and that of the original mesh are smaller in magnitude than the mean and standard deviation

of the difference between the basal friction coefficients using $N_G$ and $N_O$. Hence, the details of the mesh have less of an impact on the basal friction coefficients than the form of the effective pressure. The accumulated probability at the lower and upper bounds in Fig. A2 correspond to the regions of low/negative effective pressure near subglacial lakes and troughs and the area of negative effective pressure upstream of the subglacial lake feeding the Denman and Scott troughs (Fig 2d (v)) respectively.

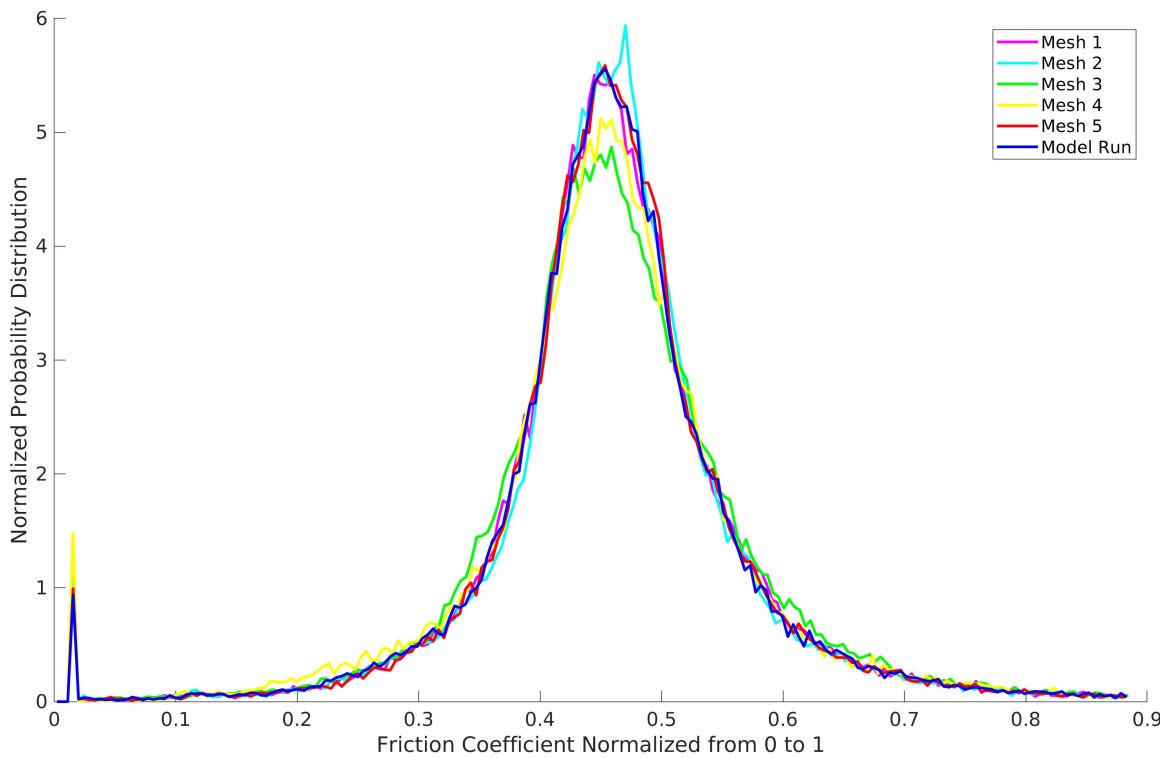

**Figure A2.** Normalized (to $3500\,\mathrm{kg}^{1/2}\,\mathrm{m}^{-2/3}\,\mathrm{s}^{-5/6}$, half of the maximum allowed value of $C$) probability distribution functions for the Schoof basal friction coefficient, $C$, for the computational mesh and five perturbed meshes. Vertical lines indicate the mean of each distribution.

## A3 Effective pressure cap

For regions of very low or negative effective pressure, the basal shear stresses are also very low or almost vanish. In these regions, the inverse method compensates by increasing the basal friction coefficient upstream and around the region of the anomalously low effective pressure, leading to an underestimate of ice surface speeds there compared with the observations. The ice surface speeds are also generally overestimated in the region of vanishing shear stresses.

To account for this, we adjust the effective pressure in regions of low or negative effective pressure, capping the minimum as 440 a percentage of the ice overburden pressure. We test caps of 0.2 %, 0.4 %, 1 %, 2 % and 4 %. For a cap of 0.4 %, approximately 2 % of the domain has an effective pressure that is linearly proportional to the ice overburden pressure, that is, 98 % of the effective pressure in the ISSM simulation is derived directly from the GlaDS simulated effective pressure. Increasing the cap to 1 % – which is used in the Budd runs – decreases the area over which the GlaDS effective pressure is used to 96 % and increasing the the the cap to 4 % decreases the area to 48 %. Hence, we use a cap of 0.4 % for our Schoof simulations and 1 % for 445 our Budd simulations.

## A4 Nonlinear Budd friction law

Here, we compare the impact of using a nonlinear exponent $m = 1/3$ in the Budd friction law (Eq. 5; as per Brondex et al., 2017, 2019) to our results with the linear exponent $m = 1$ (as per Åkesson et al., 2021; Åkesson et al., 2022; Yu et al., 2018; Choi et al., 2021; Baldacchino et al., 2022). We follow the same setup outlined in the main text and in Appendix B, capping the effective pressure at 1 % of ice overburden pressure. The L-curve analysis suggests a Tikhonov regularization value of 0.1 is optimal.

The $m = 1/3$ case had much smaller variance in the normalized basal friction coefficient compared to the $m = 1$ case (Fig. A3). That is, using $N_G$, the variance of the normalized basal friction coefficient was 0.231 for $m = 1/3$ and 0.385 for $m = 1$; using $N_O$, the variance was 0.508 for $m = 1/3$ and 0.628 for $m = 1$.

Despite the smaller normalized variance in the basal friction coefficients for the $m = 1/3$ case, there was a stronger negative correlation between the basal friction coefficient and the effective pressure in areas where ice surface speeds are greater than $10\,\mathrm{m\,a}^{-1}$ (Fig. A4). Using $N_G$, the correlation was $-0.466$ for $m = 1/3$ and $-0.304$ for $m = 1$; using $N_O$, the correlation was $-0.592$ for $m = 1/3$ and $-0.0260$ for $m = 1$ (Fig. A4). This strong negative correlation for the $m = 1/3$ case indicates that the basal friction coefficient counteracts the effects of the effective pressure, with high values in regions of low effective pressure and low values in regions of high effective pressure. This suggests that the dependency of the basal shear stress on the effective pressure is too strong when $m = 1/3$. This behaviour is particularly clear in the large area of high basal friction coefficient centered around 2200 km easting and -400 km northing in Fig. A3c, which corresponds to an area of low effective pressure (Fig. 2b). Here, the use of the effective pressure cap limits runaway values of the basal friction coefficient; increasing the cap to reduce this area of high basal friction coefficient would mean using less of the GlaDS data. For the $N_O$ run, the large increase in effective pressure upstream resulted in a strong decrease in the basal friction coefficient upstream (Fig. A3d).

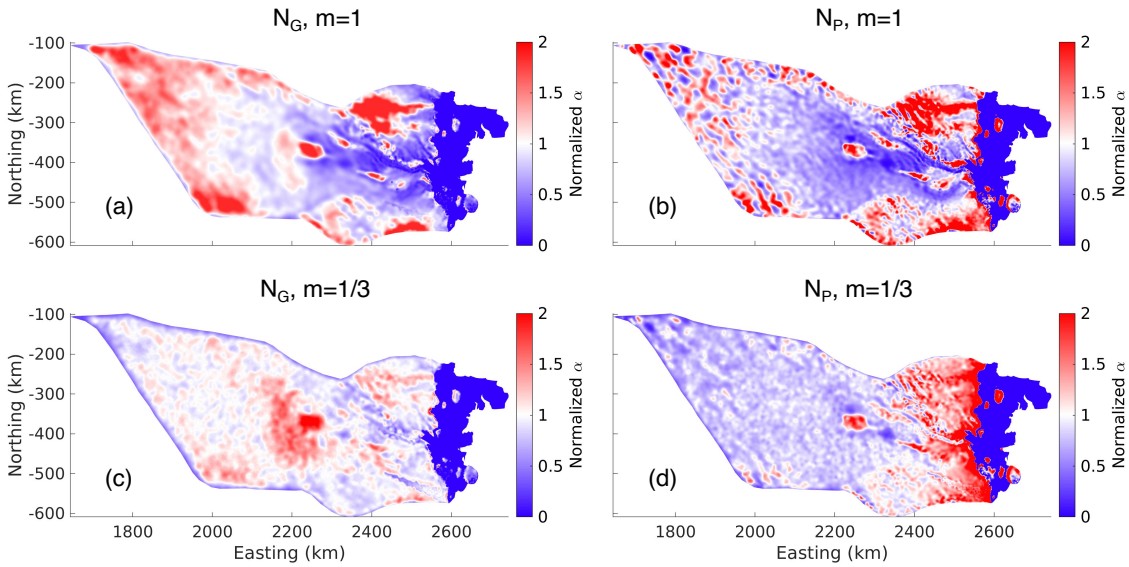

**Figure A3.** Budd basal friction coefficients ($\alpha$) normalized to the mean. **(a)** GlaDS effective pressure ($N_G$) with $m = 1/3$; **(b)** typically prescribed effective pressure ($N_O$) with $m = 1/3$; **(c)** GlaDS effective pressure ($N_G$) with $m = 1$; **(d)** typically prescribed effective pressure ($N_O$) with $m = 1$.

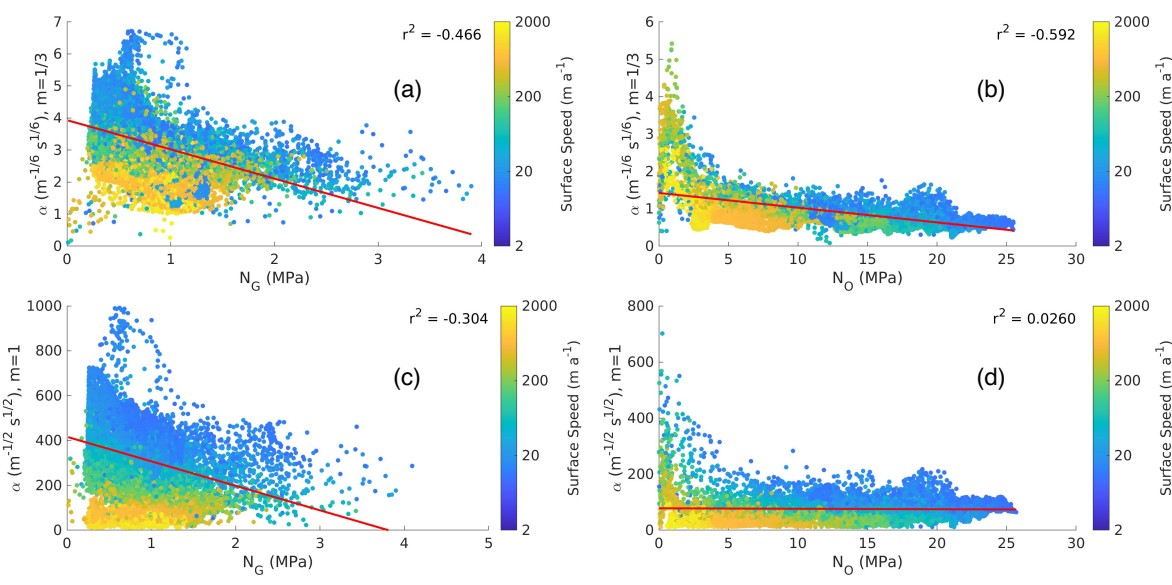

**Figure A4.** Relationship between the Budd basal friction coefficient $\alpha$ and effective pressure for **(a)** The GlaDS effective pressure ($N_G$) with $m = 1$; **(b)** typically prescibed effective pressure ($N_O$) with $m = 1$; **(c)** GlaDS effective pressure ($N_G$) with $m = 1/3$; **(d)** typically prescribed effective pressure ($N_O$) with $m = 1/3$.

## A5 Iken's bound sensitivity

Iken's bound, $C_{\max}$, mathematically describes the idea that the bed can support a maximum stress (Iken, 1981; Schoof, 2005; Gagliardini et al., 2007). In the Schoof friction law (Eq. 6), $\tau_b$ cannot exceed $C_{\max}N$, where $C_{\max}$ represents rheological properties of the till (Brondex et al., 2019) and ranges between 0.17 and 0.84 (Cuffey and Paterson, 2010). To determine an appropriate value of $C_{\max}$ we test the effect of using $C_{\max} = 0.5, 0.6, 0.7$, and 0.8 on the Schoof basal friction coefficient using $N_G$ and $N_O$.

We arrive at the same qualitative conclusions as in the main text for all four values of $C_{\max}$. In all cases, the variance of the normalized basal friction coefficient is smaller for $N_G$ than for $N_O$, and it is smaller than the variance of the normalized Budd basal friction coefficient (Table A2). Although the results for $C_{\max} = 0.5, 0.6$, and 0.7 are similar, there is a comparatively large decrease in the mean of the basal friction coefficient and increase in the variance of the normalized basal friction coefficient between $C_{\max} = 0.7$ and $C_{\max} = 0.8$ for both $N_G$ and $N_O$. For lower values of $C_{\max}$, a region of higher basal friction coefficient centered around 2200 km easting and -400 km northing (Fig. A5a) develops in the $N_G$ simulations. Solving Eq. (6) for $C$ yields

$$C = \frac{\left(\frac{\tau_b}{|u_b|^{m-1}u_b}\right)^{m/2}}{\left(1 - \left(\frac{\tau_b}{C_{\max}N}\right)^{1/m}\right)^{m/2}},\tag{A1}$$

where we see that in the limit where $\tau_b \to C_{\max}N$ then $C \to \infty$. The region of high basal friction coefficient for lower $C_{\max}$ also has low effective pressure (Fig. 2a), resulting from $\tau_b$ approaching $C_{\max}N$ for larger values of $N$ and the basal friction coefficient compensating this. To prevent potentially infinitely increasing values of the basal friction coefficient it is possible to increase the cap on the effective pressure, but again, this would reduce the area over which the GlaDS data is used and the effect of using modeled hydrology will be less impactful.

The Schoof friction law is a regularized Coulomb friction law which tends towards a Weertman sliding regime when $\tau_b << C_{\max}N$ and a Coulomb sliding regime when $\tau_b >> C_{\max}N$. Hence, the value of $C_{\max}$ will have an effect on what physical processes are being represented in the Schoof friction law. Figure A6 shows $\tau_d/(C_{\max}N)$ for various values of $C_{\max}$, where values close to zero correspond to a Weertman sliding regime and values close to one correspond to a Coulomb sliding regime. The choice of $C_{\max}$ appears to have little effect on where each of the Weertman and Coulomb sliding regimes occur, with distinct locations between the two for all values of $C_{\max}$. This suggests that the choice of $C_{\max}$ will not have a significant impact on which physical processes are being represented throughout the domain and does not justify the use of one value of $C_{\max}$ over another.

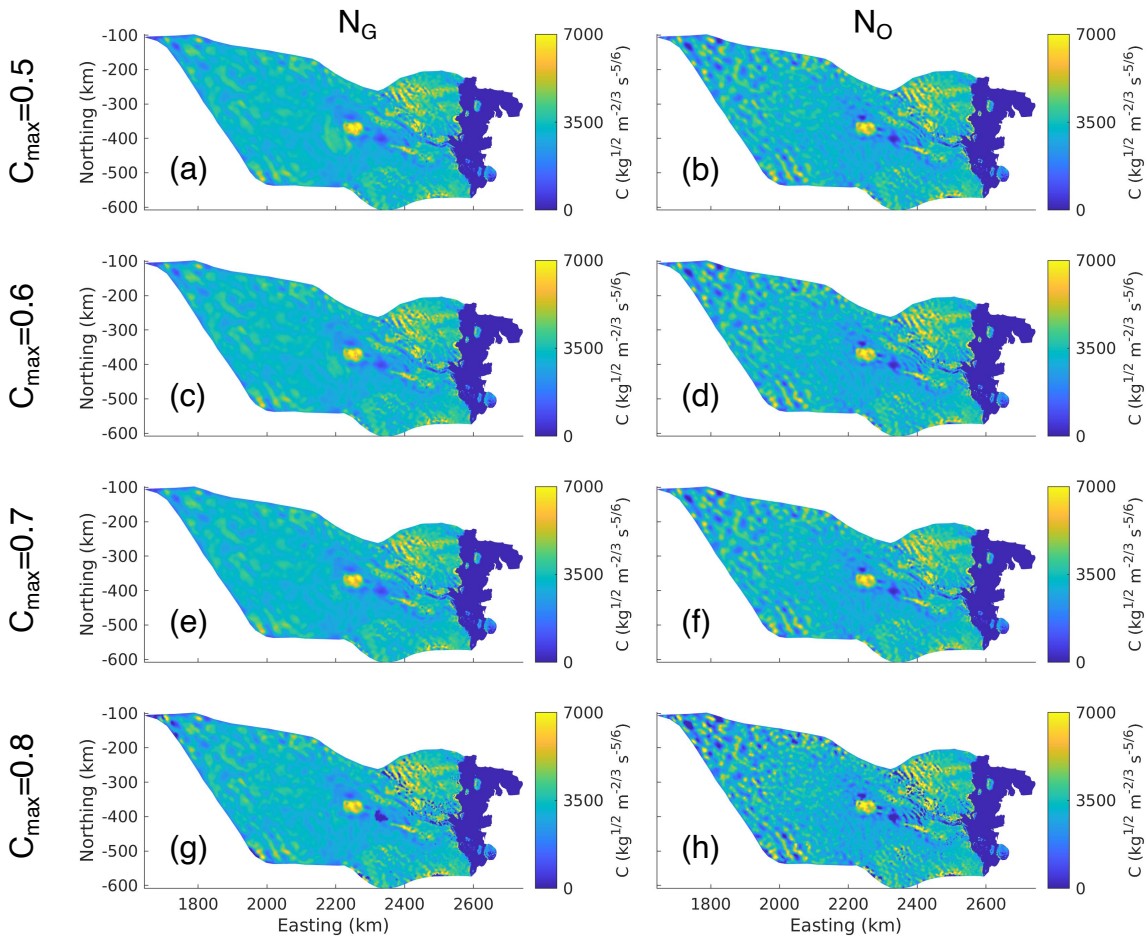

**Figure A5.** Schoof basal friction coefficient, ($C$, kg$^{1/2}$ m$^{-2/3}$ s$^{-5/6}$) for **(a)** $N_G$ and $C_{\mathrm{max}} = 0.5$; **(b)** $N_O$ and $C_{\mathrm{max}} = 0.5$; **(c)** $N_G$ and $C_{\mathrm{max}} = 0.6$; **(d)** $N_O$ and $C_{\mathrm{max}} = 0.6$; **(e)** $N_G$ and $C_{\mathrm{max}} = 0.7$; **(f)** $N_O$ and $C_{\mathrm{max}} = 0.7$; **(g)** $N_G$ and $C_{\mathrm{max}} = 0.8$; **(h)** $N_O$ and $C_{\mathrm{max}} = 0.8$.

| $C_{\mathrm{max}}$ | $\overline{C}$ (kg$^{1/2}$ m$^{-2/3}$ s$^{-5/6}$), $N_G$ | Variance of $C/\overline{C}$, $N_G$ | $\overline{C}$ (kg$^{1/2}$ m$^{-2/3}$ s$^{-5/6}$), $N_O$ | Variance of $C/\overline{C}$, $N_O$ |
|---|---|---|---|---|
| 0.5 | 3261 | 0.233 | 3215 | 0.280 |
| 0.6 | 3251 | 0.231 | 3210 | 0.283 |
| 0.7 | 3242 | 0.228 | 3218 | 0.273 |
| 0.8 | 3215 | 0.266 | 3154 | 0.337 |

**Table A2.** Schoof basal friction coefficient mean $\overline{C}$ (kg$^{1/2}$ m$^{-2/3}$ s$^{-5/6}$) and variance of normalized Schoof basal friction coefficient ($C/\overline{C}$) for various $C_{\mathrm{max}}$ with $N_G$ and $N_O$.

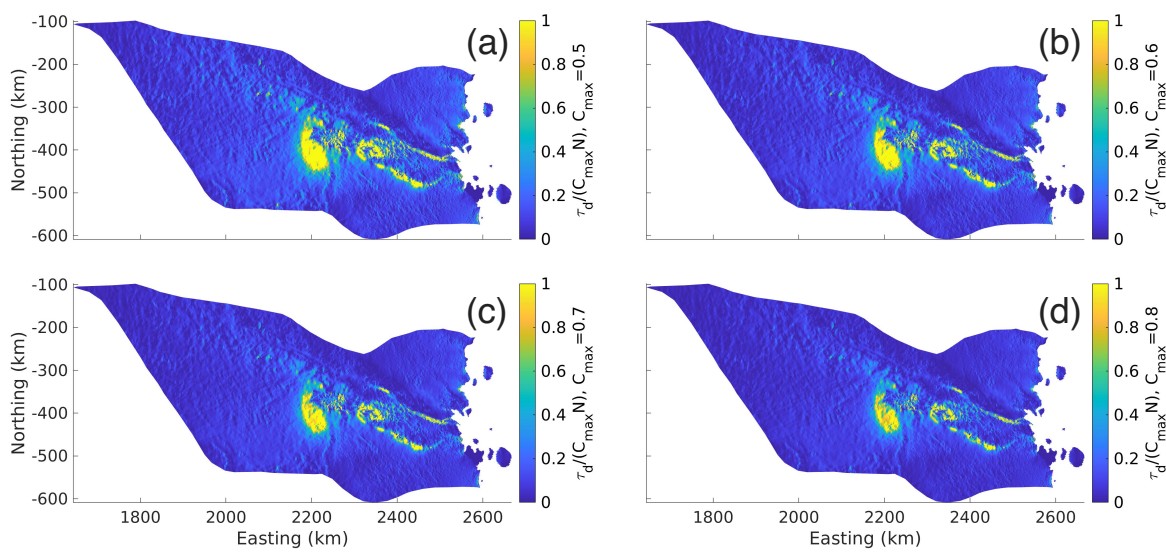

**Figure A6.** Ratio between the driving stress and the maximum stress that the bed can support ($\tau_d/(C_{\max}N)$) for **(a)** $C_{\max} = 0.5$; **(b)** $C_{\max} = 0.6$; **(c)** $C_{\max} = 0.7$; **(d)** $C_{\max} = 0.8$;

### Appendix B: Inversion cost functions

The inversion procedure described in Section 2.2 includes six inversions for the Budd and Schoof basal friction coefficients
and the ice rigidity. Each inversion minimizes a cost function which penalizes absolute ice surface velocity misfit, logarithmic
ice surface velocity misfit, and gradients in the basal friction coefficient/ice rigidity. The cost functions minimized in each
inversion are described below.

$$\mathcal{J}_a(\overrightarrow{u}) = \iint_s \frac{1}{2}\left((u_x - u_x^{\mathrm{obs}})^2 + (u_y - u_y^{\mathrm{obs}})^2\right)dS \tag{B1}$$

$$\mathcal{J}_l(\overrightarrow{u}) = \iint_s \left(\log\left(\frac{||\overrightarrow{u}|| + \varepsilon}{||\overrightarrow{u}^{\mathrm{obs}}|| + \varepsilon}\right)\right)^2 dS \tag{B2}$$

$$\mathcal{J}_t(\overrightarrow{u}) = \iint_s \frac{1}{2}||\overrightarrow{\nabla}k||^2 dS \tag{B3}$$

$$\mathcal{J}(\overrightarrow{u}) = c_a\mathcal{J}_a + c_l\mathcal{J}_l + c_t\mathcal{J}_t \tag{B4}$$

Here, $\mathcal{J}_a$ is the linear velocity misfit cost function, $\mathcal{J}_l$ is the logarithmic velocity misfit cost function, $\mathcal{J}_t$ is the regularization

cost function, $\overrightarrow{u}$ is the modeled ice surface velocity, $\overrightarrow{u}^{\text{obs}}$ is the observed ice surface velocity, $\varepsilon$ is a small number (around

machine precision) acting as the minimum observed velocity in Eq. (B2), $S$ is the two dimensional spatial domain of the

model, and $k$ is taken to be $\alpha$ in a Budd inversion, $C$ in a Schoof inversion, and $\tilde{B}$ (the rigidity) in a rigidity inversion. The

full cost function to be minimized is given in Eq. (B4) where $c_a$, $c_l$, and $c_t$ are the cost function coefficients for $\mathcal{J}_a$, $\mathcal{J}_l$, and $\mathcal{J}_t$

respectively. $c_t$ is referred to as the Tikhonov regularization coefficient.

The linear and logarithmic cost function coefficients ($c_a$, $c_l$) were chosen so that their contributions to the total cost function

would differ by an order one factor in each inversion. An L-curve analysis (Fig. B1; Hansen, 2000) was used during the

inversions for the basal friction coefficients to determine a suitable Tikhonov regularisation coefficient ($c_t$). Ideally $c_t$ is chosen

such that any smaller value of $c_t$ will not have a significant effect on $c_a \mathcal{J}_a + c_l \mathcal{J}_l$. For the ice rigidity inversions, the Tikhonov

regularisation parameter was chosen so that its contribution to the total cost function differed by an order one factor from that

of the linear misfit. The final basal friction coefficients are shown in Fig. B3, their distribution in Fig. B4, and the final rigidities

in Fig. B5. The cost function coefficients for the various model runs are displayed in Table B1.

| Cost Function Coefficient | Value I1 | Value I2 | Value I3 | Value I4- Budd | Value I4- Schoof | Value I5 |
|---|---|---|---|---|---|---|
| Linear ($c_a$) | 8000 | 2000 | 8000 | 2000 | 200 | 8000 |
| Logarithmic ($c_l$) | 60 | 10 | 25 | 10 | 2 | 25 |
| Tikhonov ($c_t$) | $10^{-18}$ | $10^{-6}$ | $10^{-18}$ | $10^{-6}$ | $10^{-9}$ | $10^{-18}$ |

**Table B1.** Cost function coefficients for each inversion. I1- ice rigidity over ice shelf; I2- Budd basal friction coefficient over grounded

domain; I3- ice rigidity over entire domain; I4-Budd- Budd basal friction coefficient over the grounded domain; I4-Schoof- Schoof basal

friction coefficient over the grounded domain; and I5 ice rigidity over the entire domain.

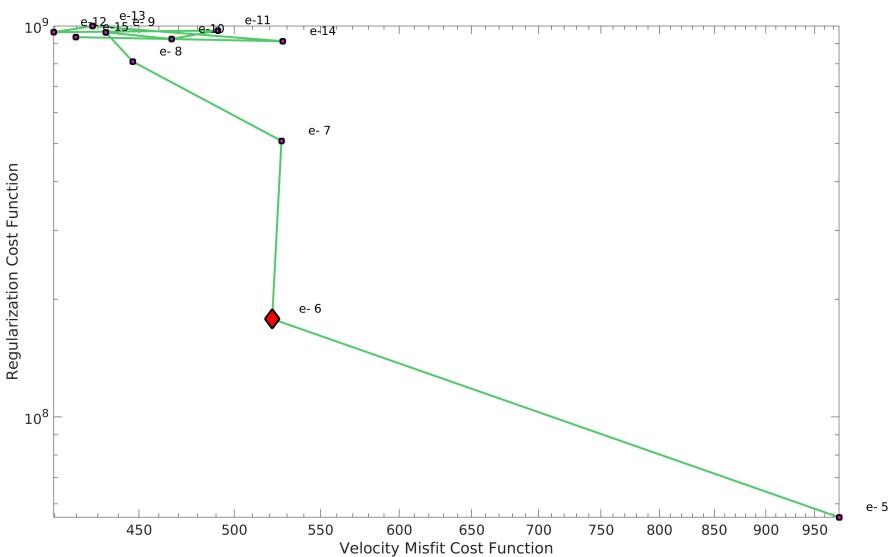

**Figure B1.** L-curve analysis for the Budd basal friction coefficient, $\alpha$, with the GlaDS output effective pressure, $N_G$ (4.5% cap and the typically prescribed effective pressure, $N_O$, outside the GlaDS domain). The red diamond is the chosen value of the Tikhonov regularization coefficient, $c_t$.

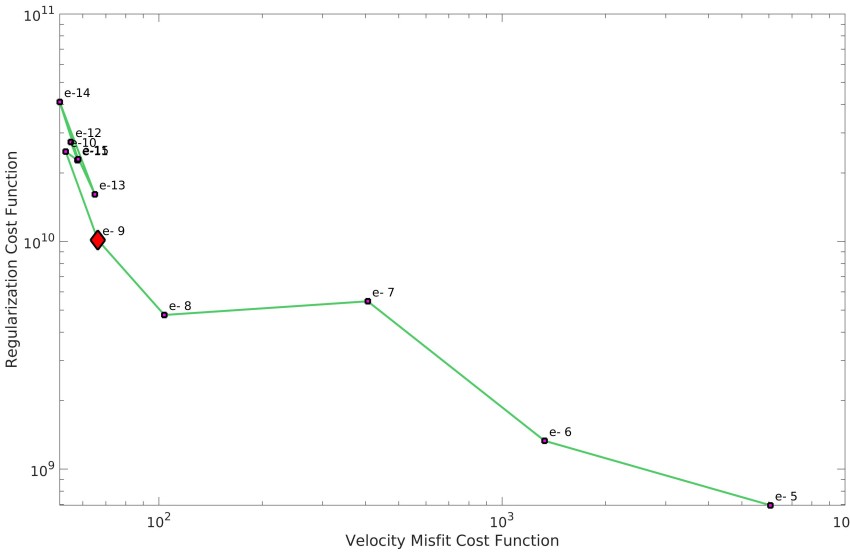

**Figure B2.** L-curve analysis for the Schoof basal friction coefficient, $C$, with the GlaDS output effective pressure $N_G$ (4.5% cap and the typically prescribed effective pressure, $N_O$, outside the GlaDS domain). The red diamond is the chosen value of the Tikhonov regularization coefficient, $c_t$.

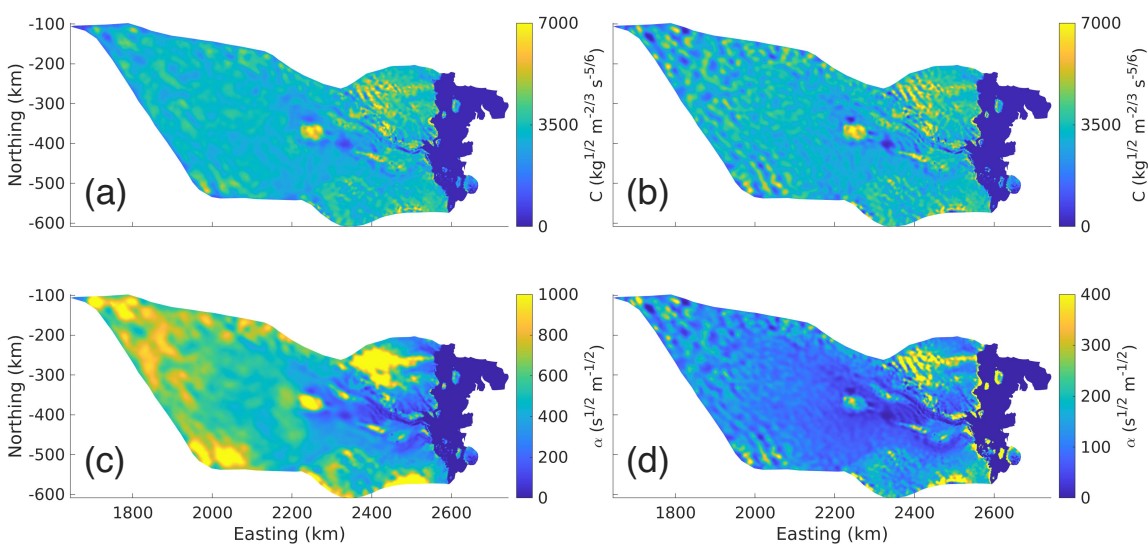

**Figure B3.** Basal Friction coefficients from inversion (note that the units and the colorbar are not the same in each subplot). **(a)** Schoof basal friction coefficient ($C$, $\mathrm{kg}^{1/2}\,\mathrm{m}^{-2/3}\,\mathrm{s}^{-5/6}$) from $N_G$ run; **(b)** Schoof basal friction coefficient ($C$, $\mathrm{kg}^{1/2}\,\mathrm{m}^{-2/3}\,\mathrm{s}^{-5/6}$) from $N_O$ run; **(c)** Budd basal friction coefficient ($\alpha$, $\mathrm{s}^{1/2}\,\mathrm{m}^{-1/2}$) from $N_G$ run; and **(d)** Budd basal friction coefficient ($\alpha$, $\mathrm{s}^{1/2}\,\mathrm{m}^{-1/2}$) from $N_O$ run.

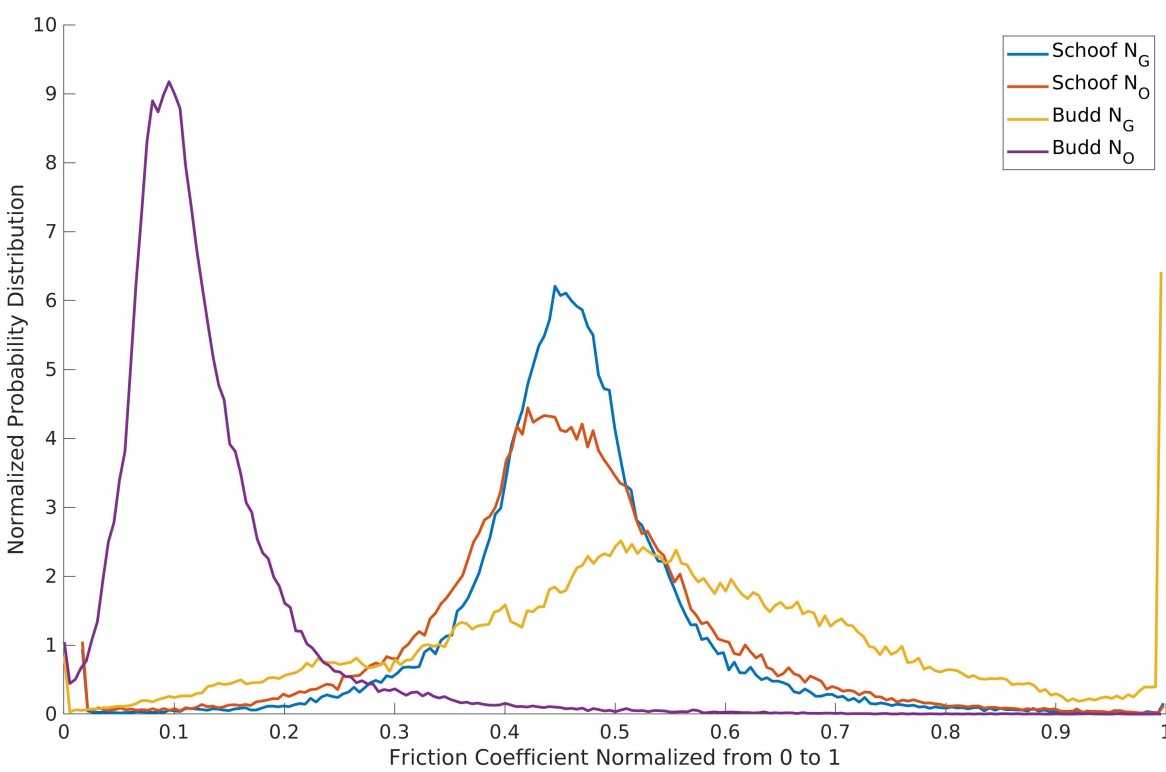

**Figure B4.** Probability distributions of the normalized basal friction coefficients for the Budd ($\alpha$) and Schoof ($C$) friction laws with GlaDS $N_G$ and prescribed $N_O$ effective pressures. The distributions are normalized to their maximum allowed values ($7000\,\mathrm{kg}^{1/2}\,\mathrm{m}^{-2/3}\,\mathrm{s}^{-5/6}$ for Schoof and $1000\,\mathrm{s}^{1/2}\,\mathrm{m}^{-1/2}$ for Budd)

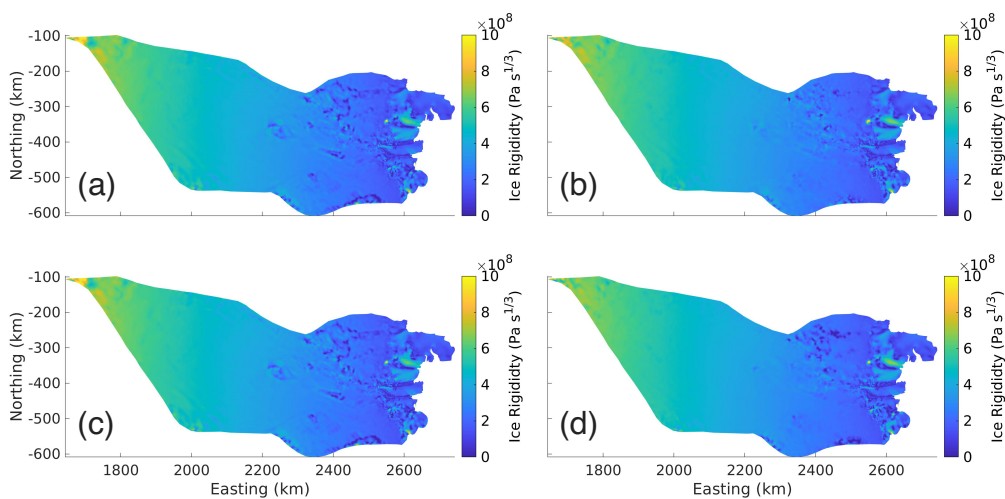

**Figure B5.** Ice Rigidities from inversion ($\tilde{B}$, Pa s$^{1/3}$). **(a)** Ice rigidity from Schoof with the GlaDS effective pressure ($N_G$) run; **(b)** Ice rigidity from Schoof with the typically prescribed effective pressure ($N_O$) run; **(c)** Ice rigidity from Budd with the GlaDS effective pressure ($N_G$) run; and **(d)** Ice rigidity from Budd with the typically prescribed effective pressure ($N_O$) run.

## Appendix C:  Empirical effective pressure parameterization

We propose a new effective pressure parameterization ($N_E$) that is empirical in nature and based on the GlaDS effective pressure output in the Denman-Scott catchment. Effective pressure is defined as $N = p_i - p_w$ where $p_i = \rho_i g H$ is the ice overburden pressure and $p_w$ is the subglacial water pressure. In this equation, $p_i$ is well known; however, $p_w$ is generally poorly known in Antarctica due to few direct measurements. Hence, a more robust parameterization of the effective pressure requires a physically-realistic prescription for this subglacial water pressure. The proposed parameterization uses the fact that subglacial water pressure has a high correlation with ice overburden pressure ($r^2 = 0.998$ over the Denman domain) and that subglacial water pressure as a fraction of ice overburden pressure maintains some value for small ice thicknesses and saturates near one for large ice thickness. We suggest the following parameterization for subglacial water pressure:

$$p_w = \rho_i g H \left( r_l + (1 - r_l) g_x(H) \right), \tag{C1}$$

where $r_l$ is the maximum allowable ratio of effective pressure to ice overburden pressure and $g_x(H)$ is a saturation term. We specify $g_x(H)$ as follows:

$$g_x(H) = \frac{H^x}{\tilde{H}^x + H^x}. \tag{C2}$$

Here, $\tilde{H}$ is a constant thickness, which is chosen so that an area where water is expected to be pressurized with fraction of overburden $\gamma$ will reach this level of pressurization by ice thickness $H_t$. This yields an $\tilde{H}$ of the following form:

$$\tilde{H} = \left( \frac{1 - \gamma}{\gamma - r_l} \right)^{1/x} H_t \tag{C3}$$

The only parameter left to be determined is $x$, and here we choose a value of $x$ so that at a typically small ice thickness, $H_s$, the subglacial water pressure as a fraction of overburden pressure will be slightly above the minimum subglacial water pressure allowed, $p_w(H) = (r_l + \epsilon)\rho_i g H$, where $\epsilon$ is a constant introduced with the purpose of having the saturation term concave down even in areas where there is low ice thickness on the glacier.

We rearrange Eq. (C3) to solve for $x$ as follows:

$$x = \frac{\ln\left( \frac{1 - r_l}{\epsilon} - 1 \right) + \ln(\gamma - r_l) - \ln(1 - \gamma)}{\ln(H_t) - \ln(H_s)}. \tag{C4}$$

Using Eq. (C1), we derive the following expression for the effective pressure, with $\tilde{H}$ from Eq. (C3) and $x$ from Eq. (C4):

$$N_E = \rho_i g H (1 - r_l) \frac{\tilde{H}^x}{\tilde{H}^x + H^x}. \tag{C5}$$

The empirical nature of this parameterization brings into question its physical validity. Unlike $N_O$ and the Brondex et al. (2017) prescribed effective pressure ($N_B$), the effective pressure proposed in Eq. (C5) does not have complete hydrological connectivity to the ocean at the grounding line. However, $N_O$ and $N_B$ can obtain values of negative or zero water pressure, which is nonphysical. The effective pressure proposed in Eq. (C5) can reach a minimum water pressure of $p_w = r_l \rho_i g H$ meaning that it will always give a physically possible value of effective pressure.

Delving further into the implications of these effective pressure parameterizations, we see that the hydraulic potential is given by $\phi = N_O - N$ (Werder et al., 2013), which means that using $N_O$ gives a constant hydraulic potential and a stagnant subglacial hydrologic system. $N_B$ implies constant hydraulic potential for bed elevations below sea level and $\phi = \rho_w g B$, where $B$ is the bed elevation, for bed elevations above sea level. Which means that water does not flow when the bed elevation is below sea level, and it flows entirely due to gravity and bed slopes when the bed elevation is above sea level. Taking the gradient in the hydraulic potential for $N_E$ yields the following equation:

$$\overrightarrow{\nabla}\phi = \rho_i g f_B(H)\overrightarrow{\nabla}H + \rho_w g \overrightarrow{\nabla}B = \rho_i g\left(f_B(H)\overrightarrow{\nabla}S + \left(\frac{\rho_w}{\rho_i} - f_B(H)\right)\overrightarrow{\nabla}B\right) \tag{C6}$$

$$f_B(H) = 1 - (1-r_l)\frac{1 + (1-x)(H/\tilde{H})^x}{(1+(H/\tilde{H})^x)^2} \tag{C7}$$

Here, $f_B(H)$ is a dimensionless factor which describes the extent to which ice thickness gradients play a role in the hydraulic potential gradient. Cuffey and Paterson (2010) considered the gradient of the hydraulic potential with a spatially constant fraction of flotation and arrive at the form of Eq. (C6) with $f_B(H)$ replaced by the fraction of flotation, $f_w$. Here, $f_B$ is not the fraction of flotation but is related to it via Eq. (C8) for $f_w \geq r_l$.

$$f_B = -\frac{xr_l}{1-r_l} + \frac{2+(x-1)(1+r_l)}{1-r_l}f_w - \frac{x}{1-r_l}f_w^2. \tag{C8}$$

$f_B$ is quadratic in $f_w$ and $f_B \geq f_w$ for $f_w \in [r_l, 1]$. This means that for $N_E$, gradients in surface elevation play a more important role in water routing than when constant fraction of flotation is considered. The importance of surface elevation gradients reaches a maximum when $f_w = 0.924$, corresponding to $f_B = 1.04$, values of $f_w$ above this threshold will actually result in smaller values of $f_B$ and a decreased role in the surface elevation gradient which is at odds with the analysis of Cuffey and Paterson (2010). Modeling (Dow et al., 2014, 2020) outputs, including those from our Denman analysis, suggest that the fraction of flotation is not uniform (Fig. 6). Compared to using $N_O$, this parameterization provides a more intuitive picture of the subglacial hydrological system, where water can flow throughout the entire domain, and where flow is dependent on both the basal topography and the ice thickness/surface topography, as is expected.

Except for the lack of complete hydrological connectivity to the ocean, the proposed parameterization in Eq. (C5) agrees more closely with our understanding of subglacial hydrology than both $N_O$ and $N_B$. The improved performance over $N_O$ and $N_B$ for the Denman Glacier is encouraging, and warrants further exploration, including exploring the validity of this parameterization compared with subglacial hydrology model outputs and its potential use in future modeling studies where full ice sheet-subglacial hydrology model coupling is unfeasible. Improvements to this parameterization, such as adding in the effects of ice velocity, may improve upon the performance of this parameterization.

*Code and data availability.* We use version 4.21 of the open-source ISSM software, which is freely available for download from https://issm.jpl.nasa.gov/download/. The datasets used to initialize GlaDS and ISSM are publicly available and cited in the main body of the text. GlaDS hydrology outputs and ISSM friction coefficients for the Denman-Scott catchment are available at Zenodo repository 8139725

*Author contributions.* KRM designed and completed the ISSM model runs and produced figures; CD ran the GlaDS model; FSM and CD provided project direction. All authors wrote the manuscript.

*Competing interests.* The authors declare no competing interests are present.

*Acknowledgements.* FSM was supported under an Australian Research Council (ARC) Discovery Early Career Research Award (DE210101433) and the ARC Special Research Initiative Securing Antarctica's Environmental Future (SR200100005). CFD was supported by the Natural Sciences and Engineering Research Council of Canada (RGPIN-03761-2017) and the Canada Research Chairs Program (950-231237). We thank the Digital Research Alliance of Canada for access to supercomputer resources.

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
