# Peer review of "Basal conditions of Denman Glacier from glacier hydrology and ice dynamics modeling"

_The Cryosphere, 2023_

## Referee Comment (RC1)

**General comments**

In this study, the authors have investigated the coupled interactions between the subglacial hydrological system and the ice sheet through the basal friction coefficient – an important and challenging tuning parameter used in friction laws – and its dependence on the form of the effective pressure. They also proposed a new empirical formulation of the effective pressure. Moreover, they highlighted the importance of subglacial processes on ice-sheet dynamics and conclude the need for geophysical observations of these processes would improve models.

This paper matches the quality criteria required by The Cryosphere. I have only a few comments and recommendations to improve the quality and scientific rigour of the manuscript as well as its understanding for readers that are less familiar with the subject and the tools used.

My first general comment concerns the choice of the Budd sliding law. Knowing that the value of $m$ has a major influence on ice dynamics, why hasn't the Budd sliding law with the exponent $m = 3$, which is the most commently used value instead of m=1? Wouldn't it make it easier to compare to the Schoof sliding law?

My second general comment concerns the calculation time required for a more complex hydrological model. At what time and space scale does a more complex model (such as GlaDS) make a significant and crucial difference in glacial dynamics and does it compensate for the additionnal time of calculations used ?

My third general comment concerns the figures. If a standardisation does not allow the scales to be the same, it must be stipulated in the text for all figures concerned so that there is no misunderstanding. Also, note that this difference in scale does not allow the same analysis and comparison resolution. Also, it is better to have complete captions (variable-symbol-units) and the same than the legend (the text written next to the colorbar). It also is more readable if both limits of the scale are written.

My final general comment concerns how the types of effective pressures are expressed in the text. I think it would be better to define in an equation $N_o$ from the beginning and not to repeat it again in the text. Why choose $N_o$ and not choose the 'limited version' by Brondex et al, 2017 from the start? I don't quite understand how considering the two brings a lot of added value (especially by comparing figures 6c and 6d). If you decide to keep both, then set a symbol for N Brondex to avoid repetitions in the text. Finally, when we see the large difference in N values between $N_o$ and $N_G$ in Figure 3, it would be good to explain how a single variable can be considered with such different values.

I therefore propose some small changes in the text or the figure calls to improve clarity and understanding. The main thing is the insistence on the terms « basal » and « subglacial » which for me are important to keep throughout the text. Finally, I also propose to elaborate on more technical details with respect to the tools used and the choice of parameters.

L37-L117. Please to specify whether these negative effective pressures are stable, at the steady state, seasonally dependent…Please add information on the stability of this case and also whether it varies over time (depending on the seasons or the tides). It could be interesting to add a sentence mentioning that the presence of zones with negative effective pressure may be persistent and does not lead to instability.

L64. Stipulate if the model reaches the steady state after 10,000 days.

L78-L346 : If you don't add information about the ice rigidity calculation, please add a reference.

L140. As you explain that low effective pressures were associated with faster flow, give a brief explanation of this case.

L173. Use the reference Huybrechts, 1990.

Remove Budd and Jensen, 1987.

If you use the references, Johnson and Fastook, 2002 and Lebrocq et al., 2009 define the hydraulic potential = 0.

L180. Accurately calculating the effective pressure is important improve ice-sheet models. However, effective pressure parametrizations used in Brondex et al., 2017 and Kazmierczak et al., 2022 are simplified for computational purposes and numerical stability, especially when the study is either focused on grounding lines or experiments are done on a continental scale. Please, add the concept of model complexity for the subject under study.

L180 : Previous studies have investigated alternative parameterizations for the effective pressure **in the absence of a coupled ice sheet-subglacial hydrology model**.
In Kazmierczak et al., 2022., the ice-sheet model is coupled to the simple subglacial water routing from Lebrocq et al., 2009 by the subglacial water depth and by the flux.
This sentence should be modified by […]**complex subglacial hydrology model due to the** computational time.

L196 : Add a reference on the **alpine-like hydrology system**.

L220 : Since equation 1 does not include $m$, this sentence is not clear. Then perhaps add the $m$ in equation 1 and mention that $m=1$.

L285-286 : Specify with which formulation of the effective pressure this conclusion is made.

L353 : Add a reference for the Paterson function.

L362 (éq. C2) : Why is Cmax (0.8) different to the value used in Brondex et al., 2017 (0.5)?

Figure 1. First time you mention Denman-**Scott** catchment. It is better to mention it in the text beforehand.

Figure 4. Could you explain how the very different effective pressures between $N_G$ and $N_O$ do not significantly impact the basal friction coefficient in the Schoof basal sliding law but impact significantly the basal friction coefficient in the Budd basal sliding law.

Table 1 : I never used GlaDs but as some parameters are different than in Werder et al., 2013, why? Specificity from Antarctica or this specific catchment? How did you obtain these parameters? The ice flow constant is the same for cavities and channel?

**Technical corrections**
**Abstract**
L10 Budd **friction** law

L10-14 Schoof **friction** law

**Introduction**
L19. Mention the dataset of Adusumilli et al., 2020 in the references. Because the data you mention are not in the paper itself.

> ➔ Adusumilli, Susheel; Fricker, Helen A.; Medley, Brooke C.; Padman, Laurie; Siegfried, Matthew R. (2020). Data from: Interannual variations in meltwater input to the Southern Ocean from Antarctic ice shelves. UC San Diego Library Digital Collections. https://doi.org/10.6075/J04Q7SHT

L20. Mention the dataset of Morlighem et al., 2020 in the references.

> ➔ Morlighem, M. (2020). MEaSUREs BedMachine Antarctica, Version 2 [Data Set]. Boulder, Colorado USA. NASA National Snow and Ice Data Center Distributed Active Archive Center. https://doi.org/10.5067/E1QL9HFQ7A8M.

L20. Denman **Glacier**

L21. « containing 1.5 m of sea level equivalent. » ➔ source? (rewrite Brancato et al., 2020)

L26-27. Replace « Coulomb laws » by **regularized Coulomb friction laws**.

Because in a Coulomb friction law s.s., the basal shear stress is independent of the basal sliding velocity.

L27. of the  Budd

L27 & 29. On the **basal** sliding velocity

L28. **subglacial** water pressure

L29. Provide examples of the « other quantities » (i.e. rugosity of the bed…)

L29. Why unexepected?

L30. **basal** friction coefficient (2)

L31. **basal** sliding velocity

L32. **basal** friction coefficient

L38. Remove (i.e. the ice overburden pressure) or modify it by (i.e. when subglacial water pressure exceeds the ice overburden pressure)

L41. **Subglacial** water pressure

**Methods.**
**2.1 GlaDS Setup**
L58. Source for the surface velocities? MEaSUREs v2?

L59. Same comments in L20 about the Morlighem et al., 2020 source

**2.2 ISSM Setup**
L68. Add reference for SSA: Morland, L.: Unconfined Ice-Shelf Flow, in: Dynamics of the West Antarctica Ice Sheet, edited by: van der Veen, C. J. and Oerlemans, J., Kluwer Acad., Dordrecht, Netherlands, 99–116, 1987.
L72. Same remark in Figure 1.
L75. **basal** friction
L76. **basal** friction law
L77. Same comment in L59.
**2.2.1 Solving for basal friction coefficients**
L85. Why eq 2 and eq C2 are not the same? If it's a mistake, the ()^m is missing in the denominator. Is it possible to isolate the coefficients of friction in order to make them more visible?

L87. Source of the Iken's bound missing and please refer to Appendix C.

L90. **Appendix C**, Table C1

L98. For the $N_o$ equation complete that B takes negative values below sea level and defines the variables.

**Results**
**3.1 Subglacial hydrology**
L105 and L106. Even if it's clearly written in L103, mention that the data indicated (length of the channel and the flow) come from the GlaDs modelling.

L108. If the two branches of 80 and 52 km of the Denman channel are figure 2a (iii) and (iv), mention the figure in the text.

L111 & Fig.2d(v). I know that the choice of the limits of the legend is there to allow a better reading, but I find it strange to mention 25 m of thickness whereas the legend stops at 10 m.

L112. We cannot really see on the figure that the strongest flux is toward Denman, so do not mention the figure but rather the data.

L116-117-118-119 **subglacial** water pressure

L121. Show these zones in the Fig. 2b.

**3.2 Ice dynamics and inversion**
L127. Schoof friction law **(éq. 2)**

L132. slow **ice** flow (maybe mention the Fig. 1b with the surface speed)

L133. Locate the Shackleton Ice Shelf in one of the figures (e.g. Figure 1).

L135. a space is missing between Fig. and 4c

L136. **basal** friction coefficient

L138. faster **ice** flow (**ice** surface speeds […]

L139. slow **ice** flow (**ice** surface speeds […]

L141. **basal** friction coefficient and please refer to (Fig. 5b).

L133-138-140. The limits given to consider an ice flow faster or slower are not the same, why?

L143. Glacier**s**

**Discussion**
**4.2 Effective pressure and basal sliding laws**

L182. : the till parametrization used in Kazmierczak et al. 2022 is from Bueler and van Pelt, 2015

L200. : The relationship between low effective pressures and low basal friction did not hold for the Budd friction law, **(Fig 3 (a), Fig 4 (e))** despite a stronger negative correlation between the friction coefficient and surface speed for the Budd friction law compared to the Schoof **friction** law **(Fig 5(a),(c)**).

L214. : when using a Weertman friction law**, which not considered the strong dependence of $\tau_b$ on the effective pressure,**

L215. : Schoof **friction** law
L216-218-221. : **Regularized** Coulomb friction laws
L229. : This study by Kazmierczak et al. (2022) examined the impact of different representations of the effective pressure – approximated in turn by height above buoyancy (van der Veen, 1987; **Huybrechts, 1990**, Winkelmann et al., 2011, **Martin et al, 2011**), reduced by the subglacial water pressure (**modified from Bueler and Brown, 2009**) or sliding related to water flux (Goeller et al., 2013) **from a simple subglacial water routing model (Lebrocq et al., 2009)**, and the effective pressure in a till (**Bueler and Van Pelt, 2015**) – on ice mass loss from Antarctica over the 21st Century.

**4.3 Empirical Parametrization**

L 254-255. : γ or δ ? use the same symbol
L260. : define Brondex et al., 2017 or write something like […] No on all the domain or, following the condition of Brondex et al., 2017, exclusively below sea level […] Fig. 6**a, b, c, d**
L264. : define the « saturation term »
L267. : 77.% (use the same number of significant digits)
L279. : **basal** friction laws

L280. : **basal** friction coefficient

L285. : Schoof **friction** law

L289. : **geophysical** observations

**Appendix A**

L304-308-313. : pressure (it is correct but keep pressure or pressure**s**)

L305. : domain

L306. : speed

L307 + Table A1 : $500 \geq v < 1000 \, ma-1$. The sign is not correct ?

L320-321. : (e.g. Bueler and Brown, 2009; Bueler and Van Pelt, 2015, van der wel et al., 2013, Huybrechts 1990, Kazmierczak et al., 2022 ; ; ; van der Veen, 1987; Winkelmann et al., 2011)

Figure A1. (a)-(b) Effective pressure $N$ (MPa) Note that the color bar scale is not the same. It is better if both limits of the scale are written.

**Appendix B**
L324-332-333-334-335-336. : **basal** friction coefficient

L327. : For Rignot, 2017, the dataset I founded has to be referenced like that :

➔ Rignot, E., J. Mouginot, and B. Scheuchl. (2017). MEaSUREs InSAR-Based Antarctica Ice Velocity Map, Version 2 [Data Set]. Boulder, Colorado USA. NASA National Snow and Ice Data Center Distributed Active Archive Center. https://doi.org/10.5067/D7GK8F5J8M8R. Date Accessed 04-20-2023.

And following the informations given to this link : https://nsidc.org/data/nsidc-0484/versions/2 these references has to be included :

➔ Mouginot, J., B. Scheuchl, and E. Rignot. 2012. Mapping of Ice Motion in Antarctica Using Synthetic-Aperture Radar Data. *Remote Sensing*. 4. **DOI: 10.3390/rs4092753.**
➔ Mouginot, J. et al. 2017. Comprehensive Annual Ice Sheet Velocity Mapping Using Landsat-8, Sentinel-1, and RADARSAT-2 Data.. *Remote Sensing (in press)*.

L336. : the precribed effective pressure B1? It is not clear… explain in the text (like in the Table B1. Caption) that the prescribed effective pressure is $N_o$.

L337. : effective pressure **calculation/representation**

Table B1. : **basal** friction - Schoof **basal** friction coefficient – Budd **basal** friction coefficient

Figure B1. Schoof **basal** friction coefficient

**Appendix C**
L344. : B is the bed elevation (m) **taking negative value below sea level**

L348. : same comments in L327

Figure C1. Budd **basal** friction coefficient – '.' Missing at the end of the sentence.

Figure C2. Schoof **basal** friction coefficient– '.' Missing at the end of the sentence.

Figure C3. On the figure : S**c**hoof (x2) + specify with $N_o$ and $N_G$– In the caption :**Basal** friction coefficients - Schoof **basal** friction coefficient (x2) Budd **basal** friction coefficient (x2). Add the units and that the colorbar scale is not the same. It is better if both limits of the scale are written.

Figure C4. **basal** friction coefficients – prescribed ➔ $N_o$?

Figure C5. It is better if both limits of the scale are written. – Caption : '.' Missing at the end of the sentence.

Table C1. Budd **basal** friction coefficient -- Schoof **basal** friction coefficient

L352. : Budd **basal** friction coefficient

L358. : **basal** friction coefficient

L361. : same comment L85

L362. : Schoof **basal** friction coefficient – Add a reference or an explanation for the Iken's bound – **basal** friction coefficient

L363. : **basal** friction coefficient

L364. : **basal** friction coefficient

**Appendix D**
L374-376. **ice** overburden **pressure**

Figure D1. It is better if both limits of the scale are written.

Add the units and that the colorbar scale is not the same. It is better if the caption and the legend are the same (with symbol and units) → Effective pressure (N) (MPa)/ friction coefficient for the Schoof friction law C […]/surface speed ($u_s$) (m a$^{-1}$)

Figure D2. It is better if both limits of the scale are written.

Add the units and that the colorbar scale is not the same. It is better if the caption and the legend are the same (with symbol and units)

**Appendix E**
L380. : define $\rho$, g and H in the $p_i$ equation

L381. : **subglacial** water pressure

L394. : to avoid a confusion with the power law exponent, maybe chose another letter than *m*.

L395. : **subglacial** water pressure (x2) –  water pressure **allowed**

**Figures**
**Figure 1.**
- (a) Bed elevation **(unit missing)** from Bechmachine v2
    o Detail : it's « Bed elevation » in the caption and « Bed topography » on the figure. Use the same formulation.
- (Morlighem et al., 2020) same comment as L20.
- (Rignot 2017) same comment as L327. → And modify by « Rignot **et al.,** 2017 »
- It misses a '.' At the end of the caption.

**Figure 2.**
- modify overburden pressure by **ice** overburden pressure.
- On L114, you mention « the nothern branch » of the Denman glacier, but in the caption you mention in Fig2a (iii) a western branch and in (iv) an eastern branch. I'm

confused. Please, is it possible to have a clarification/harmonisation between the figure and the text and also to say which one is longer compared to what is written on L108.

**Figure 3.**

Place the figure in the subsection 3.2.
(a) Effective pressure (MPa) calculated by GlaDS, $N_G$, [...]
(b)  effective pressure (MPa) [...]

**Figure 4.**

Caption : **surface** velocity/ Schoof **friction** law (2)/ Budd **friction** law (2)

**Figure 5.**

Fig. 5b S**c**hoof

It is better if both limits of the scale are written.

Fig. 5d for $r^2$ = write the same number of significant digits.

In the caption : The red line is the linear line of best fit, and the slope of this line is reported **($r^2$)**.

**Figure 6.**

a-b-c-d for the colorbar, maybe write « $N_G$/Pi » - « $N_E$/Pi » - « $N_o$/Pi » « $N_B$/Pi »? and explain it completely in the caption. Use the same wording in the legend as in the caption.

(f) The difference between the proposed empirical parameterization of effective pressure ($N_E$) and the GlaDS effective pressure ($N_G$) as a fraction of ice overburden pressure.

As figure f is explained before firgure e in the text, I would switch them.

**Table 1.**

Channel conductivity

Ice density kg.m$^{-3}$

Sheet **w**idth **b**elow **c**hannel

**Table 2.**

In the caption, use the same formulation than in the fig. 3 caption : [...] to **the ice** overbruden pressure [...] **formula**

---

## Referee Comment (RC2)

**Review of: Basal conditions of Denman Glacier from glacier hydrology and ice dynamics modeling**

May 11, 2023

**1 General impression**

The article presents the incorporation of a previously determined effective pressure in the inversion of basal friction coefficients for a shallow-shelf ice flow model of the Denman-Scott catchment for two sliding laws, namely a Budd-sliding law with linear coefficient and a regularised Coulomb law, as presented by Schoof (2005). This is a timely topic and in general I see it suited to be published in The Cryosphere.

I have three major points I would see necessary to be addressed before the publication can proceed - I put them in a separate section below.

The article in general is concisely written. The majority of figures is good to read and conveys the information well. On top of the major points, there are a few questions and suggestions I placed in my review. I hope that these may contribute to improve the quality of the manuscript.

**2 Major points to be addressed**

The first item I would see to be addressed is a **more detailed description of the inputs to the GlaDS simulation**, in terms of parameters but mainly the imposed slip velocity and melt-water production. You seem to run GlaDS as a pre-processing step to produce $N_G(\vec{x})$ for the inversions of the specific friction coefficients. Yet, the hydrology computations needs input in form of a slip velocity and a water-production that themselves will be a result of the ice-flow dynamics and hence the friction coefficients applied in the ice-flow model providing those. To me this appears to be a little bit of a cat-catches-its-tail problem. From the text (line 59): *Basal water and sliding velocity inputs are computed from ISSM (Seroussi et al., 2019)*, I would conclude that you pick initial sliding and velocities from a completely different inversion, subject to certain constraints: *The sliding velocity acts to open up distributed system cavities and, at velocities greater than 800 $ma^{-1}$, can cause model instabilities and so is capped at this value.* Can you please

clearly state what ice-flow setup you base the GlaDS computation on? What are the approximations to the Stokes equation of this initial model? What has been used to represent the sliding-law and the applied effective viscosity therein - the latter also in terms of thermodynamics (if any) or damage? How do you deduce the water production from that result? If all this is addressed, I would also hope to see some conclusion if and if so, how this initial settings could have influence on the distribution of the effective pressure arising from GlaDS and if there further might be a possibility that they could pre-condition the result of the following inversion. For the pasteurisation in GlaDS itself, I have difficulties in lack of any equations and symbols to interpret values displayed in Table 1 (see detailed comments).

The second topic I would ask to have elaborated is a **discussion on how the approximations to the Stokes equations could influence your inverted slip coefficients in regions of significant vertical shear**. From what I read in the text, I understand that you are applying the Shallow Shelf Approximation (SSA). To my understanding, the dynamics in fast flowing outlet parts will be well represented by SSA. Of my concern are rather those regions, where the onset of the outlets takes place, where I would expect internal vertical ice deformation to still play a significant role. Ignoring this component, in my view, would have a bias to over-predict the slip. This highly also links to the missing detailed information on which input the GlaDS simulations are based on. If this is also based on SSA, altered slip can bias the hydrological system (as it alters melt-water production and slip-induced opening rates) over a wide range of the catchment area.

Finally, I would also like to **better understand the whole inversion procedure and the impact of the rigidity inversion on your results**. In my view this is best achieved by presenting the inversion procedure in terms of equations. You shortly mention that you invert for rigidity of the ice (line 77), which could be somehow interpreted as inverting for a (depth-averaged) damage or temperature field. You are presenting a result of this rigidity distribution in Appendix C (which is hard to interpret) but – in my view equally important – do not provide some kind of physical interpretation of it. Also, starting the averaged viscosity based on the (atmospheric?) temperature distribution given by RACMO (line 353), in my view would need some explanation. How much, for instance, does the inverted rigidity differ from the one used to compute the bedrock velocity used in GlaDS simulations? In my opinion the reader already would benefit if you could introduce the equations of the SSA system where rigidity is plugged in (to my understanding how you define your averaged viscosity in the SSA model)

**3 Detailed comments**

Listed in order of their appearance. If some of the comments link to the main points, I indicate it - else, they are mainly meant as suggestions on how to improve readability of the manuscript or corrections to typos. I did not sync anything with the already published other review - so, sorry for cross-postings.

line 26 *One such parameter is the basal friction coefficient, which is a key component of friction laws including the Weertman (1957), Budd et al. (1979) and Coulomb laws (Schoof, 2005; Gagliardini et al., 2007).* I would refrain from calling these laws to be *Coulomb laws* but use *regularised Coulomb laws* instead. There are several occurrences of this term in the text

line 27 Typo: ...*case of the  Budd* ...

line 32 *Therefore, a friction coefficient that is both smooth – has little local variability – and has limited domain wide trends is desirable.* Is that really the case in all flow situations? Could there not be situations of either a drastic change in the properties of the substrate underneath the glacier and/or the thermodynamic conditions that would also imply a significant change of those coefficients?

line 59 *Basal water and sliding velocity inputs are computed from ISSM (Seroussi et al., 2019).* This directly links to one of my main points above: As you state yourself that the flow conditions (I presume you mean ice-dynamics) are of essence, I think you should declare in the main part of the manuscript what ice-dynamic input you used to drive the GlaDS simulations. Are you directly using the results from the cited paper (Seroussi et al., 2019)? Even then, in my opinion, the manuscript would benefit from spelling this out (methods, input data, at which time of this simulation you pick the ice-dynamics input?).

line 63 *The model is run for 10,000 days, providing outputs including channel size and discharge, distributed system discharge, water depth, and effective pressure.* What was the motivation for 10k days? Was this necessary to reach some steady state? If not, which point in time was chosen for extracting the effective pressure distribution? This links to the main point of critics, namely, the in my opinion missing details for the GlaDS-step.

Table 1 I appreciate that you report on values of used model constants, which adds to the reproducibility of your experiments. You, though, report them without context to equations (hence also not providing symbols), which to me (and perhaps any reader not using GlaDS) makes

them difficult to interpret. This links to the first main point of critics. Also, some background (sentence of motivation or a reference) on the choice of the numerical value could enhance the understanding of the reader. To pick one example: You report the "Ice flow constant", which I understand to be linked to the rate factor in Glen's flow law (though confused by the sign in the exponent of $Pa$ in units) as $2.5 \times 10^{-25} Pa^{-3} s^{-1}$. Provided I interpret it correctly, such value refers (e.g. Greve and Blatter, 2009) to a relative ice-temperature of around $-10°C$. What motivates this setting? How does this choice influence the inversion for slip coefficients and rigidity?

line 66 *ISSM is a finite-element model that uses an anisotropic mesh to simulate ice dynamics.* Just a suggestion: I understand *anisotropy* as a local variation depending on direction. I, personally, would rather use the terms "non-uniform" or "adaptive in size".

line 72 *The ISSM mesh is comprised of 66,518 nodes, with anisotropic mesh refinement for faster flowing ice using the MEaSUREs v2 ice surface speed (Rignot et al., 2011; Rignot, 2017).* How are you using the MEaSUREs ice velocity product to refine the mesh?

line 77 *The ice rigidity is calculated using inverse methods.* Can you please elaborate? Are you performing a dual-parameter inversion for sliding and rigidity? Or is this another step on top of the previous one? And what effects do you think you cover in the SSA application by inverting for rigidity (vertically averaged temperature, damage)? And what effects you neglect by the simplified physics of your ice-flow model? Are you using this rigidity only in the final inversions or already in the GlaDS runs and how does this connect to the value given in Table 1?

This links to my second point of main critics. To elaborate from my side: Temperature, in particular, is reported to be an important factor in what comes to the quality of inversions using full-stress (a.k.a. full-Stokes) models (see, e.g., Zhao et al., 2018), in particular in the regions of onset of the fast-flow outlets, where internal vertical deformation has a significant role (an effect that is not included in the here applied SSA approximation).

line 98 Variables $H$ and $B$ are not explained right after their first occurrence. Similar, you lack definition for ice and water density and the absolute value of the acceleration by gravity, $\rho_i$, $\rho_w$ and $g$, respectively. These definitions appear somewhere later in the text. $H$ and $B$ seem to be the thickness and the elevation of the ice-sheet bottom. Please, add definitions of symbols at their first occurrence. For me it would be also of benefit to directly annotate the N with a subscribed 0 ($N_0$) in the formula.

Figure 2  These are suggestions: Perhaps some elaborated color-map to distinctively highlight the excess to flotation in (c) and/or the 100% as iso-line would in my opinion enhance the information of this figure.

Figure 4  Please, add an explanation what the black line in the right column represents. Same situation also in Figure D1 – Figure A1 contains the correct description in its caption.

line 147  *This leads to a comparatively greater standard deviation in the Budd friction coefficient compared with that of Schoof ($1240\,kg^{1/2}m^{-2/3}s^{-5/6}$ for Schoof and $250\,s^{1/2}m^{-1/2}$ for Budd).* I have difficulties to interpret the relative magnitude of standard deviations between two friction coefficients based on different physics and hence of different units. Thus, I would either report normalised values (as you seem to do in your graphs) or drop this sentence.

line 148  *Despite this, the Budd friction coefficient is generally smoother than the Schoof friction coefficient, which may be a consequence of the choice of the Tikhonov regularisation coefficient used in the inversion procedure.* For the reader, I think it would be beneficial if you could explain how exactly you determined the optimal regularisation parameter in a – at least for me – difficult to interpret L-curve given in Figure C1? Was, for instance, the relation between misfit and regularisation smaller as in Schoof?

line 166  *. . . irrespective of the degree of regularization (Appendix C).* What exactly do you mean by "degree of regularization"?

line  *Our study uses inverse methods to calculate the friction coefficient for a given $\tau_b/N_G$.* Is this a hint that you invert for traction and then interpret in terms of the specific friction law?

line 285  *That is, using the Schoof law, regions of lower effective pressures tend to also have lower simulated basal friction and faster flow – evidence for the controlling role of the hydrological system.* Is that not also the case for Budd-sliding? To my understanding, the main difference is the more complex relation in the regularised Coulomb law to the effective pressure, yet, Budd-law has an inverse proportional relation of the sliding speed to the effective pressure and should result in faster flow over lower pressures.

line 303  *The domains of the ISSM and GlaDS models differ, with the GlaDS domain being a subset of the ISSM domain.* What was the main motivation to not make these domains the same? Is it so expensive to run GlaDS on a wider domain? Or are there issues with boundaries? Would that not be worth it to get rid of all these extrapolation issues?

| Figure B1 | To me the coloured lines have a different colour to the ones in the legend, such that I am not able to retrieve significant information out of this graph. Also, please explain the accumulated probabilities at the lower and upper end of the spectrum (I guess it is because of the cap). |
|---|---|
| line 345 | *We use inverse methods to calculate the basal friction coefficient $\alpha$ from the Budd friction law and the ice rigidity in the Glen flow law. The inverse method works to reduce the mismatch between the simulated and observed velocities, here taken from MEaSUREs v2 (Rignot et al., 2011; Rignot, 2017), by minimising a cost function that includes both linear and logarithmic velocity misfit components.* I (and perhaps some of the readers) would benefit from having the whole cost-function including regularisation terms being written out as equation in order to easier interpret what you are doing in your inversion. It would add to make your experiment more reproducible. I am also confused on how you introduce the inversion of rigidity (e.g., do you use a penalty term for deviation to prior?). This links to the third main point of critics. |
| Figure C1 and C2 | A suggestion to improve the information for the reader: Perhaps you can distinctively mark the parameter configuration of the run that finally was chosen to provide the optimal inversion according to your L-curve analysis. |
| line 361 | *..., and $C_{max}$ is Iken's bound, here $C_{max} = 0.8 m^{-1/3} s^{1/3}$ .* As Iken's bound is related to the roughness of the bed (Schoof, 2005), can you please explain how the exact value reported here came to be? Maybe by backing it up with information on properties of the glacier substrate? |
| line 368 | *In these regions, the inverse method compensates by increasing the friction coefficient upstream of the anomalously low effective pressure, leading to an underestimate of surface speeds there compared with the observations. The surface speeds are also generally overestimated in the region of vanishing shear stresses.* This links to my suggestion to better discuss the implication of the SSA approximation. Could it be that this effect is pronounced by the fact that stress bridging between low and higher friction region is not represented in the model/approximation? |
| line Figure C4 | To me, this figure is not clear. I see 11 different colours in the graph and only 4 in the legend - I do not get a clear idea on which curve represents what probability distribution. |

line 376   *That is, 98% of the effective pressure in the ISSM simulation is derived directly from the GlaDS simulated effective pressure.* I would drop that to the previous sentence somewhat redundant statement.

line 415

$$\nabla\phi = \rho_i g f_B(H)\nabla H + \rho_w g \nabla B, \text{ (E6)}$$

$$f_B(H) = r_l \frac{1 + (1-m)(H/\tilde{H})^m}{(1 + (H/\tilde{H})^m)^2}. \text{ (E7)}$$

*Here, $f_B(H)$ is a dimensionless factor which describes the extent to which ice thickness gradients play a role in the hydraulic potential gradient.* To me it seems that the factor $f_B(H)$ is equivalent to what in basic literature is called "flotation fraction" (e.g., see Chapter 6 in Cuffey and Patterson, 2010), which you depict for GlaDS result in Figure 2c. If you agree, please, try to make that connection for the reader.

line 417   *It is seen that in the regime of larger ice thickness $f_B(H)$ goes to 0, and the gradient in the bed elevation becomes the sole control on the direction of water flow; in the regime of small ice thickness $f_B(H)$ goes to $r_l$ , and gradients in the ice thickness become of similar importance to gradients in the bed elevation. This is a more intuitive picture of the subglacial hydrological system, where water can flow throughout the entire domain, and where flow is dependent on both the basal topography and the ice thickness, as is expected.* As you claim that this is expected, in my view that would need to be backed up by reference(s). Provided, we agree that $f_b$ is equivalent to the flotation factor, for me it is even somewhat counter intuitive. Following standard literature (e.g., see Chapter 6 in Cuffey and Patterson, 2010) the flotation fraction has to go down to $f_B = 0.56$ for bedrock gradients (and not, actually, bedrock elevation itself) to reach same influence as surface (not thickness, though) gradients. In Figure 2c, though, the whole region west of 2200 km – which I connect to large thickness (although you do not provide a graph with ice thickness) – seems to be very close to a fully pressurised hydrological system (i.e. $f_B(H) \approx 1$), which would make me expect that surface gradients would almost by an order of magnitude dominate the flow direction of water. I, though, have difficulties to relate thickness to bedrock gradients, as the earlier are partly defined by the latter, i.e., $\nabla H = \nabla(S + B)$, if $S$ is the ice-surface elevation. To sum up: I do not see your statement that $f_B(H) \to 0$ for thick parts of the sheet is reflected in Figure 2c and neither in Figure 3a. A graph depicting $f_B$ and ice thickness might help me and perhaps some readers to understand.

**References**

Cuffey, K.M and V.S.B Patterson (2010). 4th ed. Butterworth-Heinemann. ISBN: 978-0-12-369461-4.

Greve, R. and H. Blatter (2009). *Dynamics of Ice Sheets and Glaciers*. Springer. DOI: 10.1007/978-3-642-03415-2.

Schoof, C. (2005). "The Effect of Cavitation on Glacier Sliding". In: *Proceedings of the Royal Society A* 461, 609–627. DOI: 10.1098/rspa.2004.1350.

Seroussi, H et al. (2019). "initMIP-Antarctica: An Ice Sheet Model Initialization Experiment of ISMIP6". In: *The Cryosphere* 13, 1441–1471. DOI: 10.5194/tc-13-1441-2019.

Zhao, C. et al. (2018). "Basal friction of Fleming Glacier, Antarctica – Part 1: Sensitivity of inversion to temperature and bedrock uncertainty". In: *The Cryosphere* 12.8, pp. 2637–2652. DOI: 10.5194/tc-12-2637-2018. URL: https://tc.copernicus.org/articles/12/2637/2018/.

---

## Author Comment (AC2)

**Reviewer Comment Response**

Dear Dr Karlsson, Elise Kazmierczak

We thank the reviewers for their constructive comments and the improvements that they will bring to the manuscript.

In what follows, the reviewer comments are in black, our responses to the reviewer comments are in blue, and suggested edits to the manuscript are italicised. The line references are to the revised manuscript unless otherwise specified.

In addition to the changes suggested by the reviewers, we have also condensed the appendices for improved readability, by merging all appendices that relate to sensitivity analyses into subsections of Appendix A. Appendix B gives more details on the inversion procedure, and Appendix C gives more details on the new, proposed empirical parameterisation for the effective pressure.

Best regards,
Koi McArthur and co-authors.

**1 Author Comments**

General comments

In this study, the authors have investigated the coupled interactions between the subglacial hydrological system and the ice sheet through the basal friction coefficient – an important and challenging tuning parameter used in friction laws – and its dependence on the form of the effective pressure. They also proposed a new empirical formulation of the effective pressure. Moreover, they highlighted the importance of subglacial processes on ice-sheet dynamics and conclude the need for geophysical observations of these processes would improve models.

This paper matches the quality criteria required by The Cryosphere. I have only a few comments and recommendations to improve the quality and scientific rigour of the manuscript as well as its understanding for readers that are less familiar with the subject and the tools used.

Thank you for your comments on the paper, which we have addressed below.

My first general comment concerns the choice of the Budd sliding law. Knowing that the value of m has a major influence on ice dynamics, why hasn't the Budd sliding law with the exponent $m = 3$, which is the most commonly used value instead of m=1? Wouldn't it make it easier to compare to the Schoof sliding law?

Note: we have chosen to use the same convention as Brondex et al. (2019) (their equation 6), such that $m = 1/n$ where $n$ is the exponent in Glen's flow law, so that the $m$ value can be more easily compared to that of the Schoof friction law. Hence, here $m = 1/3$ corresponds to the value of 3 which the reviewer suggests.

Thanks for this comment. We chose the value $m = 1$ because it has been used in many previous modeling studies (e.g. Åkesson et al., 2021; Åkesson et al., 2022; Yu et al., 2018; Choi et al., 2021; Baldacchino et al., 2022) that use ISSM as well. Using $m = 1/3$ with the Schoof law would lead to the same issue as the $m = 1$ case, of not having the same units of the basal friction coefficient. However, as the reviewer points out, the $m = 1/3$ case will lead to a similar dependence on the basal sliding speed as the Schoof friction law.

To address this, we ran an additional simulation using the Budd friction law with $m = 1/3$. We found strong negative correlations between the Budd basal friction coefficient and the effective pressure (-0.466 for $N_G$ and -0.592 for $N_O$) compared to the correlations when $m = 1$ is used (-0.304 for $N_G$ and -0.0260 for $N_O$). When $m = 1/3$ is used, the Budd basal friction coefficient counteracts the effects of the effective pressure more than in the $m = 1$ case, with higher values in areas of low effective pressure and lower values in areas of high effective pressure. This is particularly noticeable in the region of low/negative effective pressure centered around 2200 km northing and -400 km easting, where a large area of high basal friction coefficient develops for $m = 1/3$. The solution to this would be to raise the cap on effective pressure above 1 % of ice overburden pressure, but as we have already shown in Appendix A3 this would mean replacing a significant portion of the GlaDS effective pressure data with an effective pressure that is linearly proportional to the overburden pressure. We have prepared a description of this analysis for a new Appendix in the updated manuscript. The text from this appendix is as follows:

*Here, we compare the impact of using a nonlinear exponent $m = 1/3$ in the Budd friction law (Eq. 6; as per Brondex et al., 2017,0; Kazmierczak et al., 2022) to our results with the linear exponent $m = 1$ (as per Åkesson et al., 2021; Åkesson et al., 2022; Yu et al., 2018; Choi et al., 2021; Baldacchino et al., 2022). We follow the same setup outlined in the main*

*text and in Appendix B, capping the effective pressure at 1 % of ice overburden pressure. The L-curve analysis suggests a Tikhonov regularization value of 0.1 is optimal.*

*The $m = 1/3$ case had much smaller variance in the normalized basal friction coefficient compared to the $m = 1$ case (Fig. A3). That is, using $N_G$, the variance of the normalized basal friction coefficient was 0.232 for $m = 1/3$ and 0.385 for $m = 1$; using $N_O$, the variance was 0.532 for $m = 1/3$ and 0.628 for $m = 1$.*

*Despite the smaller normalized variance in the basal friction coefficients for the $m = 1/3$ case, there was a stronger negative correlation between the basal friction coefficient and the effective pressure in areas where ice surface speeds are greater than $10\,\mathrm{m\,a^{-1}}$ (Fig. A4). Using $N_G$, the correlation was $-0.402$ for $m = 1/3$ and $-0.304$ for $m = 1$; using $N_O$, the correlation was $-0.528$ for $m = 1/3$ and $-0.0260$ for $m = 1$ (Fig. A4). This strong negative correlation for the $m = 1/3$ case indicates that the basal friction coefficient counteracts the effects of the effective pressure, with high values in regions of low effective pressure and low values in regions of high effective pressure. This suggests that the dependency of the basal shear stress on the effective pressure is too strong when $m = 1/3$. This behaviour is particularly clear in the large area of high basal friction coefficient centered around $2200\,\mathrm{km}$ easting and $-400\,\mathrm{km}$ northing in Fig. A3c, which corresponds to an area of low effective pressure (Fig. 2b). Here, the use of the effective pressure cap limits runaway values of the basal friction coefficient; increasing the cap to reduce this area of high basal friction coefficient would mean using less of the GlaDS data. For the $N_O$ run, the large increase in effective pressure upstream resulted in a strong decrease in the basal friction coefficient upstream (Fig. A3d).*

My second general comment concerns the calculation time required for a more complex hydrological model. At what time and space scale does a more complex model (such as GlaDS) make a significant and crucial difference in glacial dynamics and does it compensate for the additional time of calculations used ?

The space and time scale depends on the types of questions that are being answered. For short time scales (weeks to months), hydrology is likely to be fairly static. For very long time scales (multiple centuries) the hydrology will change but it will take a lot more computational resources to run a fully coupled ice sheet-subglacial hydrology model on these timescales, so if alterations in the the basal boundary conditions are not important to the ice dynamics modeling, then coupled variable hydrology will not be as important.

The area of interest is also an important factor when considering what spatial and temporal

scales are important for ice-hydrology interactions. For example, the dynamics of ice streams in the Siple Coast region are strongly dependent on ice-hydrology interactions, so coupling on shorter timescales (e.g. yearly to centennial timescales) would be important here (see e.g. Bougamont et al., 2015). Ice-hydrology coupling may also be essential for modelling the onset and location of ice streams (Kyrke-Smith et al., 2015). A recent study in Greenland showed that the impact of geothermal heat flow on basal ice temperatures is elevated when ice sheet and subglacial hydrology models are coupled (Smith-Johnsen et al., 2020), although this kind of analysis has not been tested more broadly across Antarctica or Greenland. In Antarctica, ice-hydrology coupled models may be essential in predicting the magnitude of basal melt and where it is injected into ice shelf cavities to accurately predict the impact on ice shelf melt; however, such analysis has not yet been conducted.

The end goal with ice-hydrology coupling is to have either an accurate parameterization of hydrology for ice dynamics modeling in situations that require too much computational time for hydrology modeling, or a coupled setup that can run efficiently. The former is what we're aiming for in this manuscript and the latter is something we're working towards. Because the latter hasn't been achieved yet (i.e. coupling of hydrology and dynamics in Antarctic) it's not yet possible to accurately define what the appropriate temporal and spatial scales would be, both for the Denman-Scott catchment, and more broadly across Antarctica.

It's likely that we have not yet answered well enough the question of when we need ice-hydrology coupling and what impact it makes, which is a strong motivator for further studies of its importance.

My third general comment concerns the figures. If a standardisation does not allow the scales to be the same, it must be stipulated in the text for all figures concerned so that there is no misunderstanding. Also, note that this difference in scale does not allow the same analysis and comparison resolution. Also, it is better to have complete captions (variable symbol-units) and the same than the legend (the text written next to the colorbar). It also is more readable if both limits of the scale are written.

To address this, we have altered sections of our analysis to consider normalised deviations from the mean of the basal friction coefficient across the grounded domain so that the same analysis can be performed for the Budd and Schoof friction laws. This is particularly the case when variances are considered, e.g. line 148 of the original manuscript. We have normalized the basal friction coefficients by their respective means in most updated figures. However,

we keep some of the original – non-normalized – basal friction coefficients where we need to make comparisons, e.g. in comparing $N_G$ with $N_O$ in the Budd friction law where there is a substantial change in mean between the two. Limits of the colorbar have been added for all figures, and where they differ between subplots a note has been added to the caption to mention this. Complete captions have been added for all figures where applicable, though we do not add complete captions to the colorbar labels for figures where it will lead to awkward spacing.

My final general comment concerns how the types of effective pressures are expressed in the text. I think it would be better to define in an equation $N_O$ from the beginning and not to repeat it again in the text. Why choose No and not choose the 'limited version' by Brondex et al, 2017 from the start? I don't quite understand how considering the two brings a lot of added value (especially by comparing figures 6c and 6d). If you decide to keep both, then set a symbol for $N$ Brondex to avoid repetitions in the text. Finally, when we see the large difference in $N$ values between $N_O$ and $N_G$ in Figure 3, it would be good to explain how a single variable can be considered with such different values.

We agree that it would be best to define $N_O$ and $N_G$ once and then reference them as such throughout the text. We chose to look at $N_O$ as opposed to the version from Brondex et al. (2017) because $N_O$ has been used in many modeling studies and we wanted to determine whether this was a reasonable value to use for the effective pressure. We will keep the Brondex et al. (2017) analysis and refer to the Brondex et al. (2017) effective pressure as $N_B$ throughout the text.

Fig. 6c and 6d look the same because of the chosen colormap scales. The upper bound of the Fig. 6c and 6d colormap is where water pressure is less than or equal to zero which holds true for the same areas of $N_O$ and the Brondex et al. (2017) effective pressure. These scales have be changed in the updated figures so that the difference between $N_O$ and the Brondex et al. (2017) effective pressure can be seen.

$N_O$ and $N_G$ have such different values because $N_O$ is not actually an effective pressure. It is the overburden hydraulic potential, but is often called the effective pressure, which is not good practice and something we wish to highlight with this manuscript. $N_O$ and $N_G$ are not the same quantity and that is why they differ so markedly. To clarify in the updated manuscript, at line 240-242 we modify:

*However, the definition for $N_O$ used here will produce high effective pressures in regions of*

*thick ice grounded above sea level...*

*to*

*However, the definition for $N_O$ here is in fact the overburden hydraulic potential, not an effective pressure, and will produce high effective pressures (i.e. low basal water pressures) in regions of thick ice grounded above sea level...*

I therefore propose some small changes in the text or the figure calls to improve clarity and understanding. The main thing is the insistence on the terms << basal >> and << subglacial >> which for me are important to keep throughout the text. Finally, I also propose to elaborate on more technical details with respect to the tools used and the choice of parameters.

We have added the words *subglacial* and *basal* throughout the manuscript where applicable.

Specific comments

L37-L117. Please to specify whether these negative effective pressures are stable, at the steady state, seasonally dependent... Please add information on the stability of this case and also whether it varies over time (depending on the seasons or the tides). It could be interesting to add a sentence mentioning that the presence of zones with negative effective pressure may be persistent and does not lead to instability.

The model reaches near steady state; there are some small temporal changes in water depth in deep pockets on the order of mm per day. There is no seasonality, as there is no surface water input to the subglacial hydrology system and tides weren't considered in the model setup. Over longer timescales there may be changes due to subglacial lake drainage and/or changes in ice geometry, but these were not apparent in our models. To clarify, on lines 90-94 we have changed: *We assume a temperate bed throughout, although regions with zero water input (in the interior or southernmost-regions of the domain) are essentially frozen to the bed. The model is run for 10,000 days, providing outputs including channel size and discharge, distributed system discharge, water depth, and effective pressure.*

*to*

*We assume a temperate bed throughout, although regions with zero water input (in the interior or southernmost-regions of the domain) are not an active part of the hydrological system.*

*The model is run for 10,000 days to near steady state (there are changes in the water sheet thickness in deep pockets on the order of $mm\,day^{-1}$), providing outputs including channel size and discharge, distributed system discharge, water depth, and effective pressure.*

To address low and negative effective pressures persisting at steady state and not causing model instability we add on lines 182-183: *These low and negative effective pressures persist at near steady state and do not cause instability in the GlaDS model.*

L64. Stipulate if the model reaches the steady state after 10,000 days.

See response to previous comment.

L78-L346 : If you don't add information about the ice rigidity calculation, please add a reference.

The order of the rigidity inversion and the basal friction coefficient inversions has been added in Section 2.2.1 as well as a reference to Appendix B has been added to line 114. We will add to section 2.2.1 lines 139-147, the following text:

*We perform the following inversion procedure. First, we invert for the ice rigidity over the floating portion of the domain. Next, we invert for the Budd basal friction coefficient over the grounded portion of the domain, using an ice rigidity on grounded ice specified by the Paterson function from Cuffey and surface temperatures from RACMOv2.3 (van Wessem et al., 2018). After the Budd inversion we invert for the ice rigidity over the entire domain. We next use the basal friction coefficient estimated using the Budd friction law to compute an initial estimate of the basal friction coefficient for the Schoof friction law. We perform inversions for the basal friction coefficients of the Budd and Schoof friction laws with the ice rigidity from the inversion prior, these are the main simulations discussed in the text that follows (it is worth noting that the Budd friction coefficient converges to the result of the initital Budd friction coefficient inversion). We perform a final rigidity inversion over the entire domain. The cost functions to be minimized for each inversion are described in detail in Appendix B.*

Additionally, the cost function to be minimized during the rigidity inversions has been added to appendix B with an explanation of the various component cost functions (absolute velocity misfit, logarithmic velocity misfit, and Tikhonov regularization).

L140. As you explain that low effective pressures were associated with faster flow, give a brief explanation of this case.

The sentence in the original manuscript is not quite correct, we have changed it as follows and leave analysis of the role of hydrology in ice dynamics to the discussion section:

*We find a positive correlation ($r^2 = 0.291$) between $N_G$ and the basal friction coefficient (Fig. 5a) where ice surface speeds are $\geq 10\,m\,a^{-1}$, in the region where ice is more dynamic closer to the grounding line.* From line 193-195. The change in the correlation coefficient is from using $C_{\max} = 0.7$ instead of $C_{\max} = 0.8$.

L173. Use the reference Huybrechts, 1990. Remove Budd and Jensen, 1987.

Budd and Jensen, 1987 has been removed; Huybrechts, 1990 added.

If you use the references, Johnson and Fastook, 2002 and Lebrocq et al., 2009 define the hydraulic potential = 0.

We have removed Johnson and Fastook, 2002 and Lebrocq et al., 2009 which we now see did not actually use $N_O$ and we have added Åkesson et al. (2021); Åkesson et al. (2022); Yu et al. (2018) which do use $N_O$.

L180. Accurately calculating the effective pressure is important improve ice-sheet models. However, effective pressure parametrizations used in Brondex et al., 2017 and Kazmierczak et al., 2022 are simplified for computational purposes and numerical stability, especially when the study is either focused on grounding lines or experiments are done on a continental scale. Please, add the concept of model complexity for the subject under study.

We have changed (lines 244-247)

*Previous studies have investigated alternative parameterizations for the effective pressure in the asbscence of a coupled ice sheet-subglacial hydrology model.*

to

*Previous studies have investigated alternative parameterizations for the effective pressure due to the computational cost or numerical instabilities associated with coupling of an ice-sheet model to a complex subglacial hydrology model. Full coupling between these models is a recent development in the field (Cook et al., 2022).*

L180 : Previous studies have investigated alternative parameterizations for the effective pressure **in the absence of a coupled ice sheet-subglacial hydrology model.** In Kazmierczak et al., 2022., the ice-sheet model is coupled to the simple subglacial water routing from Lebrocq et al., 2009 by the subglacial water depth and by the flux. This sentence should be modified by [. . .]**complex subglacial hydrology model due to the computational time.**

This has been changed, see response to above comment.

L196 : Add a reference on the **alpine-like hydrology system.**

Added reference Iken and Bindschadler (1986).

L220 : Since equation 1 does not include $m$, this sentence is not clear. Then perhaps add the $m$ in equation 1 and mention that $m = 1$.

Equation 1 has been changed to include $m$ and on lines 120-122 we change:

*where $\tau_b$ (Pa) is the basal shear stress, $\alpha$ ($s^{1/2}$ $m^{-1/2}$) is the friction coefficient, $N$ (Pa) is the effective pressure and $u_b$ ($m\,a^{-1}$) is the basal sliding speed*

to

*where $\tau_b$ (Pa) is the basal shear stress, $\alpha$ ($s^{1/2}\,m^{-1/2}$) is the friction coefficient, $N$ (Pa) is the effective pressure, $u_b$ ($m\,a^{-1}$) is the basal sliding speed, and $m = 1$ is a power law exponent taken to be linear as in Åkesson et al. (2021); Åkesson et al. (2022); Baldacchino et al. (2022); Choi et al. (2021); Yu et al. (2018)*

L285-286 : Specify with which formulation of the effective pressure this conclusion is made.

This is true for both $N_G$ and $N_O$. We have changed on lines 369-370:

*That is, using the Schoof friction law, regions of lower effective pressures tend to also have lower simulated basal friction and faster flow – evidence for the controlling role of the hydrological system.*

to

*That is, using the Schoof friction law, regions of lower effective pressures (both $N_G$ and $N_O$) tend to also have lower basal friction coefficients – evidence for the controlling role of the hydrological system.*

L353 : Add a reference for the Paterson function.

In Section 2.2.1 we now include a reference to the Paterson function, as well as an additional paragraph, as per our response to the reviewer's specific comment about lines 78 and 346.

L362 (éq. C2) : Why is $C_{\max}$ (0.8) different to the value used in Brondex et al., 2017 (0.5)?

This is a good question, and there is no clear justification for one value of $C_{\max}$ over another, including the value used in (Brondex et al., 2017). Hence, we test the sensitivity of our inversion results to various values of $C_{\max}$ (we will add this sensitivity analysis as a subsection in an appendix of the revised manuscript). We find that a value of $C_{\max} = 0.7$ yields a basal friction coefficient with the smallest variance and we update the manuscript to use this new value. We will include the following text (lines 463-488; new appendix A5):

*Iken's bound, $C_{max}$, mathematically describes the idea that the bed can support a maximum stress (Iken, 1981; Schoof, 2005; Gagliardini et al., 2007). In the Schoof friction law (Eq. 7), $\tau_b$ cannot exceed $C_{max}N$, where $C_{max}$ represents rheological properties of the till (Brondex et al., 2019) and ranges between 0.17 and 0.84 (Cuffey and Paterson, 2010). To determine an appropriate value of $C_{max}$ we test the effect of using $C_{max} = 0.5, 0.6, 0.7,$ and $0.8$ on the Schoof basal friction coefficient using $N_G$ and $N_O$.*

*We arrive at the same qualitative conclusions as in the main text for all four values of $C_{max}$. In all cases, the variance of the normalized basal friction coefficient is smaller for $N_G$ than*

*for $N_O$, and it is smaller than the variance of the normalized Budd basal friction coefficient. Although the results for $C_{max} = 0.5, 0.6$, and $0.7$ are similar, there is a comparatively large decrease in the mean of the basal friction coefficient and increase in the variance of the normalized basal friction coefficient between $C_{max} = 0.7$ and $C_{max} = 0.8$ for both $N_G$ and $N_O$. For lower values of $C_{max}$, a region of higher basal friction coefficient centered around $2200\,km$ easting and $-400\,km$ northing (Fig. A5a) develops in the $N_G$ simulations. Solving Eq. (7) for $C$ yields*

$$C = \frac{\left( \frac{\tau_b}{|u_b|^{m-1} u_b} \right)^{m/2}}{\left( 1 - \left( \frac{\tau_b}{C_{max}N} \right)^{1/m} \right)^{m/2}}, \tag{A1}$$

*where we see that in the limit where $\tau_b \to C_{max}N$ then $C \to \infty$. The region of high basal friction coefficient for lower $C_{max}$ also has low effective pressure (Fig. 2a), resulting from $\tau_b$ approaching $C_{max}N$ for larger values of $N$ and the basal friction coefficient compensating this. To prevent potentially infinitely increasing values of the basal friction coefficient it is possible to increase the cap on the effective pressure, but again, this would reduce the area over which the GlaDS data is used and the effect of using modeled hydrology will be less impactful.*

*The Schoof friction law is a regularized Coulomb friction law which tends towards a Weertman sliding regime when $\tau_b << C_{max}N$ and a Coulomb sliding regime when $\tau_b >> C_{max}N$. Hence, the value of $C_{max}$ will have an effect on what physical processes are being represented in the Schoof friction law. Figure F2 shows $\tau_d/(C_{max}N)$ for various values of $C_{max}$, where values close to zero correspond to a Weertman sliding regime and values close to one correspond to a Coulomb sliding regime. The choice of $C_{max}$ appears to have little effect on where each of the Weertman and Coulomb sliding regimes occur, with distinct locations between the two for all values of $C_{max}$. This suggests that the choice of $C_{max}$ will not have a significant impact on which physical processes are being represented throughout the domain and does not justify the use of one value of $C_{max}$ over another.*

Figure 1. First time you mention Denman-**Scott** catchment. It is better to mention it in the text beforehand.

Line 18-19 changed to: *In the East Antarctic, Denman Glacier of the Denman-Scott catchment (Fig. 1) has seen some of the fastest grounding line retreat of the last 20 years.*

Figure 4. Could you explain how the very different effective pressures between $N_G$ and $N_O$ do not significantly impact the basal friction coefficient in the Schoof basal sliding law but impact significantly the basal friction coefficient in the Budd basal sliding law.

This is because the domain is largely in a Weertman sliding regime in the Schoof friction law where the effective pressure doesn't play a significant role in the value of the basal friction coefficient. Using the Budd friction law, the effective pressure plays a large role in the basal friction coefficient throughout the entire domain so the differences between $N_G$ and $N_O$ propagate into the basal friction coefficient. To better explain this we have added the following text into the discussion of the Schoof compared to Budd friction coefficients (lines 206-212):

*Unlike the Schoof friction law, the choice of $N_O$ or $N_G$ has a significant impact on the distribution of the basal friction coefficient (Fig. C4) in the Budd friction law. This is because the upstream portions of the catchment fall into a Weertman sliding regime in the Schoof friction law where $\tau_b << C_{max}N$ (Fig. F2) and $C^2 \approx \tau_b/u_b^m$. Here, the choice of effective pressure will have minimal effect on the Schoof friction coefficient. In the Budd friction law $\alpha^2 = \tau_b/(Nu_b)$, meaning that the effective pressure plays an important role in determining the basal friction coefficient throughout the entire catchment, which propagates the large discrepancy between $N_O$ and $N_G$ to the basal friction coefficients obtained from using these various effective pressures.*

Table 1 : I never used GlaDs but as some parameters are different than in Werder et al., 2013, why? Specificity from Antarctica or this specific catchment? How did you obtain these parameters? The ice flow constant is the same for cavities and channel?

The parameter values were obtained through sensitivity testing in the Antarctic tested against geophysical data Dow et al. (2020), and are generally assumed to be applicable for Antarctic glaciers. We have now expanded on the GlaDS methods which justifies the choice of parameters reported in Table 1. We have added the following text (lines 72-76):

*As discussed in Dow (2023), when the system is overconstricted, the pressures are unrealistically high and the model ceases to converge. When the system is underconstricted, the pressures are below overburden for much of the domain. While there is some variation within the range of acceptable pressures, the output we present is the median and therefore is the most appropriate for representing the hydrology pressure in ice sheet dynamics equations. Future*

*work with full coupling of hydrology and ice dynamics can explore sensitivity to different distributed system inputs. Channel conductivity is, similarly, a median value applied in GlaDS in Antarctica (Dow et al., 2020).*

In GlaDS application to the Antarctic, the ice flow constant is the same for cavities and channels because they are of a similar size.

Technical corrections

Abstract

L10 Budd **friction** law

Changed

L10-14 Schoof **friction** law

Changed

Introduction

L19. Mention the dataset of Adusumilli et al., 2020 in the references. Because the data you mention are not in the paper itself.
→ Adusumilli, S.; Fricker, H. A.; Medley, B. C.; Padman, L.; Siegfried, M. R. (2020). Data from: Interannual variations in meltwater input to the Southern Ocean from Antarctic ice shelves. UC San Diego Library Digital Collections. https://doi.org/10.6075/J04Q7SHT

Changed

L20. Mention the dataset of Morlighem et al., 2020 in the references.
→ Morlighem, M. (2020). MEaSUREs BedMachine Antarctica, Version 2 [Data Set]. Boulder, Colorado USA. NASA National Snow and Ice Data Center Distributed Active Archive

Center. https://doi.org/10.5067/E1QL9HFQ7A8M.

Changed

L20. Denman **Glacier**

Changed

L21. << containing 1.5 m of sea level equivalent.>> → source? (rewrite Brancato et al., 2020)

We have changed *1.5m of sea level equivalent* to *1.5 m of sea level equivalent (Brancato et al., 2020)*

L26-27. Replace << Coulomb laws >> by **regularized Coulomb friction** laws.
Because in a Coulomb friction law s.s., the basal shear stress is independent of the basal sliding velocity.

Changed

L27. of the  Budd

Changed

L27 & 29. On the **basal** sliding velocity

Changed

L28. **subglacial** water pressure

Changed

L29. Provide examples of the $<<$ other quantities $>>$ (i.e. rugosity of the bed...) L29. Why unexepected?

By unexpected, we mean that the basal shear stress has a different functional dependency on basal sliding velocity and effective pressure than the basal friction law proposes. To clarify this we have changed (on lines 31-34):

*However, the dependency of the basal shear stress on other quantities, or unexpected dependency on sliding velocity and effective pressure, is implicitly captured in the friction coefficient.*

to

*However, if the basal shear stress does not actually have the functional dependence on the basal sliding velocity and effective pressure proposed in the basal friction law, or if it has a functional dependence on other properties of the bed such as roughness, substrate, or temperature, then this will be implicitly captured in the basal friction coefficient.*

L30. **basal** friction coefficient (2)

Changed

L31. **basal** sliding velocity

Changed

L32. **basal** friction coefficient

Changed

L38. Remove *(i.e. the ice overburden pressure)* or modify it by (i.e. when subglacial water pressure exceeds the ice overburden pressure)

Changed to *(i.e. when subglacial water pressure exceeds the ice overburden pressure).*

L41. **Subglacial** water pressure

Changed

Methods.

2.1 GlaDS Setup

L58. Source for the surface velocities? MEaSUREs v2?

These are not actually surface velocities. Instead they are basal sliding velocities from the ISMIP6 experiments (see Seroussi et al., 2019 reference on line 83). The word *surface* has been changed to *basal sliding* in line 80 and we have expanded on the description of how these velocities were obtained in the methods (lines 82-85).

*Basal water and sliding velocity inputs are taken from the JPL_ISSM ISMIP6 Antarctic control run final time step (Seroussi et al., 2019) which was a thermal steady state simulation using the enthalpy formulation implemented in ISSM (Seroussi et al., 2013) and the Blatter-Pattyn (BP) approximation to the full Stokes equations.*

L59. Same comments in L20 about the Morlighem et al., 2020 source

Changed

2.2 ISSM Setup

L68. Add reference for SSA: Morland, L.: Unconfined Ice-Shelf Flow, in: Dynamics of the West Antarctica Ice Sheet, edited by: van der Veen, C. J. and Oerlemans, J., Kluwer Acad., Dordrecht, Netherlands, 99–116, 1987.

Reference added

L72. Same remark in Figure 1.

Changed: *Denman-Scott catchment* now appears before Figure 1 on line 19.

L75. **basal** friction

Changed

L76. **basal** friction law

Changed

L77. Same comment in L59.

Changed

2.2.1 Solving for basal friction coefficients

L85. Why eq 2 and eq C2 are not the same? If it's a mistake, the $()^m$ is missing in the denominator. Is it possible to isolate the coefficients of friction in order to make them more visible?

Thanks for catching this. Eq 2 is correct, Eq C2 has been changed to match. The basal friction coefficients have been bolded in the equations and we have clarified that the equations are scalar.

L87. Source of the Iken's bound missing and please refer to Appendix C.

Added reference *(Appendix A5; Iken, 1981),* the appendices have been rearranged and A5 addresses the impact of Iken's bound on the inversions.

L90. **Appendix C,** Table C1

Changed to *Appendix B, Table B1.*

L98. For the No equation complete that B takes negative values below sea level and defines the variables.

We have changed the paragraph from lines 148-154:
*We compare the difference in the friction coefficients when we use two different prescriptions for the effective pressure: (1) an effective pressure given by assuming water pressure equals the ice overburden pressure plus the gravitational potential energy of the water $N = \rho_i g H + \rho_w g B$, which we refer to as $N_O$; and (2) the effective pressure taken directly from the GlaDS simulations, which we refer to as $N_G$. We cap the effective pressure at 1% of ice overburden pressure for Budd runs and 0.4% of ice overburden pressure for Schoof runs, due to numerical artefacting that arises for values smaller than this. The impact of these choice of caps is discussed in Appendix D.*
to:
*We compare the difference in the friction coefficients when we use two different prescriptions for the effective pressure. (1) An effective pressure given by assuming water pressure equals the ice overburden pressure plus the gravitational potential energy of the water $N_O = \rho_i g H + \rho_w g B$. Here $\rho_i$ is the density of ice $\rho_w$, is the density of water, g is the absolute value of the gravitational acceleration, H is the ice thickness, and B is the bed elevation which takes negative values below sea level. (2) The effective pressure taken directly from the GlaDS simulations, which we refer to as $N_G$. We cap the effective pressure at 1% of ice overburden pressure for Budd runs and 0.4% of ice overburden pressure for Schoof runs, due to numerical artefacting that arises for values smaller than this. The impact of these choice of caps is discussed in Appendix A3.*

Results 3.1 Subglacial hydrology

L105 and L106. Even if it's clearly written in L103, mention that the data indicated (length of the channel and the flow) come from the GlaDs modelling.

*We have changed on lines 161-162:* *Major subglacial hydrology channels form in the Denman-Scott catchment, with significant discharge through both the Denman (Fig. 2a (i)) and Scott Glaciers (Fig. 2a (ii)).* *to*
*The GlaDS modeling indicates that major subglacial hydrology channels form in the Denman-Scott catchment as seen in Fig. 2a, with significant discharge through both the Denman (Fig. 2a (i)) and Scott Glaciers (Fig. 2a (ii)).*

L108. If the two branches of 80 and 52 km of the Denman channel are figure 2a (iii) and (iv), mention the figure in the text.

*Changed.*

L111 & Fig.2d(v). I know that the choice of the limits of the legend is there to allow a better reading, but I find it strange to mention 25 m of thickness whereas the legend stops at 10 m.

*The upper limit of Fig. 2d has been increased to 25 m.*

L112. We cannot really see on the figure that the strongest flux is toward Denman, so do not mention the figure but rather the data.

*We have changed (lines 166-171):* *There is substantial water amalgamation with a maximum depth of 25 m in a basal depression (Fig. 2d (v)) that feeds the channels of both the Denman and Scott Glaciers, although with the strongest flux towards Denman (Fig. 2a). The bed topography of this basin feature lies at 1900 m below sea level (Fig. 1a), and subglacial water flows upslope by approximately 1200 m to drain downstream. A second 'lake-like' feature feeds the northern branch of the Denman channel and reaches a water depth of $\sim$8 m (Fig. 2d (vi)).* *to*
*There is substantial water convergence with a maximum depth of 25 m in a basal depression (Fig. 2d (v)) that feeds the channels of both the Denman and Scott Glaciers, although with*

*the strongest flux towards the eastern branch of the Denman channel (Fig. 2a (iii)). The bed topography of this basin feature lies at 1900 m below sea level (Fig. 1a), and subglacial water flows upslope by approximately 1200 m to drain downstream. A second 'lake-like' feature feeds the western branch of the Denman channel (Fig. 2a (iv)) and reaches a water depth of ∼8 m (Fig. 2d 125 (vi)).*

L116-117-118-119 **subglacial** water pressure

Changed

L121. Show these zones in the Fig. 2b.

These are already shown in Fig 2d. They have now been referenced in the text as follows on lines 176-178: *Effective pressure in the GlaDS outputs is lowest in the basin feature (Fig. 2d (v)) and the lake-like feature (Fig. 2d (vi)), reaching -0.4 MPa in the former and -0.25 MPa in the latter (Fig. 2b)*

3.2 Ice dynamics and inversion

L127. Schoof friction law **(éq. 2)**

Changed

L132. slow **ice** flow (maybe mention the Fig. 1b with the surface speed)

This sentence was a bit unclear so we have changed lines 189-192:

*The friction coefficients are relatively lower in the Denman and Scott troughs, and alternating high and low "stripes" are evident in the region of slow flow (ice surface speeds $< 50\,m\,a^{-1}$) to the west of the Denman Glacier and south of the Shackleton Ice Shelf (we note that this region was excluded from the GlaDS model; see discussion in Appendix A).*
to
*The basal friction coefficients are relatively lower in the Denman and Scott troughs, and alternating high and low "stripes" are evident in the region west of the Denman Glacier*

*and south of the Shackleton Ice Shelf (Fig. 1b (i)) where ice surface speeds are $< 50\,m\,a^{-1}$ (Fig 1a). We note that this region was excluded from the GlaDS model (see discussion in Appendix A1).*

L133. Locate the Shackleton Ice Shelf in one of the figures (e.g. Figure 1).

The Shackleton Ice Shelf was added into Fig. 1b, as per the response to the above comment.

L135. a space is missing between Fig. and 4c

Space added

L136. **basal** friction coefficient

Changed

L138. faster **ice** flow (**ice** surface speeds [. . . ]

The paragraph on lines 135-141 of the original manuscript was unclear and has been changed to:
*The Schoof basal friction coefficient estimated using $N_G$ is smoother compared with that using $N_O$ (Fig. 4a and Fig. 4d; Appendix B). Here, a smoother basal friction coefficient resulted in lower median differences and root mean square error (RMSE) and higher mean differences between the simulated and observed ice surface speeds for the $N_G$ simulation over the $N_O$ simulation (Table 2; Fig. 4c,f). We find a positive correlation between $N_G$ and the basal friction coefficient (correlation coefficient $r^2 = 0.291$, Fig. 5a) when considering areas where ice surface speeds are $\geq 10\,m\,a^{-1}$, in the region where ice is more dynamic closer to the grounding line. Similarly, there is a positive correlation between $N_O$ and the basal friction coefficient ($r^2 = 0.281$, Fig. 5b).*

L139. slow **ice** flow (**ice** surface speeds [. . . ]

L141. **basal** friction coefficient and please refer to (Fig. 5b).

L133-138-140. The limits given to consider an ice flow faster or slower are not the same, why?

On line 133 of the original manuscript, we used ice surface speeds to describe a specific region of the domain but on lines 138 and 140 we were referring to ice surface speeds throughout the whole domain. The words *fast* and *slow* have been taken out of these sections to avoid confusion.

L143. Glacier**s**

We are referring to the troughs so the *s* should be at the end of *troughs* instead of *Glaciers*. We now have: *Denman and Scott Glacier troughs*.

Discussion

4.2 Effective pressure and basal sliding laws

L182. : the till parametrization used in Kazmierczak et al. 2022 is from Bueler and van Pelt, 2015

The reference has been changed

L200. : The relationship between low effective pressures and low basal friction did not hold for the Budd friction law, **(Fig 3 (a), Fig 4 (e))** despite a stronger negative correlation between the friction coefficient and surface speed for the Budd friction law compared to the Schoof **friction** law **(Fig 5(a),(c))**.

Changed

L214. : when using a Weertman friction law**, which not considered the strong dependence of $\tau_b$ on the effective pressure,**

We have changed this to *when using a Weertman friction law, which does not consider the strong dependence of $\tau_b$ on the effective pressure,*

L215. : Schoof **friction** law

Changed

L216-218-221. : **Regularized** Coulomb friction laws

Changed

L229. : This study by Kazmierczak et al. (2022) examined the impact of different representations of the effective pressure – approximated in turn by height above buoyancy (van der Veen, 1987; **Huybrechts, 1990**, Winkelmann et al., 2011, **Martin et al, 2011**), reduced by the subglacial water pressure **(modified from Bueler and Brown, 2009)** or sliding related to water flux (Goeller et al., 2013) **from a simple subglacial water routing model (Lebrocq et al., 2009)**, and the effective pressure in a till **(Bueler and Van Pelt, 2015)** – on ice mass loss from Antarctica over the 21st Century.

Changed

4.3 Empirical Parametrization

L 254-255. : $\gamma$ or $\delta$ ? use the same symbol

The symbol is now consistently $\gamma$.

L260. : define Brondex et al., 2017 or write something like [...] $N_O$ on all the domain or, following the condition of Brondex et al., 2017, exclusively below sea level [...] Fig. **6a, b, c, d**

We have changed:

*and the Brondex et al. (2017) effective pressure*
to
*and the Brondex et al. (2017) prescribed effective pressure ($N_B$, Section 4.2).*

Note that the equations describing the Brondex et al. (2017) effective pressure now have $N_B$ instead of $N$.

L264. : define the $<<$ saturation term $>>$

We have changed:

*Fig. 6e shows the saturation term as well as the physically equivalent scatter from the GlaDS output data.*
to
*Fig. 6e shows the saturation term ($g_x(H)$) as well as the physically equivalent scatter from the GlaDS output data, $(1 - N_G/p_i - r_l)/(1 - r_l)$.*

$g_x(H)$ is now defined in lines 327-332 which now reads: *we suggest a form of the water pressure proportional to the ice overburden pressure multiplied by a term $(r_l + (1 - r_l)g_x(H))$ where $g_x(H) = H^x/(\tilde{H}^x + H^x)$ is a saturation term such that the water pressure reaches a maximum fraction of flotation in areas of high ice thickness. Here, $\tilde{H}$ is defined in Eq. (9) and x is defined in Eq. (10).*

L267. : 77.0% (use the same number of significant digits)

45.4 also has three significant figures so we keep 77.0.

L279. : **basal** friction laws

Changed

L280. : **basal** friction coefficient

Changed

L285. : Schoof **friction** law

Changed

L289. : **geophysical** observations

Changed

Appendix A

L304-308-313. : pressures (it is correct but keep pressure or pressure**s**)

We have changed on line 394 *effective pressures do not exist* to *the effective pressure does not exist.*

L305. : domains

Changed

L306. : speeds

Changed

L307 + Table A1 : $500 \geq v < 1000 \, \mathrm{m\,a^{-1}}$. The sign is not correct ?

Yes this is not correct, thank you for catching this, the sign has been flipped.

L320-321. : (e.g. Bueler and Brown, 2009; Bueler and Van Pelt, 2015, van der wel et al., 2013, Huybrechts 1990, Kazmierczak et al., 2022 ; ; ; van der Veen, 1987; Winkelmann et al., 2011)

Changed

Figure A1. (a)-(b) Effective  N (MPa) Note that the color bar scale is not the same. It is better if both limits of the scale are written.

Changed

Appendix B

L324-332-333-334-335-336. : **basal** friction coefficient

Changed

L327. : For Rignot, 2017, the dataset I founded has to be referenced like that : → Rignot, E., J. Mouginot, and B. Scheuchl. (2017). MEaSUREs InSAR-Based Antarctica Ice Velocity Map, Version 2 [Data Set]. Boulder, Colorado USA. NASA National Snow and Ice Data Center Distributed Active Archive Center. https://doi.org/10.5067/D7GK8F5J8M8R. Date Accessed 04-20-2023.

Changed

And following the informations given to this link : https://nsidc.org/data/nsidc0484/versions/2 these references has to be included :

→ Mouginot, J., B. Scheuchl, and E. Rignot. 2012. Mapping of Ice Motion in Antarctica Using Synthetic-Aperture Radar Data. Remote Sensing. 4. **DOI**: 10.3390/rs4092753. → Mouginot, J. et al. 2017. Comprehensive Annual Ice Sheet Velocity Mapping Using Landsat-8, Sentinel-1, and RADARSAT-2 Data.. Remote Sensing (in press).

Changed

L336. : the precribed effective pressure B1? It is not clear... explain in the text (like in the Table B1. Caption) that the prescribed effective pressure is $N_O$.

This was in reference to Table B1, however, this table has been removed from the manuscript as the relevant data can be summed seen Fig. B1 and C4 of the original manuscript.

L337. : effective pressure **calculation/representation**

We have changed *effective pressure used* to *form of the effective pressure.*

Table B1. : **basal** friction - Schoof **basal** friction coefficient – Budd **basal** friction coefficient

See response to your comment on line 347.

Figure B1. Schoof **basal** friction coefficient

Changed

Appendix C

L344. : B is the bed elevation (m) **taking negative value below sea level**

This has been cut from the appendices and added to the main text in Section 2.2.1 where it is specified that $B$ takes negative values below sea level.

L348. : same comments in L327

Changed

Figure C1. Budd **basal** friction coefficient – '.' Missing at the end of the sentence.

Changed

Figure C2. Schoof **basal** friction coefficient– '.' Missing at the end of the sentence.

Changed

Figure C3. On the figure : Schoof (x2) + specify with No and NG– In the caption :**Basal** friction coefficients - Schoof **basal** friction coefficient (x2) Budd **basal** friction coefficient (x2). Add the units and that the colorbar scale is not the same. It is better if both limits of the scale are written.

The caption now reads: *Basal Friction coefficients from inversion (note that the units and the colorbar are not the same in each subplot).* **(a)** *Schoof basal friction coefficient (C, $kg^{1/2}\ m^{-2/3}\ s^{-5/6}$) from $N_G$ run;* **(b)** *Schoof basal friction coefficient (C, $kg^{1/2}\ m^{-2/3}\ s^{-5/6}$) from $N_O$ run;* **(c)** *Budd basal friction coefficient ($\alpha$, $s^{1/2}\ m^{-1/2}$) from $N_G$ run; and* **(d)** *Budd basal friction coefficient ($\alpha$, $s^{1/2}\ m^{-1/2}$) from $N_O$ run.*

Figure C4. **basal** friction coefficients – prescribed $\rightarrow$ No?

Changed

Figure C5. It is better if both limits of the scale are written. – Caption : '.' Missing at the

end of the sentence.

The limits have been added and the period has been added as well.

Table C1. Budd **basal** friction coefficient – Schoof **basal** friction coefficient

Changed

L352. : Budd **basal** friction coefficient

Changed and added to section 2.2.1 of the main manuscript.

L358. : **basal** friction coefficient

Changed

L361. : same comment L85

Changed

L362. : Schoof **basal** friction coefficient – Add a reference or an explanation for the Iken's bound – **basal** friction coefficient

This is now in the main text, we have added the reference (Iken, 1981).

L363. : **basal** friction coefficient

Changed, this is now in the main text.

L364. : **basal** friction coefficient

Changed, this is now in the main text.

Appendix D

L374-376. **ice** overburden **pressure**

Changed

Figure D1. It is better if both limits of the scale are written.

We have removed this figure from the manuscript, we believe that the text in this section is sufficient to get the point across.

Add the units and that the colorbar scale is not the same. It is better if the caption and the legend are the same (with symbol and units) → Effective pressure (N) (MPa)/ friction coefficient for the Schoof friction law C [...]/surface speed (us) (m a-1)

See our response to the above comment.

Figure D2. It is better if both limits of the scale are written.

We have removed this figure from the manuscript and summarized the results in the appendix text, as follows on lines 437-440:

*For a cap of 0.4 %, approximately 2 % of the domain has an effective pressure that is linearly proportional to the ice overburden pressure; that is, 98 % of the effective pressure in the ISSM simulation is derived directly from the GlaDS simulated effective pressure. Increasing the cap to 1 % – which is used in the Budd runs – decreases the area over which the GlaDS effective pressure is used to 96 % and increasing the the cap to 4 % decreases the area to 48 %.*

Add the units and that the colorbar scale is not the same. It is better if the caption and the legend are the same (with symbol and units)

See our response to the above comment.

Appendix E

L380. : define $\rho_i$, $g$ and $H$ in the $p_i$ equation

Thank you for the suggestion. These variables have already been defined earlier in the text (the introduction of $N_O$) so we don't redefine them here.

L381. : **subglacial** water pressure

Changed

L394. : to avoid a confusion with the power law exponent, maybe chose another letter than m.

We have changed $m$ to $x$ throughout the manuscript.

L395. : **subglacial** water pressure (x2) –  water pressure **allowed**

Changed

Figures

Figure 1.
- (a) Bed elevation **(unit missing)** from Bechmachine v2 o Detail : it's << Bed elevation >> in the caption and << Bed topography >> on the figure. Use the same formulation.
- (Morlighem et al., 2020) same comment as L20.

- (Rignot 2017) same comment as L327. $\rightarrow$ And modify by $<<$ **Rignot et al., 2017** $>>$
- It misses a '.' At the end of the caption.

The new figure caption now reads: *Denman-Scott catchment.* **(a)** *Bed topography (m) from Bechmachine v2 (Morlighem, 2020);* **(b)** *Ice urface speed (m a$^{-1}$) from MEaSUREs v2 (Rignot et al., 2011, 2017; Mouginot et al., 2012, 2017), using a logarithmic color scale;* **(i)** *The Shackleton Ice Shelf. The black lines in both panels show the ice catchment outline, defined by the drainage divide and calving front, the grounding line is shown in red, and the GlaDS domain in yellow. The x$-$ and y$-$axes are eastings and northings, defined in polar stereographic coordinates referenced to WGS84. These maps were made using the Antarctic Mapping Tools (Greene et al., 2017) and RAMP Radarsat Antarctic Mapping Project (Greene, 2022) toolboxes for MATLAB.*

Figure 2.
- modify overburden pressure by **ice** overburden pressure.
- On L114, you mention $<<$ the nothern branch $>>$ of the Denman glacier, but in the caption you mention in Fig2a (iii) a western branch and in (iv) an eastern branch. I'm confused. Please, is it possible to have a clarification/harmonisation between the figure and the text and also to say which one is longer compared to what is written on L108.

It now reads *ice overburden pressure.* On line 170, *northern branch* has been changed to *western branch.*

Figure 3. Place the figure in the subsection 3.2. (a) Effective pressure (MPa) calculated by GlaDS, $N_G$, [...] (b)  effective pressure (MPa) [...]

The caption now reads:

*Effective pressure inputs.* **(a)** *effective pressure (MPa) from GlaDS, $N_G$, capped at 0.4% of ice overburden pressure for the Schoof friction law shown here;* **(b)** *prescribed effective pressure (MPa) equal to the ice overburden pressure plus the gravitational potential energy of water $N_O = \rho_i gH + \rho_w gB$.*

Figure 4. Caption : **surface** velocity/ Schoof **friction** law (2)/ Budd **friction** law (2)

The caption now reads:

*Ice dynamics outputs (note the colormap limits are not all the same).* **(a)**, **(d)**, **(g)**, **(j)** *are basal friction coefficients (C for (a) and (d), $\alpha$ for (g) and (j)), (b), (e), (h), (k) are basal friction coefficients normalized to their respective means (normalized C for (b) and (e), normalized $\alpha$ for (h) and (k)), (c), (f), (i), (l) are the differences between the simulated and observed ice surface velocities ($m\,a^{-1}$), sim-obs $u_s$.* **(a)**, **(b)**, *and* **(c)** *show outputs from the Schoof friction law with $N_G$;* **(d)**, **(e)**, *and* **(f)** *are from the Schoof friction law with $N_O$;* **(g)**, **(h)**, *and* **(i)** *are from the Budd friction law with $N_G$; and* **(j)**, **(k)**, *and* **(l)** *are from the Budd friction law with $N_O$. In each panel, the black lines are the grounding lines.*

Figure 5. Fig. 5b Schoof It is better if both limits of the scale are written. Fig. 5d for $r^2 =$ write the same number of significant digits. In the caption : The red line is the linear line of best fit, and the slope of this line is reported ($r^2$).

The colorbar limits are now shown, all correlation coefficients are to three significant digits in the plot and in the text. The caption now reads:

*Relationship between the effective pressure and basal friction coefficient for:* **(a)** *Schoof basal friction coefficient C with $N_G$;* **(b)** *Schoof basal friction coefficient C with $N_O$;* **(c)** *Budd basal friction coefficient $\alpha$ with $N_G$; and* **(d)** *Budd basal friction coefficient $\alpha$ with $N_O$. In each panel, points are colored by the natural log of the ice surface speed. The red line is the linear line of best fit, with the correlation coefficient reported ($r^2$).*

Figure 6. a-b-c-d for the colorbar, maybe write $<< N_G/P_i >>$ - $<< N_E/P_i >>$ - $<< N_O/P_i >>$ $<< N_B/P_i >>$? and explain it completely in the caption. Use the same wording in the legend as in the caption. (f) The difference between the proposed empirical parameterization of effective pressure ($N_E$) and the GlaDS effective pressure ($N_G$) as a fraction of ice overburden pressure.

The colorbars have been changed as suggested. The caption now reads:

*Effective pressures as a fraction of ice overburden pressure for:* **(a)** *GlaDS output effective pressure $N_G/p_i$,* **(b)** *empirical parameterization of effective pressure $N_E/p_i$,* **(c)** *typically prescribed effective pressure $N_O/p_i$, and* **(d)** *Brondex et al. (2017) prescribed effective*

*pressure $N_B/p_i$.* **(e)** *The difference between the empirical parameterization of effective pressure and the GlaDS output effective epressure as a fraction of ice overburden pressure (($N_E - N_G)/p_i$).* **(f)** *The saturation curve ($g_x(H)$) and the physically equivalent scatter for $N_G$ (($1 - N_G/p_i - r_l)/(1 - r_l)$). Black lines in* **(c)** *and* **(d)** *represent the $N/p_i = 1$ contour.*

As figure f is explained before firgure e in the text, I would switch them.

They have been switched.

Table 1.

Channel conductivity

Changed

Ice density $\mathrm{kg\,m^{-3}}$

Changed

Sheet width below channel

Changed

Table 2.

In the caption, use the same formulation than in the fig. 3 caption : [. . .] to **the ice** overbruden pressure [. . .] **formula**

Changed

**References**

H. Åkesson, M. Morlighem, M. O'Regan, and M. Jakobsson. Future Projections of Petermann Glacier Under Ocean Warming Depend Strongly on Friction Law. *Journal of Geophysical Research: Earth Surface*, 126(6):e2020JF005921, 2021. ISSN 2169-9011. 10.1029/2020JF005921.

F. Baldacchino, M. Morlighem, N. R. Golledge, H. Horgan, and A. Malyarenko. Sensitivity of the Ross Ice Shelf to environmental and glaciological controls. *The Cryosphere*, 16(9): 3723–3738, 2022. ISSN 1994-0424.

M. Bougamont, P. Christoffersen, S. F. Price, H. A. Fricker, S. Tulaczyk, and S. P. Carter. Reactivation of Kamb Ice Stream tributaries triggers century-scale reorganization of Siple Coast ice flow in West Antarctica. *Geophysical Research Letters*, 42(20):8471–8480, 2015. ISSN 0094-8276.

J. Brondex, O. Gagliardini, F. Gillet-Chaulet, and G. Durand. Sensitivity of Grounding Line Dynamics to the Choice of the Friction Law. *Journal of Glaciology*, 63(241):854–866, 2017. 10.1017/jog.2017.51.

J. Brondex, F. Gillet-Chaulet, and O. Gagliardini. Sensitivity of Centennial Mass Loss Projections of the Amundsen Basin to the Friction Law. *The Cryosphere*, 13(1):177–195, 2019.

Y. Choi, M. Morlighem, E. Rignot, and M. Wood. Ice dynamics will remain a primary driver of Greenland ice sheet mass loss over the next century. *Nature Communications Earth Environment*, 2(1), 2021. ISSN 2662-4435.

S. J. Cook, P. Christoffersen, and J. Todd. A Fully-Coupled 3D Model of a Large Greenlandic Outlet Glacier with Evolving Subglacial Hydrology, Frontal Plume Melting and Calving. *Journal of Glaciology*, 68(269):486–502, June 2022. ISSN 0022-1430, 1727-5652. 10.1017/jog.2021.109.

K. Cuffey. Burlington, MA.

C. Dow, M. Werder, G. Babonis, S. Nowicki, R. Walker, B. Csatho, M. Morlighem, C. F. Dow, M. A. Werder, G. Babonis, S. Nowicki, R. T. Walker, B. Csatho, and M. Morlighem. Dynamics of Active Subglacial Lakes in Recovery Ice Stream. *Journal of Geophysical Research*, 123(4):837–850, 2018. 10.1002/2017JF004409.

C. Dow, F. McCormack, D. Young, J. Greenbaum, J. Roberts, and D. Blankenship.

Totten Glacier subglacial hydrology determined from geophysics and modeling. *Earth and Planetary Science Letters*, 531:115961–, 2020. ISSN 0012-821X.

C. F. Dow. The role of subglacial hydrology in Antarctic ice sheet dynamics and stability: a modelling perspective. *Annals of Glaciology*, pages 1–6, 2023. ISSN 0260-3055.

O. Gagliardini, D. Cohen, P. Raback, and T. Zwinger. Finite-Element Modeling of Subglacial Cavities and Related Friction Law. *Journal of Geophysical Research - Earth Surface*, 112 (F2):1–11, May 2007. ISSN 0148-0227. 10.1029/2006JF000576.

C. A. Greene. RAMP Radarsat Antarctic Mapping Project, 2022. URL https://www.mathworks.com/matlabcentral/fileexchange/52031-ramp-radarsat-antarctic-mapping-project.

C. A. Greene, D. E. Gwyther, and D. D. Blankenship. Antarctic Mapping Tools for Matlab. *Computers & Geosciences*, 104:151–157, jul 2017. 10.1016/j.cageo.2016.08.003. URL https://doi.org/10.1016\%2Fj.cageo.2016.08.003.

A. Iken. The Effect of the Subglacial Water Pressure on the Sliding Velocity of a Glacier in an Idealized Numerical Model. *Journal of Glaciology*, 27(97):407–421, 1981. ISSN 0022-1430.

A. Iken and R. A. Bindschadler. Combined Measurments of Subglacial Water-Pressure and Surface Velocity of Findelengletsher, Switzerland. *Journal of Glaciology*, 32(110):101–119, 1986. ISSN 0022-1430.

E. Kazmierczak, S. Sun, V. Coulon, and F. Pattyn. Subglacial Hydrology Modulates Basal Sliding Response of the Antarctic Ice Sheet to Climate Forcing. Preprint, Ice sheets/Numerical Modelling, Mar. 2022.

T. M. Kyrke-Smith, R. F. Katz, and A. C. Fowler. Subglacial hydrology as a control on emergence,scale, and spacing of ice streams. *Journal of Geophysical Research. Earth surface*, 120(8):1501–1514, 2015. ISSN 2169-9003.

D. R. MacAyeal. A tutorial on the use of control methods in ice-sheet modeling. *Journal of Glaciology*, 39(131):91–98, 1993. ISSN 0022-1430.

A. Maritati, A. R. A. Aitken, D. A. Young, J. L. Roberts, D. D. Blankenship, and M. J. Siegert. The tectonic development and erosion of the Knox Subglacial Sedimentary Basin, East Antarctica. *Geophysical Research Letters*, 43(20):10,728–10,737, 2016. ISSN 0094-8276.

F. S. McCormack, R. C. Warner, H. Seroussi, C. F. Dow, J. L. Roberts, and A. Treverrow.

Modeling the Deformation Regime of Thwaites Glacier, West Antarctica, Using a Simple Flow Relation for Ice Anisotropy (ESTAR), volume = 127, year = 2022,. *Journal of Geophysical Research. Earth surface*, (3). ISSN 2169-9003.

M. Morlighem, E. Rignot, H. Seroussi, E. Larour, H. Ben Dhia, and D. Aubry. Spatial patterns of basal drag inferred using control methods from a full-Stokes and simpler models for Pine Island Glacier, West Antarctica. *Geophysical Research Letters*, 37(14): L14502–n/a, 2010. ISSN 0094-8276.

J. Mouginot, B. Scheuchl, and E. Rignot. Mapping of Ice Motion in Antarctica Using Synthetic-Aperture Radar Data. *Remote sensing*, 4(9):2753–2767, 2012. ISSN 2072-4292.

J. Mouginot, E. Rignot, B. Scheuchl, and R. Millan. Comprehensive Annual Ice Sheet Velocity Mapping Using Landsat-8, Sentinel-1, and RADARSAT-2 Data. *Remote sensing*, 9(4):364–, 2017. ISSN 2072-4292.

K. Poinar, C. F. Dow, and L. C. Andrews. Long-Term Support of an Active Subglacial Hydrologic System in Southeast Greenland by Firn Aquifers. *Geophysical Research Letters*, 46(9):4772–4781, 2019. ISSN 0094-8276.

E. Rignot, J. Mouginot, and B. Scheuchl. Ice Flow of the Antarctic Ice Sheet. *Science*, 333 (6048):1427–1430, Sept. 2011. ISSN 0036-8075. 10.1126/science.1208336.

E. Rignot, J. Mouginot, and B. Scheuchl. MEaSUREs InSAR-Based Antarctica Ice Velocity Map, Version 2 [Data Set]. Boulder, Colorado USA. NASA National Snow and Ice Data Center Distributed Active Archive Center., 2017. Accessed 06-10-2021.

C. Schoof. The Effect of Cavitation on Glacier Sliding. *Proceedings of the Royal Society A*, 461(2055):609–627, 2005. ISSN 1364-5021. 10.1098/rspa.2004.1350.

H. Seroussi, M. Morlighem, E. Rignot, A. Khazendar, E. Larour, and J. Mouginot. Dependence of century-scale projections of the Greenland ice sheet on its thermal regime. *Journal of Glaciology*, 59(218):1024–1034, 2013. ISSN 0022-1430.

H. Seroussi, S. Nowicki, E. Simon, A. Abe-Ouchi, T. Albrecht, J. Brondex, S. Cornford, C. Dumas, F. Gillet-Chaulet, H. Goelzer, N. R. Golledge, J. M. Gregory, R. Greve, M. J. Hoffman, A. Humbert, P. Huybrechts, T. Kleiner, E. Larour, G. Leguy, W. H. Lipscomb, D. Lowry, M. Mengel, M. Morlighem, F. Pattyn, A. J. Payne, D. Pollard, S. F. Price, A. Quiquet, T. J. Reerink, R. Reese, C. B. Rodehacke, N.-J. Schlegel, A. Shepherd, S. Sun, J. Sutter, J. Van Breedam, R. S. W. van de Wal, R. Winkelmann, and

T. Zhang. initMIP-Antarctica: An ice sheet model initialization experiment of ISMIP6. *The Cryosphere*, 13(5):1441–1471, May 2019. ISSN 1994-0416. 10.5194/tc-13-1441-2019.

S. Smith-Johnsen, N. Schlegel, B. Fleurian, and K. H. Nisancioglu. Sensitivity of the Northeast Greenland Ice Stream to Geothermal Heat. *Journal of Geophysical Research. Earth surface*, 125(1), 2020. ISSN 2169-9003.

J. M. van Wessem, W. J. Van De Berg, B. P. Y. Noël, E. Van Meijgaard, C. Amory, G. Birnbaum, C. L. Jakobs, K. Krüger, J. Lenaerts, S. Lhermitte, S. R. M. Ligtenberg, B. Medley, C. H. Reijmer, K. van Tricht, L. D. Trusel, L. H. van Ulft, B. Wouters, J. Wuite, and M. R. van den Broeke. Modelling the Climate and Surface Mass Balance of Polar Ice Sheets Using RACMO2: Part 2: Antarctica (1979-2016). *The Cryosphere*, 12(4): 1479–1498, 2018. 10.5194/tc-12-1479-2018.

H. Yu, E. Rignot, H. Seroussi, and M. Morlighem. Retreat of Thwaites Glacier, West Antarctica, over the next 100 Years Using Various Ice Flow Models, Ice Shelf Melt Scenarios and Basal Friction Laws. *The Cryosphere*, 12(12):3861–3876, 2018.

C. Zhao, R. M. Gladstone, R. C. Warner, M. A. King, T. Zwinger, and M. Morlighem. Basal friction of Fleming Glacier, Antarctica – Part 1: Sensitivity of inversion to temperature and bedrock uncertainty. *The Cryosphere*, 12(8):2637–2652, 2018. ISSN 1994-0424.

H. Åkesson, M. Morlighem, J. Nilsson, C. Stranne, and M. Jakobsson. Petermann ice shelf may not recover after a future breakup. *Nature Communications*, 13(1):2519–2519, 2022. ISSN 2041-1723.

---

## Author Comment (AC3)

**Reviewer Comment Response**

Dear Dr Karlsson, Dr Zwinger

We thank the reviewers for their constructive comments and the improvements that they will bring to the manuscript.

In what follows, the reviewer comments are in black, our responses to the reviewer comments are in blue, and suggested edits to the manuscript are italicised. The line references are to the revised manuscript unless otherwise specified.

In addition to the changes suggested by the reviewers, we have also condensed the appendices for improved readability, by merging all appendices that relate to sensitivity analyses into subsections of Appendix A. Appendix B gives more details on the inversion procedure, and Appendix C gives more details on the new, proposed empirical parameterisation for the effective pressure.

Best regards,
Koi McArthur and co-authors.

**1 Author Comments**

Review of: Basal conditions of Denman Glacier from glacier hydrology and ice dynamics modeling

May 11, 2023
**1 General impression**

The article presents the incorporation of a previously determined effective pressure in the inversion of basal friction coefficients for a shallow-shelf ice flow model of the Denman-Scott catchment for two sliding laws, namely a Budd-sliding law with linear coefficient and a regularised Coulomb law, as presented by Schoof (2005). This is a timely topic and in general I see it suited to be published in The Cryosphere. I have three major points I would see necessary to be addressed before the publication can proceed - I put them in a separate section below.

The article in general is concisely written. The majority of figures is good to read and

conveys the information well. On top of the major points, there are a few questions and suggestions I placed in my review. I hope that these may contribute to improve the quality of the manuscript.

Thank you for your comments on our paper, which we address below.

**2 Major points to be addressed**

The first item I would see to be addressed is **a more detailed description of the inputs to the GlaDS simulation**, in terms of parameters but mainly the imposed slip velocity and melt-water production. You seem to run GlaDS as a pre-processing step to produce $N_G(x)$ for the inversions of the specific friction coefficients. Yet, the hydrology computations needs input in form of a slip velocity and a water-production that themselves will be a result of the ice-flow dynamics and hence the friction coefficients applied in the ice-flow model providing those. To me this appears to be a little bit of a cat-catches-its-tail problem. From the text (line 59): *Basal water and sliding velocity inputs are computed from ISSM (Seroussi et al., 2019)*, I would conclude that you pick initial sliding and velocities from a completely different inversion, subject to certain constraints: *The sliding velocity acts to open up distributed system cavities and, at velocities greater than $800\,m\,a^{-1}$ , can cause model instabilities and so is capped at this value.* Can you please clearly state what ice-flow setup you base the GlaDS computation on? What are the approximations to the Stokes equation of this initial model? What has been used to represent the sliding-law and the applied effective viscosity therein - the latter also in terms of thermodynamics (if any) or damage? How do you deduce the water production from that result? If all this is addressed, I would also hope to see some conclusion if and if so, how this initial settings could have influence on the distribution of the effective pressure arising from GlaDS and if there further might be a possibility that they could pre-condition the result of the following inversion. For the pasteurisation in GlaDS itself, I have difficulties in lack of any equations and symbols to interpret values displayed in Table 1 (see detailed comments).

As with all inversions and model runs, we start with our best estimate and use this to iterate towards a reasonable result. In terms of the GlaDS setup, we use standard basal velocity and water input from the JPL_ISSM ISMIP model outputs of a thermal steady-state simulation (Seroussi et al., 2020), using the enthalpy formulation implemented in ISSM (Seroussi et al.,

2013) and the Blatter-Pattyn (BP) approximation to the full Stokes equations. These fields are used to initiate the hydrology model, from which the basal water pressures are used as inputs to the inverse model using ISSM. This is a soft-coupled approach between the ice sheet and hydrology systems – a precurser to full two-waycoupling between GlaDS and ISSM that we are actively working towards. Even with two-way coupling we require input fields from the ice sheet system to initialise the hydrology model. Unfortunately without two-way coupling available at the moment we are unable to run the hydrology and ice dynamics forward in time together; we do hope, however, that this study demonstrates why that is an important next step. The 800 m/year cap is a standard approach to GlaDs modeling where it is necessary to apply a limit to subglacial cavity opening for stability. This is a limitation of the model but in tests for Greenland (Poinar et al, 2019) had little impact on the basal water pressure. The ISMIP model setup is well-described, widely used and cited, and the data sets are freely accessible through GHub. Hence, we refer readers to the references that detail the setup rather than repeating them here. We now include the added detail on the GlaDS inputs of basal water and sliding velocity (at lines 82 to 90):

*Basal water and sliding velocity inputs are taken from the JPL_ISSM ISMIP6 Antarctic control run final time step (Seroussi et al., 2019) which was a thermal steady state simulation using the enthalpy formulation implemented in ISSM (Seroussi et al., 2013) and the Blatter-Pattyn (BP) approximation to the full Stokes equations. This model solved for ice viscosity of the floating ice shelf and the basal friction coefficient using data assimilation techniques (MacAyeal, 1993; Morlighem et al., 2010), which differs from our application of ISSM to assess basal boundary conditions. However, lacking an alternative starting point for basal sliding and velocity, this is the best available option prior to full coupling between hydrology and ice dynamics in ISSM. The sliding velocity acts to open up distributed system cavities and, at velocities greater than $800\,m\,a^{-1}$, can cause model instabilities and so is capped at this value. Tests of similar caps for model runs at Helheim Glacier (Poinar et al., 2019) demonstrate it has little impact on the model results.*

The second topic I would ask to have elaborated is a **discussion on how the approximations to the Stokes equations could influence your inverted slip coefficients in regions of significant vertical shear**. From what I read in the text, I understand that you are applying the Shallow Shelf Approximation (SSA). To my understanding, the dynamics in fast flowing outlet parts will be well represented by SSA. Of my concern are rather those regions, where the onset of the outlets takes place, where I would expect internal vertical ice deformation to still play a significant role. Ignoring this component, in my view, would have

a bias to over-predict the slip. This highly also links to the missing detailed information on which input the GlaDS simulations are based on. If this is also based on SSA, altered slip can bias the hydrological system (as it alters melt-water production and slip-induced opening rates) over a wide range of the catchment area.

This is a good point. Even with the Blatter-Pattyn approximation, the unenhanced Glen flow relation cannot, by nature, capture the bed-parallel vertical shear deformation profile expected in regions where the ice sheet is frozen to the bed (McCormack et al., 2022). We expect that in our simulations using SSA and the Glen flow relation, that our model overestimates flow by sliding, and underestimates deformation, most likely compensating by making the ice too stiff. However, in this work we're primarily interested in the differences between different basal friction laws, so it's likely that this is an issue inherent in each of our simulations, and it is unlikely to change our main conclusions. If we were to perform a prognostic simulation, we would certainly want to consider the relative contributions of deformation and sliding, and more appropriately treat the processes of deformation that occur in tertiary creep, e.g. anisotropy.

To clarify this we add lines 290-299:

*In this work we have used an SSA ice flow model, which fails to capture bed-parallel vertical shear deformations. This may affect the results of our inversions for the basal friction coefficient in areas of non-negligible vertical shear, such as at the onset of fast-flowing ice streams of the Denman and Scott troughs. However, the use of the Glen flow relation may also impact the capacity of even higher-order models to accurately capture bed-parallel vertical shear deformations. For example, McCormack et al. showed that even when the BP approximation to the full Stokes is used, the unenhanced Glen flow relation fails to capture the vertical shear profile expected in regions where ice is frozen to the bed of the glacier. In our simulations that use the SSA approximation and the Glen flow relation, it is possible that sliding is overestimated, and the basal friction coefficient underestimated, where vertical shear is an important deformation process. However, this is a common issue to all of our model runs and is therefore unlikely to alter the main conclusions of this work that compare how the form of the effective pressure impacts the basal friction coefficient.*

We explain in more detail the GlaDS setup in our response to your first major comment, noting that the thermal model there employs the BP approximation, and hence the inputs to the GlaDS simulations may take into account some bed-parallel vertical shear deformation. However, as argued above, in the absence of a flow relation that incorporates effects of

deformation that are present in tertiary creep, even the BP approximation may not accurately capture the contribution of bed-parallel vertical shear deformation to overall flow. Although it's outside the scope of what we focus on in this work, the relative contributions of sliding and deformation to overall flow is an important question generally, and should definitely be considered in future work.

Finally, I would also like to **better understand the whole inversion procedure and the impact of the rigidity inversion on your results**. In my view this is best achieved by presenting the inversion procedure in terms of equations. You shortly mention that you invert for rigidity of the ice (line 77), which could be somehow interpreted as inverting for a (depth averaged) damage or temperature field. You are presenting a result of this rigidity distribution in Appendix C (which is hard to interpret) but – in my view equally important – do not provide some kind of physical interpretation of it. Also, starting the averaged viscosity based on the (atmospheric?) temperature distribution given by RACMO (line 353), in my view would need some explanation. How much, for instance, does the inverted rigidity differ from the one used to compute the bedrock velocity used in GlaDS simulations? In my opinion the reader already would benefit if you could introduce the equations of the SSA system where rigidity is plugged in (to my understanding how you define your averaged viscosity in the SSA model)

The rigidity field is intended to capture processes related to the rheology of ice that are not explicitly modelled or parameterised in the flow relation. These may include effects such as: damage, anisotropy, chemical impurities, and liquid water. These ice properties or processes evolve spatially and in time, for example due to changes in the stress, temperature, and climate forcings, and such temporal evolution of ice rheology is not typically accounted for in the ice rigidity field (although this is not an issue for our diagnostic model simulations). In our study, uncertainties in the rheological parameters do add to uncertainty in the model results. Some of these uncertainties are discussed/summarised in recent papers, including Graham et al., (2018) and McCormack et al. (2022). Although it is outside the scope of the current study to consider the effects of ice rheology, we have elaborated on these uncertainties in Section 2.2.1 line 136-138:

*Though it is not the primary focus of this work, we invert for ice rigidity as well, while initializing our model for the inversion for the basal friction coefficients. By inverting for the ice rigidity we capture ice rheological processes which are not explicitly accounted for in the model such as damage, anisotropy, chemical impurities, and liquid water.*

There will be differences in the rigidities generated here using ISSM and those used for the GlaDS initialisation; however, since in the GlaDS simulation we relied on standard ISSM ISMIP6 outputs we are unable to change this. These issues in terms of the subglacial hydrology vs. ice dynamics setup in ISSM are excellent questions to address once we have a two-way coupled model that we can apply initialisation experiments to.

Using surface temperatures from RACMOv2.3 as opposed to depth averaged temperatures could impact our basal friction coefficient inversions. Zhao et al. (2018) found that initializing the temperature field to be colder resulted in a lower basal friction coefficient. Using a depth averaged temperature which may be warmer than the surface temperature could increase the value of the basal friction coefficient calculated from inversion, though this is likely an issue present in all of our model runs and hence shouldn't affect our conclusions regarding the impact of the effective pressure on the basal friction coefficient inversion. We add lines 300-307 to discuss the impact of using RACMOv2.3:

*Initializing our temperature field with surface temperatures from RACMOv2.3 (van Wessem et al., 2018) could have an impact on the rigidity and basal friction coefficient inversions we performed in this work. Zhao et al. (2018) showed that initializing a model with a colder temperature field resulted in a decrease of the basal friction coefficient computed from inversion, due to stiffer ice. It is possible that if we used a thermal model and computed depth averaged temperatures to use in our model initialization – effectively increasing the initialization temperatures – that we could similarly see an increase in the basal friction coefficients. Like with the SSA approximation, these are issues that would be common in all of our model runs and would be unlikely to alter the main conclusions we came to regarding the effect of the form of the effective pressure on the computation of the basal friction coefficient.*

**3 Detailed comments**

Listed in order of their appearance. If some of the comments link to the main points, I indicate it - else, they are mainly meant as suggestions on how to improve readability of the manuscript or corrections to typos. I did not sync anything with the already published other review - so, sorry for cross-postings.

line 26 *One such parameter is the basal friction coefficient, which is a key component of friction laws including the Weertman (1957), Budd et al. (1979) and Coulomb laws (Schoof, 2005; Gagliardini et al., 2007).* I would refrain from calling these laws to be Coulomb laws

but use regularised Coulomb laws instead. There are several occurrences of this term in the text

Changed

line 27 Typo: ...*case of the  Budd* ...

Changed

line 32 *Therefore, a friction coefficient that is both smooth – has little local variability – and has limited domain wide trends is desirable.* Is that really the case in all flow situations? Could there not be situations of either a drastic change in the properties of the substrate underneath the glacier and/or the thermodynamic conditions that would also imply a significant change of those coefficients?

The point we had intended to make here is that if temperature, substrate, or other variables play an important role in basal friction then, in the ideal case, we might either directly model them or find parameterizations that could be incorporated into friction laws so that their effects could be removed from the friction coefficient. We have reworded:

*However, the dependency of the basal shear stress on other quantities, or unexpected dependency on sliding velocity and effective pressure, is implicitly captured in the friction coefficient. Therefore, a spatially variable friction coefficient suggests a friction law which either fails to capture the proper functional dependency on sliding velocity and effective pressure, or omits the dependency of the basal shear stress on other quantities. Therefore, a friction coefficient that is both smooth – has little local variability – and has limited domain wide trends is desirable.*
to
*However, if the basal shear stress does not actually have the functional dependence on the basal sliding velocity and effective pressure proposed in the basal friction law, or if it has a functional dependence on other properties of the bed such as roughness, substrate, or temperature, then this will be implicitly captured in the basal friction coefficient. Therefore, a spatially variable basal friction coefficient suggests a friction law which either fails to capture the proper functional dependency on basal sliding velocity and effective pressure, or omits*

*the dependency of the basal shear stress on other quantities. By consequence, a basal friction coefficient that is both smooth – has little local variability – and has limited domain-wide trends is desirable.*

line 59 *Basal water and sliding velocity inputs are computed from ISSM (Seroussi et al., 2019).* This directly links to one of my main points above: As you state yourself that the flow conditions (I presume you mean icedynamics) are of essence, I think you should declare in the main part of the manuscript what ice-dynamic input you used to drive the GlaDS simulations. Are you directly using the results from the cited paper (Seroussi et al., 2019)? Even then, in my opinion, the manuscript would benefit from spelling this out (methods, input data, at which time of this simulation you pick the ice-dynamics input?).

We have addressed these concerns in the response to the major comment.

line 63 *The model is run for 10,000 days, providing outputs including channel size and discharge, distributed system discharge, water depth, and effective pressure.* What was the motivation for 10k days? Was this necessary to reach some steady state? If not, which point in time was chosen for extracting the effective pressure distribution? This links to the main point of critics, namely, the in my opinion missing details for the GlaDS-step.

This point in time was chosen in order to reach near steady-state conditions. It's not fully steady state because the presence of lakes and large channels means that there are always some small adjustments happening in small regions of the domain. However, for our analysis, these small regions of change do not impact our results. We have now clarified this in the text as follows:  *The model is run for 10,000 days in order to reach near steady state (there are changes in the water sheet thickness in deep pockets on the order of $mm\,day^{-1}$), providing outputs including channel size and discharge, distributed system discharge, water depth, and effective pressure.*

Table 1 I appreciate that you report on values of used model constants, which adds to the reproducibility of your experiments. You, though, report them without context to equations (hence also not providing symbols), which to me (and perhaps any reader not using GlaDS) makes them difficult to interpret. This links to the first main point of critics. Also, some background (sentence of motivation or a reference) on the choice of the numerical

value could enhance the understanding of the reader. To pick one example: You report the "Ice flow constant", which I understand to be linked to the rate factor in Glen's flow law (though confused by the sign in the exponent of Pa in units) as $2.5 \times 10^{-25} \text{Pa}^{-3}\text{s}^{-1}$. Provided I interpret it correctly, such value refers (e.g. Greve and Blatter, 2009) to a relative ice-temperature of around -10∘C. What motivates this setting? How does this choice influence the inversion for slip coefficients and rigidity?

We have discussed most of these points in response to the major comment. For example, we now discuss the choice of parameters and include equations to demonstrate how they are incorporated in GlaDS (lines 61-77).

*The model parameters used in our simulations are summarized in Table 1. The bedrock bump height, cavity spacing, and sheet conductivity act together in the distributed system equations to either constrict or open the system to water flow as show in :*

$$w - \tilde{A}h|N|^{n-1}N + \frac{\partial h_e}{\partial t} + \overrightarrow{\nabla} \cdot \overrightarrow{q} = m \tag{1}$$

$$\overrightarrow{q} = -kh^{\alpha}|\overrightarrow{\nabla}\phi|^{\beta-2}\overrightarrow{\nabla}\phi \tag{2}$$

*Where $\tilde{A}$ is the rheological constant of the ice multiplied by an order one factor depending on the cavity geometry, $h$ is the hydrology sheet thickness, $N$ is the effective pressure, $n$ is the exponent in Glen's flow law, $h_e$ is the englacial storage, $t$ is time, $m$ is a prescribed source term, $\overrightarrow{q}$ is the hydrology sheet discharge, $k$ is the sheet conductivity, $\phi$ is the hydraulic potential, and $\alpha = 5/4$ and $\beta = 3/2$ are exponents in the Darcy-Weisbach law describing fully turbulent flow. $w$ is the hydrology sheet opening rate equal to $u_b(h_r - h)/l_r$ when $h < h_r$ and zero otherwise, here $u_b$ is the basal ice speed, $h_r$ is the typical bedrock bump height and $l_r$ is the typical cavity spacing.*

*As discussed in Dow (2023), when the system is overconstricted the pressures are unrealistically high and the model ceases to converge. When the system is underconstricted the pressures are below overburden for much of the domain. While there is some variation within the range of acceptable pressures, the output we present is the median and therefore is the most appropriate for representing the hydrology pressure in ice sheet dynamics equations. Future work with full coupling of hydrology and ice dynamics can explore sensitivity to different distributed system inputs. Channel conductivity is, similarly, a median value applied in GlaDS in Antarctica (Dow et al., 2020). The ice flow constant is set for an average ice column temperature of*

*-10°C.*

We also now include the parameter symbols in the table. In terms of the ice flow constant, this parameter is used differently in GlaDS compared to ISSM. For the former, it is the mechanism by which channels and cavities close (Eq. 1 of the revised manuscript), whereas with ISSM the rigidity field is initialised with a temperature field and then updated as the model is run (Eq. 5 of the updated manuscript). Unfortunately, we do not know the impact of changing the rigidity/rheology on the GlaDS outputs because, to date, coupled complex hydrology and ice dynamics modeling have not been applied in the Antarctic and these questions have not been examined. We hope our future work with full coupling will allow us to examine these questions.

line 66 *ISSM is a finite-element model that uses an* *anisotropic* *mesh to simulate ice dynamics.* Just a suggestion: I understand anisotropy as a local variation depending on direction. I, personally, would rather use the terms "non-uniform" or "adaptive in size".

Changed to *non-uniform*

line 72 *The ISSM mesh is comprised of 66,518 nodes, with anisotropic mesh refinement for faster flowing ice using the MEaSUREs v2 ice surface speed (Rignot et al., 2011; Rignot, 2017).* How are you using the MEaSUREs ice velocity product to refine the mesh?

This is described in Appendix A2. We have changed:

*The ISSM mesh is comprised of 66,518 nodes, with anisotropic mesh refinement for faster flowing ice using the MEaSUREs v2 ice surface speed (Rignot et al., 2011; Rignot, 2017).*
to
*The ISSM mesh is comprised of 66,518 nodes, with non-uniform mesh refinement for faster-flowing ice using the MEaSUREs v2 ice surface speed (Rignot et al., 2011, 2017; Mouginot et al., 2012, 2017), described in Appendix A2.*

line 77 The ice rigidity is calculated using inverse methods. Can you please elaborate? Are you performing a dual-parameter inversion for sliding and rigidity? Or is this another step on top of the previous one? And what effects do you think you cover in the SSA application by inverting for rigidity (vertically averaged temperature, damage)? And what effects you

neglect by the simplified physics of your ice-flow model? Are you using this rigidity only in the final inversions or already in the GlaDS runs and how does this connect to the value given in Table 1? This links to my second point of main critics. To elaborate from my side: Temperature, in particular, is reported to be an important factor in what comes to the quality of inversions using full-stress (a.k.a. full Stokes) models (see, e.g., Zhao et al., 2018), in particular in the regions of onset of the fast-flow outlets, where internal vertical deformation has a significant role (an effect that is not included in the here applied SSA approximation).

The order of inversions is now discussed in this section of the methods. That is, we invert for rigidity over the ice shelf, then for the Budd friction law basal friction coefficient, we invert for rigidity over the entire domain, we invert for the Schoof and Budd friction coefficients. We now include in Section 2.2.1: *We perform the following inversion procedure. First, we invert for the ice rigidity over the floating portion of the domain. Next, we invert for the Budd basal friction coefficient over the grounded portion of the domain, using an ice rigidity on grounded ice specified by the Paterson function from Cuffey and surface temperatures from RACMOv2.3 (van Wessem et al., 2018). After the Budd inversion we invert for the ice rigidity over the entire domain. We next use the basal friction coefficient estimated using the Budd friction law to compute an initial estimate of the basal friction coefficient for the Schoof friction law. We perform inversions for the basal friction coefficients of the Budd and Schoof friction laws with the ice rigidity from the inversion prior, these are the main simulations discussed in the text that follows (it is worth noting that the Budd friction coefficient converges to the result of the initital Budd friction coefficient inversion). We perform a final rigidity inversion over the entire domain. The cost functions to be minimized for each inversion are described in detail in Appendix B.*

We have now included in Appendix C the cost functions which we minimize for each inversion. Namely:

$$\mathcal{J}_a(\overrightarrow{u}) = \iint_s \frac{1}{2} \left( (u_x - u_x^{\text{obs}})^2 + (u_y - u_y^{\text{obs}})^2 \right) dS, \tag{B1}$$

$$\mathcal{J}_l(\overrightarrow{u}) = \iint_s \left( \log \left( \frac{||\overrightarrow{u}|| + \varepsilon}{||\overrightarrow{u}^{\text{obs}}|| + \varepsilon} \right) \right)^2 dS, \tag{B2}$$

$$\mathcal{J}_t(\overrightarrow{u}) = \iint_s \frac{1}{2} ||\overrightarrow{\nabla} k||^2 dS, \tag{B3}$$

$$\mathcal{J}(\overrightarrow{u}) = c_a \mathcal{J}_a + c_l \mathcal{J}_l + c_t \mathcal{J}_t. \tag{B4}$$

*Here, $\mathcal{J}_a$ is the linear velocity misfit cost function, $\mathcal{J}_l$ is the logarithmic velocity misfit cost function, $\mathcal{J}_t$ is the regularization cost function, $\overrightarrow{u}$ is the modeled ice surface velocity, $\overrightarrow{u}^{obs}$ is the observed ice surface velocity, $\varepsilon$ is a small number (around machine precision) acting as the minimum observed velocity in Eq. (B2), S is the two dimensional spatial domain of the model, and k is taken to be $\alpha$ in a Budd inversion, C in a Schoof inversion, and $\bar{B}$ (the rigidity) in a rigidity inversion. The full cost function to be minimized is given in Eq. (B4) where $c_a$, $c_l$, and $c_t$ are the cost function coefficients for $\mathcal{J}_a$, $\mathcal{J}_l$, and $\mathcal{J}_t$ respectively. $c_t$ is referred to as the Tikhonov regularization coefficient.*

We have addressed the physical processes that are captured by inverting for rigidity in our response to the main comment. These rigidity inversions have no relation to the GlaDS modeling and are used to obtain simulated ice surface velocities close to observed ice surface velocities prior to and after inversions for the basal friction coefficients of the Budd and Schoof friction laws.

line 98 Variables $H$ and $B$ are not explained right after their first occurrence. Similar, you lack definition for ice and water density and the absolute value of the acceleration by gravity, $\rho_i$, $\rho_w$ and $g$, respectively. These definitions appear somewhere later in the text. $H$ and $B$ seem to be the thickness and the elevation of the ice-sheet bottom. Please, add definitions of symbols at their first occurrence. For me it would be also of benefit to directly annotate the $N$ with a subscribed 0 ($N_0$) in the formula.

We have changed the formula for $N_O$ as suggested and have now included the following variable definitions: *(1) An effective pressure given by assuming water pressure equals the ice overburden pressure plus the gravitational potential energy of the water $N_O = \rho_i g H + \rho_w g B$. Here, $\rho_i$ is the density of ice $\rho_w$, is the density of water, g is the absolute value of the gravitational acceleration, H is the ice thickness, and B is the bed elevation which takes negative values below sea level.*

Figure 2 These are suggestions: Perhaps some elaborated color-map to distinctively highlight the excess to flotation in (c) and/or the 100% as iso-line would in my opinion enhance the information of this figure.

The 100 % isoline has been added to Fig. 2c.

Figure 4 Please, add an explanation what the black line in the right column represents. Same situation also in Figure D1 – Figure A1 contains the correct description in its caption.

Changed

line 147 *This leads to a comparatively greater standard deviation in the Budd friction coefficient compared with that of Schoof (1240 kg$^{1/2}$ m$^{-2/3}$ s$^{-5/6}$ for Schoof and 250 s$^{1/2}$ m$^{-1/2}$ for Budd).* I have difficulties to interpret the relative magnitude of standard deviations between two friction coefficients based on different physics and hence of different units. Thus, I would either report normalised values (as you seem to do in your graphs) or drop this sentence.

To address this, we now report these as normalized values (i.e. adjusted by the mean value of each friction coefficient). We have changed:

*This leads to a comparatively greater standard deviation in the Budd friction coefficient compared with that of Schoof (1240 kg$^{1/2}$ m$^{-2/3}$ s$^{-5/6}$ for Schoof and 250 s$^{1/2}$ m$^{-1/2}$ for Budd).*
to
*This leads to a greater standard deviation in the Budd friction coefficient compared with that of Schoof when both are normalized to their respective means (0.228 for Schoof and 0.385 for Budd).*

line 148 *Despite this, the Budd friction coefficient is generally smoother than the Schoof friction coefficient, which may be a consequence of the choice of the Tikhonov regularisation coefficient used in the inversion procedure.* For the reader, I think it would be beneficial if you could explain how exactly you determined the optimal regularisation parameter in a – at least for me – difficult to interpret L-curve given in Figure C1? Was, for instance, the relation between misfit and regularisation smaller as in Schoof?

We use the same strategy for picking the optimal regularization parameter as in Schoof. To clarify this we have added in Appendix B: *Ideally $c_t$ is chosen such that any smaller value of $c_t$ will not have a significant affect on $c_a \mathcal{J} a + c_l \mathcal{J}_l$.* Where these cost functions are described as per our response to your comment on line 77.

line 166 *...irrespective of the degree of regularization (Appendix C).* What exactly do you

mean by "degree of regularization"?

Changed *degree of regularization* to *magnitude of the Tikhonov regularization coefficient.*

line *Our study uses inverse methods to calculate the friction coefficient for a given* $\tau_b/N_G$. Is this a hint that you invert for traction and then interpret in terms of the specific friction law?

This was poorly worded, we mean that we invert with both $\tau_b$ and $N_G$ given. This has been changed to: *Our study uses inverse methods to calculate the basal friction coefficient for a given* $\tau_b$ *and* $N_G$.

line 285 *That is, using the Schoof law, regions of lower effective pressures tend to also have lower simulated basal friction and faster flow – evidence for the controlling role of the hydrological system.* Is that not also the case for Budd-sliding? To my understanding, the main difference is the more complex relation in the regularised Coulomb law to the effective pressure, yet, Budd-law has an inverse proportional relation of the sliding speed to the effective pressure and should result in faster flow over lower pressures.

Yes, we meant lower basal friction coefficient which had a negative correlation with the effective pressure for the Budd simulations, not basal friction which is roughly the same to get a similar ice surface velocity match. The line has been changed to: *That is, using the Schoof friction law, regions of lower effective pressures (both* $N_G$ *and* $N_O$*) tend to also have lower basal friction coefficients – evidence for the controlling role of the hydrological system.*

line 303 *The domains of the ISSM and GlaDS models differ, with the GlaDS domain being a subset of the ISSM domain.* What was the main motivation to not make these domains the same? Is it so expensive to run GlaDS on a wider domain? Or are there issues with boundaries? Would that not be worth it to get rid of all these extrapolation issues?

The GlaDS domain differs from the ISSM domain since GlaDS cannot run on ice shelves and because the limited number of nodes available for GlaDS runs (20,000) requires restricting the domain to the primary hydrological outlets of Denman and Scott glaciers. This node limitation will not be an issue for future iterations of GlaDS model outputs as we are aiming

to switch over to the parallelised ISSM GlaDS environment.

Furthermore, the GlaDS domain was set up based on the hydrological catchment calculated from overburden potential, which is different than the ice flow catchment. To clarify, we have changed:

*The domains of the ISSM and GlaDS models differ, with the GlaDS domain being a subset of the ISSM domain.*
to
*The domains of the ISSM and GlaDS models differ, with the GlaDS domain being a subset of the ISSM domain due to the differing subglacial hydrological and ice catchments and limits to the GlaDS domain size requiring restriction to the primary hydrological outlets of Denman and Scott Glaciers.*

Figure B1 To me the coloured lines have a different colour to the ones in the legend, such that I am not able to retrieve significant information out of this graph. Also, please explain the accumulated probabilities at the lower and upper end of the spectrum (I guess it is because of the cap).

The figure has been changed so that it is composed of line graphs, not overlaid translucent bar graphs. We now also include in Appendix B the following explanation of the accumulated probabilities: *The accumulated probability at the lower and upper bounds in Fig. B1 correspond to the regions of low/negative effective pressure near subglacial lakes and troughs and the area of negative effective pressure upstream of the subglacial lake feeding the Denman and Scott troughs (Fig 2v) respectively.*

line 345 *We use inverse methods to calculate the basal friction coefficient $\alpha$ from the Budd friction law and the ice rigidity in the Glen flow law. The inverse method works to reduce the mismatch between the simulated and observed velocities, here taken from MEaSUREs v2 (Rignot et al., 2011; Rignot, 2017), by minimising a cost function that includes both linear and logarithmic velocity misfit components.* I (and perhaps some of the readers) would benefit from having the whole cost-function including regularisation terms being written out as equation in order to easier interpret what you are doing in your inversion. It would add to make your experiment more reproducible. I am also confused on how you introduce the inversion of rigidity (e.g., do you use a penalty term for deviation to prior?). This links to the third main point of critics.

The cost functions have been added to Appendix B where the order of the inversions is described in detail. See our response to your comment on line 77.

Figure C1 and C2 A suggestion to improve the information for the reader: Perhaps you can distinctively mark the parameter configuration of the run that finally was chosen to provide the optimal inversion according to your L-curve analysis.

The marker for the chosen parameter has been enlargened, turned red, and turned into a diamond.

line 361 ..., *and $C_{max}$ is Iken's bound, here $C_{max} = 0.8 m^{-1/3} s^{1/3}$ . As Iken's bound is related to the roughness of the bed (Schoof, 2005),* can you please explain how the exact value reported here came to be? Maybe by backing it up with information on properties of the glacier substrate?

Thank you for the comment. Maritati et al., (2016) suggests that the bed is rough in the high elevation areas of the bed and smooth in the low elevation areas of the bed. We decided that the best way to choose a value of $C_{\max}$ would be to perform a sensitivity analysis. We have now performed a sensitivity analysis with values of $C_{\max} = 0.5, 0.6, 0.7, 0.8$. We found that each value of $C_{\max}$ produced the same qualitative conclusions we came to using a value of $C_{\max} = 0.8$. That is, the Schoof friction law produced more desirable results than the Budd friction law and the GlaDS effective pressure produced more desirable results than $N_O$. We decided that a value of $C_{\max} = 0.7$ was the best choice to go with and we have changed this throughout the manuscript. The results of the sensitivity analysis are put into Appendix A5.

*Iken's bound, $C_{max}$, mathematically describes the idea that the bed can support a maximum stress (Iken, 1981; Schoof, 2005; Gagliardini et al., 2007). In the Schoof friction law (Eq. 7), $\tau_b$ cannot exceed $C_{max}N$, where $C_{max}$ represents rheological properties of the till (Brondex et al., 2019) and ranges between 0.17 and 0.84 (Cuffey and Paterson, 2010). To determine an appropriate value of $C_{max}$ we test the effect of using $C_{max} = 0.5, 0.6, 0.7,$ and 0.8 on the Schoof basal friction coefficient using $N_G$ and $N_O$.*

*We arrive at the same qualitative conclusions as in the main text for all four values of $C_{max}$.*

*In all cases, the variance of the normalized basal friction coefficient is smaller for $N_G$ than for $N_O$, and it is smaller than the variance of the normalized Budd basal friction coefficient. Although the results for $C_{max} = 0.5, 0.6$, and 0.7 are similar, there is a comparatively large decrease in the mean of the basal friction coefficient and increase in the variance of the normalized basal friction coefficient between $C_{max} = 0.7$ and $C_{max} = 0.8$ for both $N_G$ and $N_O$. For lower values of $C_{max}$, a region of higher basal friction coefficient centered around $2200\,km$ easting and $-400\,km$ northing (Fig. A5a) develops in the $N_G$ simulations. Solving Eq. (7) for $C$ yields*

$$C = \frac{\left(\frac{\tau_b}{|u_b|^{m-1}u_b}\right)^{m/2}}{\left(1 - \left(\frac{\tau_b}{C_{max}N}\right)^{1/m}\right)^{m/2}}, \tag{A1}$$

*where we see that in the limit where $\tau_b \to C_{max}N$ then $C \to \infty$. The region of high basal friction coefficient for lower $C_{max}$ also has low effective pressure (Fig. 2a), resulting from $\tau_b$ approaching $C_{max}N$ for larger values of $N$ and the basal friction coefficient compensating this. To prevent potentially infinitely increasing values of the basal friction coefficient it is possible to increase the cap on the effective pressure, but again, this would reduce the area over which the GlaDS data is used and the effect of using modeled hydrology will be less impactful.*

*The Schoof friction law is a regularized Coulomb friction law which tends towards a Weertman sliding regime when $\tau_b << C_{max}N$ and a Coulomb sliding regime when $\tau_b >> C_{max}N$. Hence, the value of $C_{max}$ will have an effect on what physical processes are being represented in the Schoof friction law. Figure F2 shows $\tau_d/(C_{max}N)$ for various values of $C_{max}$, where values close to zero correspond to a Weertman sliding regime and values close to one correspond to a Coulomb sliding regime. The choice of $C_{max}$ appears to have little effect on where each of the Weertman and Coulomb sliding regimes occur, with distinct locations between the two for all values of $C_{max}$. This suggests that the choice of $C_{max}$ will not have a significant impact on which physical processes are being represented throughout the domain and does not justify the use of one value of $C_{max}$ over another.*

line 368 *In these regions, the inverse method compensates by increasing the friction coefficient upstream of the anomalously low effective pressure, leading to an underestimate of surface speeds there compared with the observations. The surface speeds are also generally overestimated in the region of vanishing shear stresses.* This links to my suggestion to better discuss the implication of the SSA approximation. Could it be that this effect is pronounced by the

fact that stress bridging between low and higher friction region is not represented in the model/approximation?

Morlighem et al. (2010) compared basal friction coefficients calculated using inversion and three different ice flow models: full Stokes (FS), the Blatter-Pattyn approximation (BP; Blatter, 1995; Pattyn, 2003), and the Shallow Shelf Approximation (SSA; MacAyeal, 1989). They found that all three ice flow models produce similar results for most of the Pine Island Glacier catchment. The greatest differences occur in the $\sim$100 km region upstream of the grounding line, where bridging stresses are not negligible, with the result that both the BP and SSA models tend to overestimate basal drag there compared with the FS model. We have yet to runs FS over our domain of study. However, we note that the effect of ignoring bridging stresses in increasing the basal friction coefficient is common across the friction laws estimated in our study. The implications of this choice would be more significant in a prognostic simulation – with grounding line migration – than in the diagnostic experiments conducted here.

line Figure C4 To me, this figure is not clear. I see 11 different colours in the graph and only 4 in the legend - I do not get a clear idea on which curve represents what probability distribution.

The graph has been changed to a line graph where it should be easier to identify the 4 distributions.

line 376 *That is, 98% of the effective pressure in the ISSM simulation is derived directly from the GlaDS simulated effective pressure.* I would drop that to the previous sentence somewhat redundant statement.

This has been merged into the sentence prior.

line 415

$$\overrightarrow{\nabla}\phi = \rho_i g f_B(H)\overrightarrow{\nabla}H + \rho_w g\overrightarrow{\nabla}B, \tag{E6}$$

$$f_B(H) = r_l \frac{1 + (1 - m)(H/\tilde{H})^m}{(1 + (H/\tilde{H})^m)^2}. \tag{E7}$$

*Here, $f_B(H)$ is a dimensionless factor which describes the extent to which ice thickness gradients play a role in the hydraulic potential gradient.* To me it seems that the factor $f_B(H)$ is equivalent to what in basic literature is called "flotation fraction" (e.g., see Chapter 6 in Cuffey and Patterson, 2010), which you depict for GlaDS result in Figure 2c. If you agree, please, try to make that connection for the reader.

In Chapter 6 of Cuffey and Paterson (2010) a spatially uniform fraction of flotation is assumed. Here the fraction of flotation varies spatially and so the gradient of the water pressure includes a contribution from the gradient of the fraction of flotation. This is therefore a more physically realistic application of hydraulic potential estimates. Because the gradient of the fraction of flotation is not zero, $f_B$ is not the fraction of flotation. A change to the manuscript described in our response to your below comment has been made to clarify this.

line 417 *It is seen that in the regime of larger ice thickness $f_B(H)$ goes to 0, and the gradient in the bed elevation becomes the sole control on the direction of water flow; in the regime of small ice thickness $f_B(H)$ goes to $r_l$ , and gradients in the ice thickness become of similar importance to gradients in the bed elevation. This is a more intuitive picture of the subglacial hydrological system, where water can flow throughout the entire domain, and where flow is dependent on both the basal topography and the ice thickness, as is expected.* As you claim that this is expected, in my view that would need to be backed up by reference(s). Provided, we agree that $f_b$ is equivalent to the flotation factor, for me it is even somewhat counter intuitive. Following standard literature (e.g., see Chapter 6 in Cuffey and Patterson, 2010) the flotation fraction has to go down to $f_B = 0.56$ for bedrock gradients (and not, actually, bedrock elevation itself) to reach same influence as surface (not thickness, though) gradients. In Figure 2c, though, the whole region west of $2200 \, \text{km}$ – which I connect to large thickness (although you do not provide a graph with ice thickness) – seems to be very close to a fully pressurised hydrological system (i.e. $f_B(H) \approx 1$), which would make me expect that surface gradients would almost by an order of magnitude dominate the flow direction of water. I, though, have difficulties to relate thickness to bedrock gradients, as the earlier are partly defined by the latter, i.e., $\nabla H = \nabla(S + B)$, if $S$ is the ice-surface elevation. To sum up: I do not see your statement that $f_B(H) \to 0$ for thick parts of the sheet is reflected in Figure 2c

and neither in Figure 3a. A graph depicting $f_B$ and ice thickness might help me and perhaps some readers to understand.

Thank you for your comment, after looking into this, it appears that we have made a mistake in the calculation of $f_B$ which we have corrected.

$$f_B(H) = 1 - (1 - r_l)\frac{1 + (1 - x)(H/\tilde{H})^x}{(1 + (H/\tilde{H})^x)^2} \tag{E7}$$

$f_B$ is not the fraction of flotation as we have explained in our response to the previous comment, but is a quadratic function of the fraction of flotation ($f_w$), with values larger than $f_w$ for $f_w \in [r_l, 1]$ so surface gradients play a more important role in the hydraulic potential gradient for the empirical parameterization than when a spatially uniform fraction of flotation is considered. We present new analysis relating $f_B$ to the fraction of flotation, updating the manuscript as follows:

$$\overrightarrow{\nabla}\phi = \rho_i g f_B(H)\overrightarrow{\nabla}H + \rho_w g \overrightarrow{\nabla}B = \rho_i g\left(f_B(H)\overrightarrow{\nabla}S + \left(\frac{\rho_w}{\rho_i} - f_B(H)\right)\overrightarrow{\nabla}B\right) \tag{G6}$$

$$f_B(H) = 1 - (1 - r_l)\frac{1 + (1 - x)(H/\tilde{H})^x}{(1 + (H/\tilde{H})^x)^2} \tag{G7}$$

*Here, $f_B(H)$ is a dimensionless factor that describes the extent to which ice thickness gradients play a role in the hydraulic potential gradient. Cuffey and Paterson (2010) considered the gradient of the hydraulic potential with a spatially constant fraction of flotation and arrive at the form of Eq.(C6) with $f_B(H)$ replaced by the fraction of flotation, $f_w$. Here, $f_B$ is not the fraction of flotation but is related to it via Eq. (C8) for $f_w \geq r_l$.*

$$f_B = -\frac{xr_l}{1 - r_l} + \frac{2 + (x - 1)(1 + r_l)}{1 - r_l}f_w - \frac{x}{1 - r_l}f_w^2. \tag{G8}$$

*$f_B$ is quadratic in $f_w$ and $f_B \geq f_w$ for $f_w \in [r_l, 1]$. This means that for $N_E$, gradients in surface elevation play a more important role in water routing than when a constant fraction of flotation is considered. The importance of surface elevation gradients reach a maximum when $f_w = 0.924$, corresponding to $f_B = 1.04$, values of $f_w$ above this threshold will actually result in smaller values of $f_B$ and a decreased role in the surface elevation gradient which is at odds with the analysis of Cuffey and Paterson (2010). Modeling (Dow*

*et al, 2014, 2020) outputs, including those from our Denman analysis, suggest that the fraction of flotation is not uniform (Fig. 6). Compared to using $N_O$, this parameterization provides a more intuitive picture of the subglacial hydrological system, where water can flow throughout the entire domain, and where flow is dependent on both the basal topography and the ice thickness/surface topography, as is expected.*

**References**

H. Åkesson, M. Morlighem, M. O'Regan, and M. Jakobsson. Future Projections of Petermann Glacier Under Ocean Warming Depend Strongly on Friction Law. *Journal of Geophysical Research: Earth Surface*, 126(6):e2020JF005921, 2021. ISSN 2169-9011. 10.1029/2020JF005921.

F. Baldacchino, M. Morlighem, N. R. Golledge, H. Horgan, and A. Malyarenko. Sensitivity of the Ross Ice Shelf to environmental and glaciological controls. *The Cryosphere*, 16(9): 3723–3738, 2022. ISSN 1994-0424.

M. Bougamont, P. Christoffersen, S. F. Price, H. A. Fricker, S. Tulaczyk, and S. P. Carter. Reactivation of Kamb Ice Stream tributaries triggers century-scale reorganization of Siple Coast ice flow in West Antarctica. *Geophysical Research Letters*, 42(20):8471–8480, 2015. ISSN 0094-8276.

J. Brondex, O. Gagliardini, F. Gillet-Chaulet, and G. Durand. Sensitivity of Grounding Line Dynamics to the Choice of the Friction Law. *Journal of Glaciology*, 63(241):854–866, 2017. 10.1017/jog.2017.51.

J. Brondex, F. Gillet-Chaulet, and O. Gagliardini. Sensitivity of Centennial Mass Loss Projections of the Amundsen Basin to the Friction Law. *The Cryosphere*, 13(1):177–195, 2019.

Y. Choi, M. Morlighem, E. Rignot, and M. Wood. Ice dynamics will remain a primary driver of Greenland ice sheet mass loss over the next century. *Nature Communications Earth Environment*, 2(1), 2021. ISSN 2662-4435.

S. J. Cook, P. Christoffersen, and J. Todd. A Fully-Coupled 3D Model of a Large Greenlandic Outlet Glacier with Evolving Subglacial Hydrology, Frontal Plume Melting and Calving. *Journal of Glaciology*, 68(269):486–502, June 2022. ISSN 0022-1430, 1727-5652. 10.1017/jog.2021.109.

K. Cuffey. Burlington, MA.

C. Dow, M. Werder, G. Babonis, S. Nowicki, R. Walker, B. Csatho, M. Morlighem, C. F. Dow, M. A. Werder, G. Babonis, S. Nowicki, R. T. Walker, B. Csatho, and M. Morlighem. Dynamics of Active Subglacial Lakes in Recovery Ice Stream. *Journal of Geophysical Research*, 123(4):837–850, 2018. 10.1002/2017JF004409.

C. Dow, F. McCormack, D. Young, J. Greenbaum, J. Roberts, and D. Blankenship. Totten Glacier subglacial hydrology determined from geophysics and modeling. *Earth and Planetary Science Letters*, 531:115961–, 2020. ISSN 0012-821X.

C. F. Dow. The role of subglacial hydrology in Antarctic ice sheet dynamics and stability: a modelling perspective. *Annals of Glaciology*, pages 1–6, 2023. ISSN 0260-3055.

O. Gagliardini, D. Cohen, P. Raback, and T. Zwinger. Finite-Element Modeling of Subglacial Cavities and Related Friction Law. *Journal of Geophysical Research - Earth Surface*, 112 (F2):1–11, May 2007. ISSN 0148-0227. 10.1029/2006JF000576.

C. A. Greene. RAMP Radarsat Antarctic Mapping Project, 2022. URL https://www.mathworks.com/matlabcentral/fileexchange/52031-ramp-radarsat -antarctic-mapping-project.

C. A. Greene, D. E. Gwyther, and D. D. Blankenship. Antarctic Mapping Tools for Matlab. *Computers & Geosciences*, 104:151–157, jul 2017. 10.1016/j.cageo.2016.08.003. URL https://doi.org/10.1016\%2Fj.cageo.2016.08.003.

A. Iken. The Effect of the Subglacial Water Pressure on the Sliding Velocity of a Glacier in an Idealized Numerical Model. *Journal of Glaciology*, 27(97):407–421, 1981. ISSN 0022-1430.

A. Iken and R. A. Bindschadler. Combined Measurments of Subglacial Water-Pressure and Surface Velocity of Findelengletsher, Switzerland. *Journal of Glaciology*, 32(110):101–119, 1986. ISSN 0022-1430.

E. Kazmierczak, S. Sun, V. Coulon, and F. Pattyn. Subglacial Hydrology Modulates Basal Sliding Response of the Antarctic Ice Sheet to Climate Forcing. Preprint, Ice sheets/Numerical Modelling, Mar. 2022.

T. M. Kyrke-Smith, R. F. Katz, and A. C. Fowler. Subglacial hydrology as a control on emergence,scale, and spacing of ice streams. *Journal of Geophysical Research. Earth surface*, 120(8):1501–1514, 2015. ISSN 2169-9003.

D. R. MacAyeal. A tutorial on the use of control methods in ice-sheet modeling. *Journal of Glaciology*, 39(131):91–98, 1993. ISSN 0022-1430.

A. Maritati, A. R. A. Aitken, D. A. Young, J. L. Roberts, D. D. Blankenship, and M. J. Siegert. The tectonic development and erosion of the Knox Subglacial Sedimentary Basin, East Antarctica. *Geophysical Research Letters*, 43(20):10,728–10,737, 2016. ISSN 0094-8276.

F. S. McCormack, R. C. Warner, H. Seroussi, C. F. Dow, J. L. Roberts, and A. Treverrow. Modeling the Deformation Regime of Thwaites Glacier, West Antarctica, Using a Simple Flow Relation for Ice Anisotropy (ESTAR), volume = 127, year = 2022,. *Journal of Geophysical Research. Earth surface*, (3). ISSN 2169-9003.

M. Morlighem, E. Rignot, H. Seroussi, E. Larour, H. Ben Dhia, and D. Aubry. Spatial patterns of basal drag inferred using control methods from a full-Stokes and simpler models for Pine Island Glacier, West Antarctica. *Geophysical Research Letters*, 37(14): L14502–n/a, 2010. ISSN 0094-8276.

J. Mouginot, B. Scheuchl, and E. Rignot. Mapping of Ice Motion in Antarctica Using Synthetic-Aperture Radar Data. *Remote sensing*, 4(9):2753–2767, 2012. ISSN 2072-4292.

J. Mouginot, E. Rignot, B. Scheuchl, and R. Millan. Comprehensive Annual Ice Sheet Velocity Mapping Using Landsat-8, Sentinel-1, and RADARSAT-2 Data. *Remote sensing*, 9(4):364–, 2017. ISSN 2072-4292.

K. Poinar, C. F. Dow, and L. C. Andrews. Long-Term Support of an Active Subglacial Hydrologic System in Southeast Greenland by Firn Aquifers. *Geophysical Research Letters*, 46(9):4772–4781, 2019. ISSN 0094-8276.

E. Rignot, J. Mouginot, and B. Scheuchl. Ice Flow of the Antarctic Ice Sheet. *Science*, 333 (6048):1427–1430, Sept. 2011. ISSN 0036-8075. 10.1126/science.1208336.

E. Rignot, J. Mouginot, and B. Scheuchl. MEaSUREs InSAR-Based Antarctica Ice Velocity Map, Version 2 [Data Set]. Boulder, Colorado USA. NASA National Snow and Ice Data Center Distributed Active Archive Center., 2017. Accessed 06-10-2021.

C. Schoof. The Effect of Cavitation on Glacier Sliding. *Proceedings of the Royal Society A*, 461(2055):609–627, 2005. ISSN 1364-5021. 10.1098/rspa.2004.1350.

H. Seroussi, M. Morlighem, E. Rignot, A. Khazendar, E. Larour, and J. Mouginot. Dependence of century-scale projections of the Greenland ice sheet on its thermal regime. *Journal of Glaciology*, 59(218):1024–1034, 2013. ISSN 0022-1430.

H. Seroussi, S. Nowicki, E. Simon, A. Abe-Ouchi, T. Albrecht, J. Brondex, S. Cornford, C. Dumas, F. Gillet-Chaulet, H. Goelzer, N. R. Golledge, J. M. Gregory, R. Greve, M. J. Hoffman, A. Humbert, P. Huybrechts, T. Kleiner, E. Larour, G. Leguy, W. H. Lipscomb, D. Lowry, M. Mengel, M. Morlighem, F. Pattyn, A. J. Payne, D. Pollard, S. F. Price, A. Quiquet, T. J. Reerink, R. Reese, C. B. Rodehacke, N.-J. Schlegel, A. Shepherd, S. Sun, J. Sutter, J. Van Breedam, R. S. W. van de Wal, R. Winkelmann, and T. Zhang. initMIP-Antarctica: An ice sheet model initialization experiment of ISMIP6. *The Cryosphere*, 13(5):1441–1471, May 2019. ISSN 1994-0416. 10.5194/tc-13-1441-2019.

S. Smith-Johnsen, N. Schlegel, B. Fleurian, and K. H. Nisancioglu. Sensitivity of the Northeast Greenland Ice Stream to Geothermal Heat. *Journal of Geophysical Research. Earth surface*, 125(1), 2020. ISSN 2169-9003.

J. M. van Wessem, W. J. Van De Berg, B. P. Y. Noël, E. Van Meijgaard, C. Amory, G. Birnbaum, C. L. Jakobs, K. Krüger, J. Lenaerts, S. Lhermitte, S. R. M. Ligtenberg, B. Medley, C. H. Reijmer, K. van Tricht, L. D. Trusel, L. H. van Ulft, B. Wouters, J. Wuite, and M. R. van den Broeke. Modelling the Climate and Surface Mass Balance of Polar Ice Sheets Using RACMO2: Part 2: Antarctica (1979-2016). *The Cryosphere*, 12(4): 1479–1498, 2018. 10.5194/tc-12-1479-2018.

H. Yu, E. Rignot, H. Seroussi, and M. Morlighem. Retreat of Thwaites Glacier, West Antarctica, over the next 100 Years Using Various Ice Flow Models, Ice Shelf Melt Scenarios and Basal Friction Laws. *The Cryosphere*, 12(12):3861–3876, 2018.

C. Zhao, R. M. Gladstone, R. C. Warner, M. A. King, T. Zwinger, and M. Morlighem. Basal friction of Fleming Glacier, Antarctica – Part 1: Sensitivity of inversion to temperature and bedrock uncertainty. *The Cryosphere*, 12(8):2637–2652, 2018. ISSN 1994-0424.

H. Åkesson, M. Morlighem, J. Nilsson, C. Stranne, and M. Jakobsson. Petermann ice shelf may not recover after a future breakup. *Nature Communications*, 13(1):2519–2519, 2022. ISSN 2041-1723.

---

## Referee Report (RR1)

**2$^{\text{nd}}$ Review of: Basal conditions of Denman Glacier from glacier hydrology and ice dynamics modeling**

August 20, 2023

**1    General impression**

The authors have addressed all of my comments from the previous review. Necessary corrections have been applied and most unclear parts explained. In my opinion, the manuscript has improved during this iteration and in general is good to be published.

The applied changes and additions, in particular the newly introduced notation of the melt rate (including the equation of GlaDS where it contributes) and the argumentation on justification of limiters in sliding velocities for me revealed a few new questions, I still would like to see to be addressed. I do not think they are of major concerns, yet, clear statements and more information on the melt-rates leading to the resulting water pressure are in my view needed to get a clearer picture. If this is come after, I recommend publication.

**2    Still open point(s) recommended be addressed**

The remaining issues I see with the current version for me boil down to not provided complete information on the **distribution of the basal melt-water production**, i.e., the water source passed to GlaDS input obtained from the initial runs done by Seroussi et al. (2019), that in the revised text now has been introduced with the symbol $m$ (which is somewhat unfortunately coinciding with the already used exponent in the sliding law). Although I asked about clarification on the water source input in my previous review, it might not have come clear that in my view it would be beneficial to show the melt-water distribution in a figure, such that the reader would get a better understanding on the distribution of the water sources and the hydrological balance in general. With this information, one could get a better estimation on how water- and consequently also effective pressure correlate with the supply of water. This, in my opinion, also would help to evaluate the newly introduced statement for justification of the unaffected

water pressure distribution by the sliding velocity cap (see in 3 *Detailed comments*).

**3   Detailed comments**

I am referring to line numbers in the file (`tc-2023-28-manuscript-version4.pdf`)

**rebuttal letter**   In the rebuttal you write:   *In terms of the GlaDS setup, we use standard basal velocity and water input from the JPL ISSM ISMIP model outputs of a thermal steady-state simulation (Seroussi et al., 2020), . . . .*
I did not find such a reference (the year), neither in the revised paper, nor in the reference list of the rebuttal letter. Did you mean (Seroussi et al., 2019)?

**line 72**   *As discussed in Dow (2023), when the system is overconstricted the pressures are unrealistically high and the model ceases to converge. When the system is underconstricted the pressures are below ice overburden for much of the domain. While there is some variation within the range of acceptable pressures, the output we present is the median and therefore is the most appropriate for representing the hydrology pressure in ice sheet dynamics equations.*
Browsing through the reference (Dow, 2023), I cannot really learn what exactly the terms *over-* and *underconstricted* mean. Can you please explain? Is it in terms of imposed water supply or conditions of the hydro-potential at the boundaries or constraints on the channels? What I mainly conclude from Dow (2023) is that it is difficult to get a working set of parameters for Antarctic subglacial water sheets (which I can confirm from our own attempts with GlaDS) and - instead of testing out the whole parameter space - those cutoffs are introduced to get converged results. For me, that deserves more justification or explanation. I would like to have spelled out how you define an *acceptable waterpressure* in lack of available measurements and observations of Antarctic subglacial hydrological systems?

**line 90**   *Tests of similar caps for model runs at Helheim Glacier (Poinar et al., 2019) demonstrate it has little impact on the model results.*
Can you please provide more evidence/argumentation why the situation at a Greenlandic outlet glacier should transfer to a system at Antarctica? I see differences, for instance, in terms of water sources. In Greenland runoff has certainly an impact that even can introduce a strong seasonal variation (hence question the assumption of a steady state), whereas I would expect friction heating to be the dominating source of water production for an outlet glacier in Antarctica (see e.g.,

Dow, 2023). As mentioned before, a picture showing the spatial distribution of water supply could help with getting more insight and justification of the applied analogy between these two ice sheets.

line 104, eq. (5) In the new version you introduce the effective viscosity $\mu$, but in the component-wise SSA equations (3) and (4) – which I would combine to one equation number – before you use $\bar{\mu}$. Can you please either explain, how you come from one to the other or correct the annotation? I then presume that the rigidity $B$ is the field you invert for. Please, add the symbol rather than the wording *Ice Rigidity* to the annotation of Figure B5.

line 390 GlaDs $\rightarrow$ GlaDS

references Quite a few of the references, like Dow (2023), are missing the DOI - please provide those.

**References**

Dow, Christine F. (2023). "The role of subglacial hydrology in Antarctic ice sheet dynamics and stability: a modelling perspective". In: *Annals of Glaciology*, 1–6. DOI: `10.1017/aog.2023.9`.

Seroussi, H et al. (2019). "initMIP-Antarctica: An Ice Sheet Model Initialization Experiment of ISMIP6". In: *The Cryosphere* 13, 1441–1471. DOI: `10.5194/tc-13-1441-2019`.

---

## Referee Report (RR2)

Thank you to the authors for their answers and for having taken into consideration my remarks and suggestions.

I really appreciated that you took the time to run additionnal simulations using the Budd friction law with m=1/3 to answer to my first general comment that related to the choice of the m=1.

I also noticed that you modified the figures as suggested. Here, I found the addition of the sensitivity tests with various $C_{max}$ values really interesting and I am convinced this will be an added value to your manuscript.

From my side, this paper is ready for publication, although I still have a few specific comments (mainly typo's and suggestions for symbols and references) for the revised manuscript, which I listed below in order of their appearance.

Note : I noticed a few differences between your italic responses and the revised manuscript. Here, while commenting I have always considered the revised manuscript. Finally, I also noticed some layout issues in the revised manuscript, which are probably due to the « track changes format », but which I nevertheless report here for clarity.

Figure 1 (b) : typo: Ice **s**urface (s is missing)

Eq 1 : maybe use another letter than m, which is already used for another variable in Eq. (6)-(7)

Eq 2 : maybe $h_w$ is better than h for the hydrology sheet thickness

L74 : ice overburden **pressure**

L87 : « Blatter-Pattyn approximation to the full Stokes equations **(Blatter, 1995, Pattyn, 2003)** »

- Blatter, H.: Velocity and stress fields in grounded glaciers: a simple algorithm for including deviatoric stress gradients, J. Glaciol., 41, 333–344, https://doi.org/10.3189/S002214300001621X, 1995.
- Pattyn, F.: A new three-dimensional higher-order thermomechanical ice-sheet model: basic sensitivity, ice-stream development and ice flow across subglacial lakes, J. Geophys. Res., 108, 2382, https://doi.org/10.1029/2002JB002329, 2003.

L101 : layout issue

Eq 5 : maybe use another letter than B for the ice rigidity, since this is already used for the bedrock elevation

L125 : layout issue

L154 : specify the value of the water density (because Akesson et al, 2021 used sea water density value and Yu et al, 2018 used fresh water density value in the No calculation)

L197 : typo : (Fig. 1a**)**.

L229 : typo : GlaD**S**

L247 : my apologize, as opposed to what I suggested in my original review, Budd and Jensen (1987) is a good reference when N is expressed as an hydrological potential. Huybrechts (1990) used an N corresponding to an « height above buoyancy » (Budd et al., 1987 ; Van der Veen, 1987 ; referred to as HAB in Pattyn, 1996), which fits more with the Brondex et al., (2017), $N_B$, defined in L257

L253 : (Fig. 3**b**) → and place it : [...] below sea level **(Fig. 3b)**, yielding [...]

L273 : typo : start with the ',' of the L272's end

L345-L352 : repetiton of [...] H is defined in Eq. (9) and x is defined in Eq. (10) [...]

Figure 6 (e) : typo : [...] GlaDS output effective epressure [...]

L390 : typo : GlaD**S**

L428 : layout issue

L448 : typo : (Fig**. 2d (v)**)

L462 : remove Kazmierczak et al., 2022

L462-463 : layout issues

L466 : typo : var~ia~iance

L481-482 : layout issues

L526 : same comment than in Eq 5 and B is written with a – on top not like in the Eq. (5)

L546 : the number of the figure showing the final rigidities is missing (I imagine it is Fig. B5)

Table B1 : typo : **V**alue I4- Budd

Table B1 caption : typo : '-' is missing after I5

Fig B5 : layout issue, I cannot read the caption

L773 : Akesson et al. 2022 is not at the right place in the bibliography (it is at the end of the final bibliography)

---

## Author Response (AR2)

**Reviewer Comment Response**

**1   Editor Comments**

Public justification (visible to the public if the article is accepted and published):

Dear Koi McArthur and co-author,

In view of the reports from the referees, I am happy to recommend your manuscript for publication subject to minor revisions. Please go through the recommendations from the referees and make adjustments in a revised version of your manuscript as appropriate.

In response to the request from referee 1 re. the need for more information on the distribution of the basal meltwater production, I suggest this could be added as Fig. 1c. I have two technical comments that I outline below. I look forward to receiving a new version of your submissions. Best,

Nanna B Karlsson

Thank you Dr. Karlsson. We have added the meltwater added to GlaDS as Fig 1c. and your technical comments are addressed below. We keep the reviewer comments in black and our responses in blue. New text or changes to to manuscript are italicized.

Additional private note (visible to authors and reviewers only):

Referee 2 notes that Åkesson et al. 2022 should be at the beginning of the manuscript. This is not correct. The letter Å goes at the end and thus your placement is correct.

Thank you, we have left it at the end.

L 306: Word missing "... to the full Stokes is used..."

This has been changed to *...to the full Stokes equations is used...*

Let me know if you cannot access the second reports from the referees, and I will send them to you.

**2   Reviewer 1 Comments**

Thank you to the authors for their answers and for having taken into consideration my remarks and suggestions.

I really appreciated that you took the time to run additionnal simulations using the Budd friction law with $m = 1/3$ to answer to my first general comment that related to the choice of the $m = 1$.

I also noticed that you modified the figures as suggested. Here, I found the addition of the sensitivity tests with various $C_{\max}$ values really interesting and I am convinced this will be an added value to your manuscript.

From my side, this paper is ready for publication, although I still have a few specific comments (mainly typo's and suggestions for symbols and references) for the revised manuscript, which I listed below in order of their appearance.

Note : I noticed a few differences between your italic responses and the revised manuscript. Here, while commenting I have always considered the revised manuscript. Finally, I also noticed some layout issues in the revised manuscript, which are probably due to the $<<$ track changes format $>>$, but which I nevertheless report here for clarity.

Thank you for your comments, we believe that they have improved the quality of the manuscript. We are glad to hear that you believe that the manuscript is almost ready for publication, we address your following comments below.

Figure 1 (b) : typo: Ice surface (s is missing)

Changed.

Eq 1 : maybe use another letter than m, which is already used for another variable in Eq. (6)-(7)

the variable $m$ has now been changed to $\eta$ throughout the manuscript.

Eq 2 : maybe $h_w$ is better than $h$ for the hydrology sheet thickness

$h$ is what is used in Werder et al., (2013), so we keep it here.

L74 : ice overburden **pressure**

Changed.

L87 : $<<$ Blatter-Pattyn approximation to the full Stokes equations **(Blatter, 1995, Pattyn, 2003)** $>>$

- Blatter, H.: Velocity and stress fields in grounded glaciers: a simple algorithm for including deviatoric stress gradients, J. Glaciol., 41, 333–344, `https://doi.org/10.3189/S002214300001621X`, 1995.

- Pattyn, F.: A new three-dimensional higher-order thermomechanical ice-sheet model: basic sensitivity, ice-stream development and ice flow across subglacial lakes, J. Geophys. Res., 108, 2382, `https://doi.org/10.1029/2002JB002329`, 2003.

Changed.

L101 : layout issue

This is a track changes layout issue.

Eq 5 : maybe use another letter than $B$ for the ice rigidity, since this is already used for the bedrock elevation

We have now changed to $\tilde{B}$ instead of $B$.

L125 : layout issue

This is a track changes layout issue.

L154 : specify the value of the water density (because Akesson et al, 2021 used sea water density value and Yu et al, 2018 used fresh water density value in the No calculation)

We have now specified that we used sea water (this ensures complete hydrological connectivity to the ocean at the grounding line). Line 153 now says $\rho_w$ *is the density of sea water*.

L197 : typo : (Fig. 1a).

Changed.

L229 : typo : GlaD**S**

Changed.

L247 : my apologize, as opposed to what I suggested in my original review, Budd and Jensen (1987) is a good reference when $N$ is expressed as an hydrological potential. Huybrechts (1990) used an $N$ corresponding to an $<<$ height above buoyancy $>>$ (Budd et al., 1987 ; Van der Veen, 1987 ; referred to as HAB in Pattyn, 1996), which fits more with the Brondex et al., (2017), $N_B$, defined in L257

We are unsure what the reviewer is looking for here. We have removed the Huybrechts reference and we made sure that all the references use $N_O$ as an effective pressure, not a hydraulic potential.

L253 : (Fig. 3b) $\rightarrow$ and place it : [. . . ] below sea level **(Fig. 3b)**, yielding [. . . ]

Changed.

L273 : typo : start with the '.' of the L272's end

This is a track changes layout issue.

L345-L352 : repetiton of [...] H is defined in Eq. (9) and x is defined in Eq. (10) [...]

The second occurrence has been removed. Line 342 now reads *Here, $\rho_i$ is the density of ice, g is the gravitational acceleration, H is ice thickness, and the effective pressure is given by Eq. (7).*

Figure 6 (e) : typo : [...] GlaDS output effective pressure [...]

Changed.

L390 : typo : GlaD**S**

Changed.

L428 : layout issue

This is a track changes layout issue.

L448 : typo : (Fig. 2d (v))

Changed.

L462 : remove Kazmierczak et al., 2022

Changed.

L462-463 : layout issues

This is a track changes layout issue.

L466 : typo : variance

Changed.

L481-482 : layout issues

This is a track changes layout issue.

L526 : same comment than in Eq 5 and $B$ is written with a – on top not like in the Eq. (5)

This has been asnwered in response to your Eq. (5) comment.

L546 : the number of the figure showing the final rigidities is missing (I imagine it is Fig. B5)

This is a track changes layout issue.

Table B1 : typo : **V**alue I4- Budd

Changed.

Table B1 caption : typo : '-' is missing after I5

The fifth inversion has the same cost function coefficients for both the Budd and Schoof runs so we do not need to distinguish between them.

Fig B5 : layout issue, I cannot read the caption

This is a track changes layout issue.

L773 : Akesson et al. 2022 is not at the right place in the bibliography (it is at the end of the final bibliography)

This reference is in the correct location confirmed by Dr. Karlsson in editing.

**3 Reviewer 2 Comments**

2nd Review of: Basal conditions of Denman Glacier from glacier hydrology and ice dynamics modeling

August 20, 2023

**General impression**

The authors have addressed all of my comments from the previous review. Necessary corrections have been applied and most unclear parts explained. In my opinion, the manuscript has improved during this iteration and in general is good to be published.

The applied changes and additions, in particular the newly introduced notation of the melt rate (including the equation of GlaDS where it contributes) and the argumentation on justification of limiters in sliding velocities for me revealed a few new questions, I still would like to see to be addressed. I do not think they are of major concerns, yet, clear statements and more information on the melt-rates leading to the resulting water pressure are in my view needed to get a clearer picture. If this is come after, I recommend publication.

We appreciate your comments and agree that they have improved the quality of the manuscript. We address your comments on our new iteration of the manuscript below.

**2 Still open point(s) recommended be addressed**

The remaining issues I see with the current version for me boil down to not provided complete information on the distribution of the basal meltwater production, i.e., the water source passed to GlaDS input obtained from the initial runs done by Seroussi et al. (2019), that in the revised text now has been introduced with the symbol $m$ (which is somewhat unfortunately coinciding with the already used exponent in the sliding law). Although I asked about clarification on the water source input in my previous review, it might not have come clear that in my view it would be beneficial to show the melt-water distribution in a figure, such that the reader would get a better understanding on the distribution of the water sources and the hydrological balance in general. With this information, one could get a better estimation on how water- and consequently also effective pressure correlate with the supply of water. This, in my opinion, also would help to evaluate the newly introduced statement for justification of the unaffected water pressure distribution by the sliding velocity cap (see in 3 *Detailed comments*).

To avoid confusion we agree that it would be a good idea to not use the symbol $m$ for two different variables. We now denote the melt water input into GlaDS as $\eta$. We have included

a map of the water input to GlaDS in Fig. 1c. The regions of the greatest melt water production do not occur where ice is flowing faster than the cap of $800\,\mathrm{m\,a^{-1}}$ except within a small region of the Denman trough. The small overlap between the areas of melt water production and high ice velocity justify the use of the cap.

**3 Detailed comments**

I am referring to line numbers in the file (tc-2023-28-manuscript-version4.pdf) rebuttal letter In the rebuttal you write: *In terms of the GlaDS setup, we use standard basal velocity and water input from the JPL ISSM ISMIP model outputs of a thermal steady-state simulation (Seroussi et al.,2020), . . . .* I did not find such a reference (the year), neither in the revised paper, nor in the reference list of the rebuttal letter. Did you mean (Seroussi et al., 2019)?

Yes, we mean (Seroussi et al., 2019). Apologies for the confusion.

line 72 *As discussed in Dow (2023), when the system is overconstricted the pressures are unrealistically high and the model ceases to converge. When the system is underconstricted the pressures are below ice overburden for much of the domain. While there is some variation within the range of acceptable pressures, the output we present is the median and therefore is the most appropriate for representing the hydrology pressure in ice sheet dynamics equations.* Browsing through the reference (Dow, 2023), I cannot really learn what exactly the terms *over-* and *underconstricted* mean. Can you please explain? Is it in terms of imposed water supply or conditions of the hydro-potential at the boundaries or constraints on the channels? What I mainly conclude from Dow (2023) is that it is difficult to get a working set of parameters for Antarctic subglacial water sheets (which I can confirm from our own attempts with GlaDS) and - instead of testing out the whole parameter space - those cutoffs are introduced to get converged results. For me, that deserves more justification or explanation. I would like to have spelled out how you define an *acceptable waterpressure* in lack of available measurements and observations of Antarctic subglacial hydrological systems?

By overconstricted and underconstricted we refer to the ease at which water can flow through the hydrology system. Overconstricted means that it is difficult for the water to flow through the system due to a combination of low bedrock bump height and sheet conductivity and high water input. This leads to unrealistically pressurized water (water pressure is above ice overburden pressure for much of the domain), and a model which fails to converge. Underconstricted means that water flows through the hydrology system with greater ease due to relatively high sheet conductivity, bedrock bump height, and little water input. This leads to water pressures which are unrealistically low (much of the domain is below $50\,\%$ of overburden) which is unexpected for steady-state Antarctic systems that are not driven by

surface water input. We lack the ability to know exactly what the correct water pressure is given lack of measurements as you state, so extensive tuning exercises of basal sheet conductivity would not be particularly useful. Instead we use our best estimate which is where most of the domain is near overburden pressure (with water pressure generally >0.8 of overburden). It would be an interesting question to pursue whether small changes in the basal water pressure would have larger impacts on basal friction parameter applications but that is beyond the scope of this study. However, to clarify these issues we include in lines 72-80.

*As discussed in Dow (2023), when the system is overconstricted (i.e. it is difficult for water to flow through the hydrology system) the pressures are unrealistically high – much of the domain is above ice overburden pressure – and the model fails to converge. When the system is underconstricted (i.e. water flows through the hydrology system with ease) the pressures are well below ice overburden pressure for much of the domain (much of the domain is below 50 % of ice overburden pressure), which is unrealistically low for steady-state Antarctic systems that are not driven by surface water input. The variables controlling the constriction of the hydrology system are $k$, $h_r$, and $\eta$, with a more constricted system arising from larger $\eta$ and smaller $k$ and $h_r$. We test order of magnitude changes in $k$ to determine a suitable level of constriction of the system. While there is some variation within the range of acceptable pressures, the output we present is the median and therefore is the most appropriate for representing the hydrology pressure in ice sheet dynamics equations without further information from in situ measurements for example. Future work with full coupling of hydrology and ice dynamics can explore sensitivity to different distributed system inputs.*

line 90 *Tests of similar caps for model runs at Helheim Glacier (Poinar et al., 2019) demonstrate it has little impact on the model results.* Can you please provide more evidence/ argumentation why the situation at a Greenlandic outlet glacier should transfer to a system at Antarctica? I see differences, for instance, in terms of water sources. In Greenland runoff has certainly an impact that even can introduce a strong seasonal variation (hence question the assumption of a steady state), whereas I would expect friction heating to be the dominating source of water production for an outlet glacier in Antarctica (see e.g., Dow, 2023). As mentioned before, a picture showing the spatial distribution of water supply could help with getting more insight and justification of the applied analogy between these two ice sheets.

The caps applied to Helheim Glacier in Poinar et al. (2019) assumed winter conditions which would be steady state and comparable. However, the main point we wanted to make here

was that large velocities lead to cavity opening which is faster than the model is able to converge. This has to do with the model configuration, not the area being tested (Greenland vs Antarctica). Determining a workaround for applying this cap when ice velocities are large is a potential improvement for future modeling studies. We now have on lines 94-95: *Tests of similar caps for model runs of winter conditions at Helheim Glacier Poinar et al. (2019) demonstrate it has little impact on the effective pressure.*

line 104, eq. (5) In the new version you introduce the effective viscosity $\mu$, but in the component-wise SSA equations (3) and (4) – which I would combine to one equation number – before you use $\bar{\mu}$. Can you please either explain, how you come from one to the other or correct the annotation? I then presume that the rigidity $B$ is the field you invert for. Please, add the symbol rather than the wording *Ice Rigidity* to the annotation of Figure B5.

$\bar{\mu}$ is the depth averaged effective viscosity $\mu$ which we mention in the manuscript but before the explanation of what $\mu$ is. We have now combined the two SSA equations into a single equation and explain what $\mu$ and $\bar{\mu}$ are before the equations rather than after. On lines 104-106 we have:

*The inverse model uses the shallow-shelf approximation (SSA; MacAyeal, 1989; Morland, 1987) to the full Stokes equations, described in Eq. (3) with $\bar{\mu}$ the depth-averaged effective viscosity $\mu$ which is given in Eq. (4). The SSA is described in full in Larour et al. (2012).*

You are correct, $\tilde{B}$ is what we invert for. The caption of Figure B5 has now been changed to:

*Ice Rigidities from inversion ($\tilde{B}$, $Pa\,s^{1/3}$).* **(a)** *Ice rigidity from Schoof with the GlaDS effective pressure ($N_G$) run;* **(b)** *Ice rigidity from Schoof with the typically prescribed effective pressure ($N_O$) run;* **(c)** *Ice rigidity from Budd with the GlaDS effective pressure ($N_G$) run; and* **(d)** *Ice rigidity from Budd with the typically prescribed effective pressure ($N_O$) run.*

line 390 GlaDs → GlaDS

Changed.

references Quite a few of the references, like Dow (2023), are missing the DOI - please provide those.

We have now added DOIs for the papers that were missing them.

References Dow, Christine F. (2023). "The role of subglacial hydrology in Antarctic ice sheet dynamics and stability: a modelling perspective". In: Annals of Glaciology, 1–6. doi: 10.1017/aog.2023.9. Seroussi, H et al. (2019). "initMIP-Antarctica: An Ice Sheet

Model Initialization Experiment of ISMIP6". In: The Cryosphere 13, 1441–1471. doi: 10.5194/tc-13-1441-2019.

**References**

C. F. Dow. The role of subglacial hydrology in antarctic ice sheet dynamics and stability: a modelling perspective. *Annals of Glaciology*, pages 1–6, 2023. ISSN 0260-3055. 10.1017/aog.2023.9.

E. Larour, H. Seroussi, M. Morlighem, and E. Rignot. Continental scale, high order, high spatial resolution, ice sheet modeling using the Ice Sheet System Model (ISSM). *Journal of Geophysical Research*, 117(F01022):1–20, 2012. 10.1029/2011JF002140.

D. R. MacAyeal. A tutorial on the use of control methods in ice-sheet modeling. *Journal of Glaciology*, 39(131):91–98, 1993. ISSN 0022-1430. 10.3189/S0022143000015744.

L. W. Morland. Unconfined ice-shelf flow. In C. J. Van der Veen and J. Oerlemans, editors, *Dynamics of the West Antarctic Ice Sheet*, pages 99–116, Dordrecht, 1987. Springer Netherlands. ISBN 978-94-009-3745-1. https://doi.org/10.1007/978-94-009-3745-1_6.

K. Poinar, C. F. Dow, and L. C. Andrews. Long-term support of an active subglacial hydrologic system in southeast greenland by firn aquifers. *Geophysical Research Letters*, 46(9):4772–4781, 2019. ISSN 0094-8276. https://doi.org/10.1029/2019GL082786.

H. Seroussi, S. Nowicki, E. Simon, A. Abe-Ouchi, T. Albrecht, J. Brondex, S. Cornford, C. Dumas, F. Gillet-Chaulet, H. Goelzer, N. R. Golledge, J. M. Gregory, R. Greve, M. J. Hoffman, A. Humbert, P. Huybrechts, T. Kleiner, E. Larour, G. Leguy, W. H. Lipscomb, D. Lowry, M. Mengel, M. Morlighem, F. Pattyn, A. J. Payne, D. Pollard, S. F. Price, A. Quiquet, T. J. Reerink, R. Reese, C. B. Rodehacke, N.-J. Schlegel, A. Shepherd, S. Sun, J. Sutter, J. Van Breedam, R. S. W. van de Wal, R. Winkelmann, and T. Zhang. initMIP-Antarctica: An ice sheet model initialization experiment of ISMIP6. *The Cryosphere*, 13(5):1441–1471, May 2019. ISSN 1994-0416. 10.5194/tc-13-1441-2019.

M. A. Werder, I. J. Hewitt, C. G. Schoof, and G. E. Flowers. Modeling channelized and distributed subglacial drainage in two dimensions. *Journal of Geophysical Research*, 118: 1–19, 2013. ISSN 2169-9011. doi:10.1002/jgrf.20146,.

---

## Author Response (AR3)

**Reviewer Comment Response**

**1 Editor Comments**

Dear Koi McArthur and co-authors,

Thank you for the revised version of your manuscript and for addressing the comments from the referee thoroughly. I am happy to recommend the manuscript for publication in The Cryosphere (pending a few technical corrections that I list below).

Your work provides a convincing argument for the careful treatment of subglacial processes in model studies, and importantly the results you present will inform future studies of subglacial properties and ice-sheet dynamical behaviour.

Best,

Nanna B. Karlsson

Thank you Nanna, we are pleased to hear that you think our manuscript is almost ready for publication. We address your comments below in blue text.

Additional private note (visible to authors and reviewers only):

Two very minor notes:

- Eqs. 3 and 4: due to the way LaTeX displays track changes it is difficult for me to check that these equations are correct. Please double-check when you submit your revised version that the equations are correct.

Yes, sorry, we see that the track changes made the equation formatting very strange. We have confirmed that the equations are correct.

- I was a bit confused by the use of the word "hydrology" as an adjective, e.g., "hydrology system" or "hydrology pressure". From a grammatical point of view, I would expect the adjective of "hydrology" to be "hydrologic". Perhaps there is a distinction between the two terms that I am not aware of but please check.

We have changed all instances of "hydrology system" and "hydrology pressure" to "hydrologic system" and "hydrologic pressure" respectively.

Congratulations on a very nice manuscript.

Nanna